# First-Order Minimax Bilevel Optimization

**Yifan Yang**[*], **Zhaofeng Si**[*], **Siwei Lyu and Kaiyi Ji**
Department of Computer Science and Engineering
University at Buffalo
Buffalo, NY 14260
{yyang99, zhaofeng, siweilyu, kaiyiji}@buffalo.edu

## Abstract

Multi-block minimax bilevel optimization has been studied recently due to its great potential in multi-task learning, robust machine learning, and few-shot learning. However, due to the complex three-level optimization structure, existing algorithms often suffer from issues such as high computing costs due to the second-order model derivatives or high memory consumption in storing all blocks' parameters. In this paper, we tackle these challenges by proposing two novel fully first-order algorithms named FOSL and MemCS. FOSL features a fully single-loop structure by updating all three variables simultaneously, and MemCS is a memory-efficient double-loop algorithm with cold-start initialization. We provide a comprehensive convergence analysis for both algorithms under full and partial block participation, and show that their sample complexities match or outperform those of the same type of methods in standard bilevel optimization. We evaluate our methods in two applications: the recently proposed multi-task deep AUC maximization and a novel rank-based robust meta-learning. Our methods consistently improve over existing methods with better performance over various datasets.

## 1 Introduction

In this paper, we study a general multi-block minimax bilevel optimization problem given by

$$\min_{x \in \mathbb{R}^{d_x}} \max_{y \in \mathbb{R}^{d_y}} F(x, y, \mathbf{z}^*) := \frac{1}{n} \sum_{i=1}^{n} f_i\big(x, y, z_i^*(x)\big) = \frac{1}{n} \sum_{i=1}^{n} \mathbb{E}_\xi \big[ f_i\big(x, y, z_i^*(x); \xi_i\big) \big]$$

$$\text{s.t. } z_i^*(x) = \arg\min_{z \in \mathbb{R}^{d_z}} g_i(x, z) = \mathbb{E}_\zeta \big[ g_i(x, z; \zeta_i) \big] \tag{1}$$

where the upper- and lower-level function $f_i$ and $g_i$ for block $i$ take the expectation form w.r.t. the random variables $\xi, \zeta$, and are jointly continuously differentiable, $\mathbf{z}^* = \big( z_1^*(x), ..., z_n^*(x) \big) \in \mathbb{R}^{d_z \times n}$ contains all lower-level optimal solutions, and $n$ is the number of blocks. The above problem has various applications in machine learning, including deep AUC maximization [24, 23], meta-learning [16, 3], hyperparameter optimization [17], and robust learning [17]. This paper focuses on the setting with a nonconvex-strongly-concave minimax upper-level problem and a strongly-convex lower-level problem.

To date, barring a few works on optimizing special cases of this problem [17, 24], the solution algorithm to its general form has not been well studied. The primary obstacle lies in the significant computational cost per iteration, arising from the inherent structure of multi-block minimax bilevel optimization. To address this challenge, [17] considered a special case where $y$ is the simplex variable, and introduced a single-loop gradient descent-ascent algorithm, based on the two-timescale bilevel

---

[*]These authors contributed equally to this work.

38th Conference on Neural Information Processing Systems (NeurIPS 2024).

framework in [22]. [24] proposed a single-loop matrix-vector-based algorithm for a special case of our problem, where each upper-level function $f_i$ is evaluated only at the $i_{th}$ coordinate of $y$. However, these methods require computing the expensive second-order derivatives (i.e., the Hessian matrix or Hessian-vector product) per iteration, and the more efficient first-order approaches have not been explored yet. In this paper, we propose two efficient first-order minimax bilevel algorithms and further apply them to two novel ML applications. Our contributions are summarized as follows.

- By converting the original minimax bilevel problem into a simple minimax problem, we first propose a fully first-order single-loop algorithm named FOSL, which is easy to implement by updating $x, y$ and $\mathbf{z}$ simultaneously, and is computationally efficient without the calculation of any second-order Hessian or Jacobian matrices. We provide a convergence analysis for FOSL under a practical block sampling without replacement setting and show that its sample complexity matches the best-known result of the same type of methods in standard bilevel optimization. Technically, we characterize the gap between the reformulated and original problems and need to deal with the interplay among four variables in the error analysis.

- In the settings where the number of blocks is substantial (e.g., in few-shot meta-learning), it becomes impractical to store all block-specific parameters to perform the single-loop optimization. To this end, we also propose a memory-efficient method named MemCS via a cold-start initialization, which randomly initializes a new weight for each sampled block, without saving it for the next iteration. We then analyze the convergence of MemCS under the partial-block and full-block participation, and show that it can achieve a better sample complexity than the same type of methods in standard bilevel optimization.

- We further apply our approaches to two ML applications: deep AUC maximization and robust meta-learning. The first application pertains to the established field of AUC Maximization, while the second explores a novel application known as Rank-based Robust Meta-Learning. We show the effectiveness of our methods over a variety of datasets including CIFAR100, CelebA, CheXpert, OGBG-MolPCBA, Mini-ImageNet and Tiered-ImageNet.

## 2   Related Work

**(Minimax) bilevel optimization.**   Bilevel optimization, introduced in [2], has been extensively studied, with constraint-based methods [13, 20, 50, 51] and gradient-based methods [1, 45, 14, 49, 57] emerging as two predominant types of approaches. The constraint-based methods (e.g., [34, 38, 30, 35, 54]) reformulated the lower-level problem as a value-function-based constraint, and solved it via different constrained optimization algorithms. More recently, [16, 23] studied the minimax bilevel optimization problem and proposed single-loop algorithms with applications to robust machine learning and deep AUC maximization. In this paper, we propose two efficient, fully first-order algorithms with solid performance guarantees. In recent years, there has been a growing interest in gradient-based methods due to their efficiency in solving machine-learning problems. Within this category, Iterative Differentiation (ITD) based methods [6, 7, 14, 39, 49, 27] and Approximate Implicit Differentiation (AID) based methods [1, 5, 33, 45, 41, 14, 27, 22] are two important classes distinguished by their approaches to approximating hypergradients.

**Deep AUC maximization (DAM).**   DAM methods are aimed to mitigate the impact of imbalanced data in binary classification by directly maximizing the *area under the ROC curve* (AUC), a performance metric less affected by imbalanced data. As the AUC is difficult to optimize directly, research on DAM primarily focuses on devising effective optimization methods for its continuous surrogates [21, 4, 46, 8]. [37] proposed to reformulate the deep AUC maximization problem as a minimax optimization problem, providing the foundation for stochastic DAM algorithms developed in recent years [59, 60, 18, 24]. Among them, the most relevant work [24] formulated the DAM problem as a multi-block minimax optimization problem. In this work, we will use this form of DAM to demonstrate the effectiveness of our algorithm. A more comprehensive overview of DAM methods can be found in the survey [56].

**Robust meta-learning.**   Meta-learning provides effective solutions to multi-task learning in few-shot learning settings. In meta-learning, one trains a meta-model that can be quickly turned into a model that adapts to new tasks with only a few updates. Meta-learning algorithms in real-world applications must be robust to handle corrupted or low-quality data such as outliers. The majority of robust meta-learning methods encompass filtering [55], re-weighting [48, 28, 31, 36], and re-labeling[43, 52, 61]

on the sample level. Moreover, some other works focus on improving task-level robustness [28, 58]. In this work, we show that robust meta-learning can be formulated as a minimax bilevel optimization problem and solved with the proposed algorithm.

## 3 Algorithms

### 3.1 Reformulation as a Minimax Problem

Motivated by [34, 38, 30] in single-machine bilevel optimization, we reformulate the lower-level problem as a value-function-based constraint and aim to solve the following equivalent problem:

$$\min_x \max_y \frac{1}{n} \sum_{i=1}^{n} f_i(x, y, z_i) \quad \text{s.t.} \ g_i(x, z_i) - g_i(x, z_i^*) \le 0, \tag{2}$$

where $z_i^* := \arg\min_z g_i(x, z)$. Inspired by [30], we form a Lagrangian $\mathcal{L}_i$ with Lagrangian multiplier $\lambda \ge 0$ to approximate the original problem for each block $i$ in eq. (2), as

$$\mathcal{L}_i(x, y, z_i, v_i) = f_i(x, y, z_i) + \lambda\big(g_i(x, z_i) - g_i(x, v_i)\big),$$

where $v_i$ is used to approximate the lower-level solution $z_i^*(x)$ of the $i_{th}$ block. Then, we turn to solve the following surrogate minimax problem:

$$\min_{x, \mathbf{z}} \max_{y, \mathbf{v}} \mathcal{L}(x, y, \mathbf{z}, \mathbf{v}) := \frac{1}{n} \sum_{i=1}^{n} \mathcal{L}_i(x, y, \mathbf{z}\boldsymbol{e}_i, \mathbf{v}\boldsymbol{e}_i), \tag{3}$$

where $\mathbf{z} = (z_1, ..., z_n) \in \mathbb{R}^{d_z \times n}$, $\mathbf{v} = (v_1, ..., v_n) \in \mathbb{R}^{d_z \times n}$ and the standard basis vector $\boldsymbol{e}_i$ has only one non-zero element of 1 at the $i_{th}$ coordinate. We show later in Section 4.2 that the gap between the gradients $\nabla F(x, y^*(x), \mathbf{z}^*(x))$ and $\nabla \mathcal{L}(x, y^*(x), \mathbf{z}_\lambda^*(x), \mathbf{z}^*(x))$ of the original and surrogate problems can be effectively bounded by $\mathcal{O}(1/\lambda)$, where $y^*(x)$ denotes the maximize of outer-objective $F(x, \cdot, \mathbf{z}^*(x))$ and each vector $z_{\lambda,i}^*(x)$ in $\mathbf{z}_\lambda^*(x) := (z_{\lambda,1}^*(x), ..., z_{\lambda,n}^*(x))$ denotes the minimizer of the Lagrangian function $\mathcal{L}_i(x, y^*(x), \cdot, v)$ (where $z_{\lambda,i}^*(x)$ has not reliance on $v$). This validates the effectiveness of the Lagrangian approximation for $\lambda$ sufficiently large. Next, we propose two efficient algorithms, namely FOSL and MemCS, to solve the surrogate problem in eq. (3).

### 3.2 FOSL: Fully First-Order Single-Loop Method

As shown in Algorithm 1, we first sample a subset $I_t \subset \mathcal{S} := \{1, ..., n\}$ of blocks without replacement. Noting that $z_i$ and $v_i$ are both block-specific variables, we then apply a stochastic ascent and descent step to update $v_i$ and $z_i$ for all block $i \in I_t$ as

$$v_{i,t+1} = v_{i,t} + \eta_v\big(-\nabla_z g_i(x_t, v_{i,t}; \xi_{v,i}^t)\big)$$

$$z_{i,t+1} = z_{i,t} - \eta_z \nabla_z \mathcal{L}_i\big(x_t, y_t, z_{i,t}, v_{i,t}; \xi_{z,i}^t\big),$$

where the gradient of $\mathcal{L}_i$ w.r.t. $z$ has no dependence on $v$. Since the solutions w.r.t. variables $x$ and $y$ depend on all blocks, we use the average of stochastic gradient estimators from the selected blocks in $I_t$ to update $y$ and $x$ as

$$y_{t+1} = y_t + \eta_y \frac{1}{|I_t|} \sum_{i \in I_t} \nabla_y f_i(x_t, y_t, v_{i,t}; \xi_{y,i}^t)$$

$$x_{t+1} = x_t - \eta_x \frac{1}{|I_t|} \sum_{i \in I_t} \nabla_x \mathcal{L}_i(x_t, y_t, z_{i,t}, v_{i,t}; \xi_{x,i}^t).$$

Note that our algorithm takes a simpler fully single-loop structure via updating $\{v_{i,t}, z_{i,t}\}_{i \in I_t}, x_t$ and $y_t$ simultaneously at each iteration. Hence, it can also benefit from the hardware parallelism. In addition, different from the methods in [17, 24] that need to compute the higher order Hessian- or Jacobian-vector products, our method only uses the first-order gradients.

### 3.3 MemCS: Memory-Efficient Cold-Start Method

Note that in the single-loop optimization of Algorithm 1, all block-specific parameters $v_{i,t}$ and $z_{i,t}$ of blocks in $I_t$ need to be stored for the updates at iteration $t + 1$. However, in some ML applications,

---

**Algorithm 1** Fully First-Order Single-Loop Method (FOSL)

---

1: **Input:** initialization $\{x_0, y_0, z_0, v_0\}$, number of iteration rounds $T$, learning rates $\{\eta_x, \eta_y, \eta_z, \eta_v\}$
2: **for** $t = 0, 1, 2, ..., T$ **do**
3:      Sample blocks $I_t \subset S$
4:      **for** $i \in I_t$ **do**
5:          $v_{i,t+1} = v_{i,t} - \eta_v \nabla_z g_i(x_t, v_{i,t}; \xi_{v,i}^t)$
6:          $z_{i,t+1} = z_{i,t} - \eta_z \nabla_z \mathcal{L}_i(x_t, y_t, z_{i,t}, v_{i,t}; \xi_{z,i}^t)$
7:      **end for**
8:      $y_{t+1} = y_t + \eta_y \frac{1}{|I_t|} \sum_{i \in I_t} \nabla_y f_i(x_t, y_t, v_{i,t}; \xi_{y,i}^t)$
9:      $x_{t+1} = x_t - \eta_x \frac{1}{|I_t|} \sum_{i \in I_t} \nabla_x \mathcal{L}_i(x_t, y_t, z_{i,t}, v_{i,t}; \xi_{x,i}^t)$
10: **end for**

---

---

**Algorithm 2** Memory-Efficient Cold-Start (MemCS)

---

1: **Input:** initialization $\{x_0, y_0\}$, number of iteration rounds $T$, learning rates $\{\eta_x, \eta_y, \eta_z, \eta_v\}$
2: **for** $t = 0, 1, 2, ..., T - 1$ **do**
3:      Sample blocks $I_t \subset S$
4:      **for** $i \in I_t$ **do**
5:          Random initializations $v_{i,t}^0, z_{i,t}^0$
6:          **for** $k = 0, 1, 2, ..., K - 1$ **do**
7:              $v_{i,t}^{k+1} = v_{i,t}^k - \eta_v \nabla_z g_i(x_t, v_{i,t}^k)$
8:              $z_{i,t}^{k+1} = z_{i,t}^k - \eta_z \nabla_z \mathcal{L}_i(x_t, y_t, z_{i,t}^k, v_{i,t}^k)$
9:          **end for**
10:      **end for**
11:      $y_{t+1} = y_t + \eta_y \frac{1}{|I_t|} \sum_{i \in I_t} \nabla_y f_i(x_t, y_t, v_{i,t}^K)$
12:      $x_{t+1} = x_t - \eta_x \frac{1}{|I_t|} \sum_{i \in I_t} \nabla_x \mathcal{L}_i(x_t, y_t, z_{i,t}^K, v_{i,t}^K)$
13: **end for**

---

such as few-shot meta-learning, the number of blocks/tasks is often large, and hence Algorithm 1 can suffer from significant memory consumption. To address this challenge, we propose a memory-efficient method named MemCS in Algorithm 2. Differently from the single-loop updates in FOSL, MemCS contains a sub-loop of $K$ steps of gradient descent[1] in updating the block-specific variables $v_{i,t}^k$ and $z_{i,t}^k$ for $k = 0, ..., K - 1$ with **random initialization** $v_{i,t}^0$ and $z_{i,t}^0$. After obtaining the outputs $v_{i,t}^K, z_{i,t}^K$ of this sub-loop, the remaining step is to update $y_t$ and $x_t$ via gradient ascent and descent similarly as in FOSL.

## 4 Main Results

### 4.1 Assumptions

**Definition 4.1.** A mapping $f$ is $L$-Lipschitz continuous if $\|f(x_1) - f(x_2)\| \leq L\|x_1 - x_2\|$, for any $x_1, x_2$. We say $f$ is $L$-smooth if $\nabla f$ is $L$-Lipschitz continuous.

Since the overall objective is nonconvex w.r.t. $x$, we aim to find an $\epsilon$-accurate stationary point.

**Definition 4.2.** We call $\bar{x}$ as an $\epsilon$-accurate stationary point of the objective function $\Phi(x)$ if $\mathbb{E}\|\nabla \Phi(\bar{x})\|^2 \leq \epsilon$, where $\epsilon \in (0, 1]$ and $\bar{x}$ is the output of an algorithm.

We use the following assumptions in the subsequent description. Note that these assumptions are widely adopted in existing studies [24, 17].

**Assumption 4.3.** For any $x \in \mathbb{R}^{d_x}$, $y \in \mathbb{R}^{d_y}$, $z \in \mathbb{R}^{d_z}$ and $i \in \{1, 2, ..., n\}$, $f_i(x, y)$ and $g_i(x, y)$ are twice continuously differentiable, $f_i(x, y, z)$ is $\mu_f$-strongly concave w.r.t. $y$ and $g_i(x, z)$ is $\mu_g$-strongly convex w.r.t. $z$.

The following assumption imposes the *Lipschitz continuity* on the upper- and lower-level functions and their derivatives.

---

[1]For MemCS, we focus on the few-shot setting such as meta-learning, where each block contains a small number of samples, and hence we use gradient descent here. However, the algorithm can also be extended to the stochastic setting.

**Assumption 4.4.** For any $x \in \mathbb{R}^{d_x}$, $z \in \mathbb{R}^{d_z}$ and $i \in \{1, 2, ..., n\}$, $f_i(x, y, z)$ is $L_{f,0}$-Lipschitz continuous w.r.t. $x$, $g_i(x, z)$ is $L_{g,0}$-Lipschitz continuous w.r.t. $x$, $f_i(x, y, z)$ is $L_{f,1}$-smooth and $g_i(x, y)$ is $L_{g,1}$-smooth. In addition, the second-order derivatives $\nabla^2 f_i(x, y, z)$ and $\nabla^2 g_i(x, y)$ are $L_{f,2}$- and $L_{g,2}$-Lipschitz continuous.

Next, we make a *bounded variance* assumption for the gradients in the stochastic setting.

**Assumption 4.5.** There exist constants $\sigma_f$ and $\sigma_g$ such that the variances $\mathbb{E}\|\nabla f_i(x, y, z) - \nabla f_i(x, y, z; \xi)\|^2 \le \sigma_f^2$ and $\mathbb{E}\|\nabla g_i(x, z) - \nabla g_i(x, z; \zeta)\|^2 \le \sigma_g^2$.

The following assumption on *block heterogeneity* measures the similarities of the upper-level gradients $\nabla_y f_i(x, z)$ for all $i$. This has not been discussed in previous works, but it is necessary for our approach as we explore a more general outer-maximization solution $y^*(x)$ for $F$, rather than for single $f_i$.

**Assumption 4.6.** For any $x \in \mathbb{R}^{d_x}$, $z \in \mathbb{R}^{d_z}$, there exist constants $\beta_{th} \ge 1$ and $\sigma_{th} \ge 0$ such that

$$\frac{1}{n}\sum_{i=1}^{n}\mathbb{E}\|\nabla_y f_i(x, y, z)\|^2 \le \beta_{th}^2 \mathbb{E}\|\nabla_y F(x, y, z)\|^2 + \sigma_{th}^2.$$

We have $\beta_{th} = 1$ and $\sigma_{th} = 0$ when all $g_i$'s are identical.

## 4.2 Convergence analysis

For simplicity, we fix the number of involved blocks $|I_t| = P$ for all $t$. Let $y^*(x)$ be the maximizer of $F$ function w.r.t. $y$. Then, the overall objective of the original problem in eq. (1) w.r.t. $x$ is given by

$$\Phi(x) := F\big(x, y^*(x), \mathbf{z}^*(x)\big),$$

where $\mathbf{z}^*(x)$ is the lower-level minimizer and $y^*(x)$ is the maximizer of $F(x, \cdot, \mathbf{z}^*(x))$. In addition, recall that the objective function of the surrogate problem in eq. (3) w.r.t. $x$ is given by $\mathcal{L}^*(x) := \mathcal{L}\big(x, y^*(x), \mathbf{z}_\lambda^*(x), \mathbf{z}^*(x)\big)$. We next characterize the difference between the gradients of the original and the surrogate problems.

**Proposition 4.7.** *Under Assumptions 4.3, 4.4, the gap between $\nabla\Phi(x)$ and $\nabla\mathcal{L}^*(x)$ can be bounded as*

$$\big\|\nabla\Phi(x) - \nabla\mathcal{L}^*(x)\big\| = \mathcal{O}(1/\lambda).$$

Due to the limit of space, the proof of Proposition 4.7 can be found in Lemma D.5 in the appendix. For a properly large $\lambda$, Proposition 4.7 guarantees that the stationary points of the original and surrogate problems are close to each other. However, too large $\lambda$ can explode the gradient estimation variance, resulting in a much slower convergence rate. This trade-off makes the selection of $\lambda$ important, as shown in our theorems later.

We next give an upper bound on the gradient norm $\mathbb{E}\|\nabla\mathcal{L}^*(x_t)\|^2$ of the surrogate problem. Denote $h_x^t := \nabla_x \mathcal{L}_i(x_t, y_t, z^t, v^t; \xi_{x,i}^t)$ and its expectation as $\widetilde{h}_x^t$.

**Proposition 4.8.** *Under Assumptions 4.3, 4.4, 4.5, the consecutive iterates of Algorithm 1 satisfy:*

$$\mathbb{E}\|\nabla\mathcal{L}^*(x_t)\|^2 \le \frac{2}{\eta_x}\mathbb{E}\big[\mathcal{L}^*(x_{t+1}) - \mathcal{L}^*(x_t)\big] - \mathbb{E}\|\widetilde{h}_x^t\|^2 + \eta_x L_{*,1}\mathbb{E}\|h_x^t\|^2 + 3L_{f,1}^2\mathbb{E}\|y_t - y^*(x_t)\|^2$$

$$+ \frac{3L_{\lambda,1}^2}{n}\sum_{i=1}^{n}\mathbb{E}\Big[\big\|z_{i,t} - z_{\lambda,i}^*(x_t)\big\|^2 + \big\|v_{i,t} - z_i^*(x_t)\big\|^2\Big]$$

*for all $t \in \{0, 1, ..., T-1\}$, where $L_{\lambda,1}$ and $L_{*,1}$ are given in Lemma D.1 and Lemma D.6 in the appendix respectively.*

The proof of Proposition 4.8 can be found in Lemma E.1 in the appendix. The same result can be obtained for Algorithm 2 by replacing $v_{i,t}$ and $z_{i,t}$ with $v_{i,t}^K$ and $z_{i,t}^K$. This proposition shows that the convergence rate of our algorithm relies on how fast the iterates $y_t, z_{i,t}$ and $v_{i,t}$ converge to their optimal solutions at each iteration $t$. We next characterize the distance of $y_t$ to its optimal solution.

**Proposition 4.9.** *Under Assumptions 4.3, 4.4, 4.5, the iterates of $y_t$ generated according to Algorithm 1 satisfy*

$$\mathbb{E}\|y_{t+1} - y^*(x_{t+1})\|^2 - \mathbb{E}\|y_t - y^*(x_t)\|^2 \leq -\mathcal{O}(\eta_y) \cdot \mathbb{E}\|y_t - y^*(x_t)\|^2 + \mathcal{O}\left(\frac{\eta_y^2}{|I_t|}\right) \cdot (\sigma_f^2 + \sigma_{th}^2)$$

$$+ \mathcal{O}(\eta_y) \cdot \frac{1}{n}\sum_{i=1}^{n} \mathbb{E}\|v_{i,t} - z_i^*(x_t)\|^2 + \mathcal{O}\left(\frac{\eta_x^2}{\eta_y}\right)\mathbb{E}\|\widetilde{h}_x^t\|^2 + \mathcal{O}(\eta_x^2)\mathbb{E}\|h_x^t\|^2,$$

*for all $t \in \{0, ..., T-1\}$.*

The proof of Proposition 4.9 refers to Lemma E.3 in the appendix. It can be seen that with properly small stepsizes $\eta_x$ and $\eta_y$, there exists a descent of the optimal distance $\mathbb{E}\|y_t - y^*(x_t)\|^2$, which is critical for the final convergence analysis. Similar results are obtained for $v_{i,t}$ and $z_{i,t}$. Combining the above propositions and the auxiliary lemmas in the appendix, we get the following result for Algorithm 1.

**Theorem 4.10** (Convergence of FOSL). *Suppose Assumptions 4.3, 4.4, 4.5, 4.6 are satisfied. Set parameters $\eta_x = \mathcal{O}(T^{-\frac{5}{7}})$, $\eta_y = \mathcal{O}(T^{-\frac{4}{7}})$, $\eta_z = \mathcal{O}(T^{-\frac{5}{7}})$, $\eta_v = \mathcal{O}(T^{-\frac{4}{7}})$ and $\lambda = \mathcal{O}(T^{\frac{1}{7}})$. Then, we have*

$$\frac{1}{T}\sum_{t=0}^{T-1}\mathbb{E}\|\nabla\Phi(x_t)\|^2 \leq \frac{2C_{gap}}{\lambda^2} + \frac{4(\Psi_0 - \Psi_T)}{T\eta_x} + \frac{4\eta_x\lambda^2}{P}\left(1 + \frac{\eta_x}{\eta_y} + \frac{\eta_x\lambda^2}{(\eta_z\lambda)} + \frac{\eta_x\lambda^2}{\eta_v}\right)C_2$$

$$+ 4(\eta_y + (\eta_z\lambda)\lambda^2 + \eta_v\lambda^2)C_3$$

$$\leq \mathcal{O}(T^{-\frac{2}{7}}),$$

*where $C_{gap}$ is defined in Lemma D.5, $C_2, C_3$ are defined in eq. (39) in the appendix, and $\Psi_t := \mathcal{L}^*(x_t) + K_y\mathbb{E}\|y_t - y^*(x_t)\|^2 + K_z\frac{1}{n}\sum_{i=1}^{n}\mathbb{E}\|z_{i,t} - z^*_{\lambda,i}(x_t)\|^2 + K_v\frac{1}{n}\sum_{i=1}^{n}\mathbb{E}\|v_{i,t} - z_i^*(x_t)\|^2$.*

Next, we characterize the sample complexity for FOSL.

**Corollary 4.11.** *Under the same setting of Theorem 4.10, our algorithm finds an $\epsilon$-accurate stationary solution after $T = \mathcal{O}(\epsilon^{-\frac{7}{2}})$ interactions. The total sample complexity for all involved blocks is $PT = \mathcal{O}(P\epsilon^{-\frac{7}{2}})$.*

Compared with existing works [17, 24], our algorithm is free from second-order derivative computations. In addition, the sample complexity of our algorithm matches the best result [30] of the same type of methods in standard single-block bilevel optimization.

Next, we analyze the convergence for Algorithm 2 under the partial- and full-block participation.

**Theorem 4.12** (Convergence of MemCS). *Suppose Assumptions 4.3, 4.4, 4.5, 4.6 are satisfied. Assume there exists some $B > 0$ such that $\|z_i^*(x_t)\| \leq B$ for any $x_t$, $i = 1, ..., N$. For the partial-block participation, by setting parameters $\eta_x = \mathcal{O}(P^{\frac{1}{5}}T^{-\frac{2}{3}})$, $\eta_y = \mathcal{O}(P^{-\frac{1}{5}}T^{-\frac{1}{2}})$, $\eta_z = \mathcal{O}(P^{-\frac{1}{10}}T^{-\frac{1}{6}})$, $\eta_v = \mathcal{O}(1)$, $\lambda = \mathcal{O}(P^{\frac{1}{10}}T^{\frac{1}{6}})$ and taking $\epsilon_{sub} = \mathcal{O}(P^{-\frac{2}{5}}T^{-\frac{2}{3}})$, we have*

$$\frac{1}{T}\sum_{t=0}^{T-1}\mathbb{E}\|\nabla\Phi(x_t)\|^2 \leq \frac{2C_{gap}}{\lambda^2} + \frac{2(\Psi_0 - \Psi_T)}{T\eta_x} + \frac{4\eta_x\lambda^2}{P}\left(1 + \frac{\eta_x}{\eta_y}\right)C_2 + \frac{\eta_y}{P}\frac{24L_{f,1}^2\sigma_{th}^2}{\mu_f}$$

$$+ 4\left(3L_{\lambda,1}^2 + \frac{12L_{f,1}^4}{\mu_f^2}\right)\epsilon_{sub} \leq \mathcal{O}(P^{-\frac{1}{5}}T^{-\frac{1}{3}}),$$

*where $L_{\lambda,1} := 3\lambda L_{g,1}$, $C_{gap}$ is defined by Lemma D.5 in the appendix and $\Psi_t := \mathcal{L}^*(x_t) + K_y\mathbb{E}\|y_t - y^*(x_t)\|^2$. For the full-block participation, by setting $\eta_x = \mathcal{O}(1)$, $\eta_y = \mathcal{O}(1)$, $\eta_z = \mathcal{O}(T^{-\frac{1}{2}})$, $\eta_v = \mathcal{O}(1)$, $\lambda = \mathcal{O}(T^{\frac{1}{2}})$ and taking $\epsilon_{sub} = \mathcal{O}(T^{-2})$, we have*

$$\frac{1}{T}\sum_{t=0}^{T-1}\mathbb{E}\|\nabla\Phi(x_t)\|^2 \leq \frac{2C_{gap}}{\lambda^2} + \frac{4(\Psi_0 - \Psi_T)}{T\eta_x} + 12\left(L_{\lambda,1}^2 + \frac{4L_{f,1}^4}{\mu_f^2}\right)\epsilon_{sub} \leq \mathcal{O}(T^{-1}).$$

Note that the assumption $\|z_i^*(x_t)\| \leq B$ can be removed when the domain of $x$ is a closed convex set and projected gradient descent is used to update $x$ [12, 15]. We next characterize the sample complexity for MemCS.

**Corollary 4.13.** *Under the same setting of Theorem 4.12,*

- *For partial-block participation, our algorithm finds an $\epsilon$-accurate stationary solution of $\Phi(x)$ after $T = \mathcal{O}(P^{-\frac{3}{5}}\epsilon^{-3})$ outer iterations and $K = \mathcal{O}(\log\frac{1}{\epsilon})$ inner iterations. The total sample complexity for all involved blocks is $PKT = \widetilde{\mathcal{O}}(P^{\frac{2}{5}}\epsilon^{-3})$.*

- *For full-block participation, our algorithm finds an $\epsilon$-accurate stationary solution of $\Phi(x)$ after $T = \mathcal{O}(\epsilon^{-1})$ outer iterations and $K = \mathcal{O}(\log\frac{1}{\epsilon})$ inner iterations. The total sample complexity for all involved blocks is $nKT = \widetilde{\mathcal{O}}(n\epsilon^{-1})$.*

Note that the per-block sample complexity of our MemCS algorithm takes an order of $\epsilon^{-1}$, which improves that of the same-type F$^2$SA [30] by an order of $\epsilon^{-0.5}$, based on a refined analysis on the smoothness of the overall objective function. Corollary 4.13 also shows that MemCS achieves a linear convergence speedup w.r.t. the number $P$ of blocks. As far as we know, this is the first linear speedup result in multi-block minimax bilevel optimization.

## 5 Applications and Experiments

In this section, we conduct extensive experiments in two applications: deep AUC maximization and rank-based robust meta-learning. More experimental results such as time and space comparison are provided in Appendix B.

### 5.1 Deep AUC Maximization

#### 5.1.1 Formulation

Deep AUC Maximization (DAM) addresses machine learning challenges presented by imbalanced datasets. In particular, the AUC (Area Under the ROC Curve) measures the likelihood that a positive sample's prediction score will be higher than that of a negative sample. As outlined by [24], the DAM issue is structured as a multi-block minimax bilevel optimization problem:

$$\min_{\mathbf{w},a,b}\max_{\alpha}\sum_{j=1}^{m}\Phi_j\big(\mathbf{u}_j^*(\mathbf{w}_j),a_j,b_j,\alpha_j\big)\quad s.t.\ \mathbf{u}_j^*(\mathbf{w}_j)=\arg\min_{\mathbf{u}_j}g_j(\mathbf{u}_j,\mathbf{w}_j),$$

where $g_j(\mathbf{u}_j,\mathbf{w}_j):=\frac{1}{2}\big\|\mathbf{u}_j-\big(\mathbf{w}_j-\tilde{\eta}\nabla L_{AVG}(\mathbf{w}_j)\big)\big\|^2$, $L_{AVG}(\mathbf{w}_j):=\frac{1}{n}\sum_{i=1}^{n}\ell(\mathbf{w}_j;x_i,y_i)$, $\ell$ denotes the task loss (e.g., the cross-entropy loss in binary classification tasks), and $\Phi_j$ denotes the sample-level AUC loss function. The detailed formulation can be found in Appendix A.1. With the method in Section 3, we reformulate this problem as a single-level minimax optimization problem:

$$\min_{\mathbf{w},\mathbf{u},a,b}\max_{\alpha,\mathbf{v}}\mathcal{L}(\mathbf{w},\mathbf{u},a,b,\alpha,\mathbf{v}),$$

where $\mathcal{L}(\mathbf{w},\mathbf{u},a,b,\alpha,\mathbf{v}):=\sum_{j=1}^{m}\Phi_j(\mathbf{u}_j,a_j,b_j,\alpha_j)+\lambda\big(g_j(\mathbf{u}_j,\mathbf{w}_j)-g_j(\mathbf{v}_j,\mathbf{w}_j)\big)$ is the Lagrangian function of AUC loss function, and $\mathbf{v}_j$ is the approximate optimal solution of $g_j$.

#### 5.1.2 Results

**Settings.** Following the work [24], we assess our methodology using four datasets, namely *CIFAR100* [29], *CelebA* [40], *CheXpert* [26] and *OGBG-MolPCBA* [25], whose details are provided in Appendix B.1. We evaluate the effectiveness of our algorithm by comparing it with direct optimization on multi-block minimax AUC loss (mAUC) and compositional training on mAUC loss (mAUC-CT). The test AUC scores of mAUC and mAUC-CT for different datasets in Table 1 are derived from the original paper. Detailed configuration of experiments can be found in Appendix B.2.

**Results.** The results of deep AUC maximization on different datasets are shown in Table 1. The results indicate that our FOSL outperforms the mAUC method in terms of test AUC scores on all datasets and achieves comparative or better performance than mAUC-CT on various datasets. We proceed to visualize the AUC loss during the initial stages of training for all methods on CelebA, as depicted in Figure 1a and 1b. The figures illustrate that, in the initial stage, our method and mAUC-CT [24] exhibit a faster loss drop than mAUC, and our method achieves the fastest overall convergence rate. Furthermore, our approach exhibits a smaller fluctuation compared to other baseline methods.

Table 1: Test AUC score with 95% confidence interval on different datasets for AUC maximization.

| | CIFAR100 | CelebA | CheXpert | OGBG-MolPCBA |
|---|---|---|---|---|
| mAUC[24] | 0.9044 (0.0015) | 0.9062 (0.0042) | 0.8084 (0.1455) | 0.7793 (0.0028) |
| mAUC-CT[24] | 0.9272 (0.0014) | 0.9192 (0.0004) | **0.8198 (0.1495)** | 0.8406 (0.0044) |
| FOSL(ours) | **0.9540 (0.0009)** | **0.9267 (0.0018)** | 0.8166 (0.0051) | **0.8516 (0.0014)** |

(a)   (b)   (c)   (d)

Figure 1: Visualization results of FOSL experiments. (a) Training AUC loss over **iteration rounds** during the initial stages of training. (b) Training AUC loss over **time** during the initial training phase. (c) Impact of $\lambda$ on test AUC score throughout training on the CIFAR100 dataset. (d) Impact of $\lambda$ on test AUC score throughout training on the CelebA dataset.

**Impact of $\lambda$.** To evaluate the impact of the hyper-parameter $\lambda$ on training with FOSL algorithm, we conduct an ablation study on the CIFAR100 and the CelebA datasets. The test AUC scores along with training are depicted in Figure 1c and 1d. As shown in Figure 1c, our method sustains robust performance across a wide range of choices for $\lambda$. For example, training within a $\lambda$ range of [2, 8] shows that the speed of convergence and the final test performance are not sensitive to the change of $\lambda$. A similar observation also holds for the CelebA dataset as shown in Figure 1d.

## 5.2 Robust Meta-learning with Rank-based Loss

### 5.2.1 Formulation

Our objective is to devise a robust meta-learning approach wherein, during each iteration, tasks with large loss values are filtered out, and the meta-model is updated with the remaining tasks. This approach effectively reduces the impact of tasks with noisy samples (noisy tasks), because deep learning models tend to acquire clean and simple patterns in their initial training stages [19], such that noisy samples/tasks often have large loss values. Further justification can be found in Figure 2.

We first define $g_{[i]}$ as the $i_{th}$ largest element of the set $\mathcal{G} = \{g_1, g_2, ..., g_n\}$, such that $g_{[n]} \leq g_{[n-1]} \leq ... \leq g_{[1]}$. Denote the task-specific loss as $g_i(\phi, w_i)$, where $\phi$ is the parameter of the meta-model and $w_i$ is the task-specific parameter. The proposed formulation is then given by:

$$\min_{\phi} F(\phi, \mathbf{w}^*) := \frac{1}{k} \sum_{i=n-k+1}^{n} g_{[i]}\big(\phi, w_{[i]}^*(\phi)\big) \quad s.t. \ w_i^*(\phi) = \arg\min_{w_i} g_i(\phi, w_i),$$

where $g_{[i]}\big(\phi, w_{[i]}^*(\phi)\big)$ is the $i_{th}$ largest task-specific loss given $\mathbf{w}^* := \big[w_1^*(\phi), ..., w_n^*(\phi)\big]^T$, and $w_{[i]}^*(\phi)$ is the corresponding optimal task-specific parameter.

With the Lemma 1 in [42], by introducing an auxiliary variable $\gamma$, we can reformulate the problem as:

$$\min_{\phi} \max_{\gamma} F(\phi, \mathbf{w}^*, \gamma) = \frac{1}{k} \sum_{i=1}^{n} f_i\big(\phi, w_i^*(\phi), \gamma\big) = \frac{1}{k} \sum_{i=1}^{n} \left\{ g_i^*(\phi) - [g_i^*(\phi) - \gamma]_+ - \frac{n-k}{n}\gamma \right\}$$
$$s.t. \ w_i^*(\phi) = \arg\min_{w_i} g_i(\phi, w_i).$$

Details about the derivation of the above formulation can be found in Appendix A.2. This formulation poses a non-convex optimization challenge for the primal variable $\phi$, making it challenging to address using conventional optimization methods. Nevertheless, our proposed algorithm enables efficient resolution of this problem by reformulating the problem into a single-level minimax optimization problem as: $\min_{\phi, \mathbf{w}} \max_{\gamma, \mathbf{v}} \mathcal{L}(\phi, \mathbf{w}, \gamma, \mathbf{v})$, where $\mathcal{L}(\phi, \mathbf{w}, \gamma, \mathbf{v}) := \frac{1}{n} \sum_{i=1}^{n} f_i(\phi, w_i, \gamma) + \lambda\big(g_i(\phi, w_i) - g_i(\phi, v_i)\big)$ is the Lagrangian function of the rank based loss function, $\mathbf{v}$ is an approximate optimal task-specific parameter of the lower-level problem.

Table 2: Test accuracy (%) on the Mini-ImageNet and the Tiered-ImageNet datasets for meta-learning.

| Dataset | Method | Clean | Flip | Rand |
|---|---|---|---|---|
| Mini | MAML | 64.75 | 52.75 | 52.50 |
| | MemCS(ours) | **69.25** | **57.50** | **60.25** |
| Tiered | MAML | 66.25 | 44.75 | 54.25 |
| | MemCS(ours) | **67.25** | **57.00** | **59.75** |

Table 3: Test accuracy (%) on Mini-ImageNet and Tiered-ImageNet with different noisy ratio for *Flip* setting.

| Dataset | Method | Noisy ratio | | | | |
|---|---|---|---|---|---|---|
| | | 0 | 0.2 | 0.4 | 0.6 | 0.8 |
| Mini | MAML | 64.75 | 59.50 | 56.50 | 52.75 | 42.00 |
| | MemCS | **69.25** | **65.00** | **61.75** | **57.50** | **53.50** |
| Tiered | MAML | 66.25 | 63.75 | 53.25 | 44.75 | 39.00 |
| | MemCS | **67.25** | **66.50** | **62.75** | **57.00** | **54.50** |

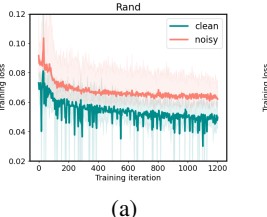 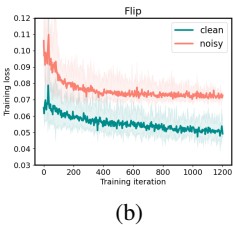 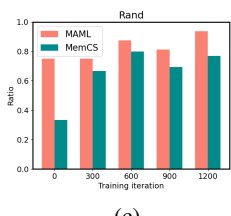 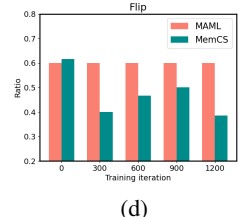

(a)        (b)        (c)        (d)

Figure 2: The portion of tasks being noisy during training for MAML and MemCS on Mini-ImageNet.

### 5.2.2 Results

**Settings.** We perform meta-learning experiments on few-shot learning tasks, focusing on the capability of rapid adaptation to new tasks with limited samples. Adhering to standard few-shot learning configurations, we carry out *5-ways 5-shot* learning experiments on Mini-ImageNet [53] (referred to as *Mini* in Table 2) and Tiered-ImageNet [47] (referred to as *Tiered* in Table 2), where each task involves a 5-class classification task, with five samples per class used as training data. Since our robust meta-learning formulation is built on that of MAML [6], we compare our method with MAML on both clean dataset and noisy dataset to evaluate the effectiveness and robustness of our algorithm. In the noisy setting, we adopt a standard noisy training scheme in meta-learning, where the labels in a noisy task are randomly flipped. Specifically, we employ two label flipping strategies: *Flip,* where in each iteration, a certain portion of tasks (60% in Table 2) are designated as noisy, and each sample within the task is assigned to one of all labels with equal probability; and *Rand,* where a random noisy ratio is assigned to each task in every iteration, determining the proportion of samples to be flipped by randomly assigning another label to them. Detailed configuration of experiments can be found in Appendix B.2.

**Results.** Table 2 displays the test accuracies. These results show that in the presence of noisy tasks, both MAML and our MemCS method undergo a decline in performance across both datasets, yet our approach manages to sustain a reasonable performance. To further evaluate the resilience of our MemCS method against MAML, we executed additional experiments with escalating noise levels in the *Flip* scenario, with these findings detailed in Table 3. The data clearly show a performance decrease for both methodologies as the noise ratio intensifies. Nonetheless, our approach exhibits a notably more gradual decline in performance as the noise ratio escalates, especially at higher noise levels, signifying superior robustness compared to MAML.

**Discussion.** To show the effectiveness of our approach in facilitating robust learning, we have visualized the average losses for both clean and noisy tasks separately within the MAML training framework, as demonstrated in Figure 2. The graphical representation uncovers a consistent pattern where the losses associated with noisy tasks consistently exceed those related to clean tasks throughout the training period. This pattern underscores our approach's capacity to lessen the detrimental effects of noisy tasks. Further, we examine the noisy tasks in the update mechanism at five distinct intervals during the training phase, illustrated in Figure 2. The findings show that our methodology successfully deters the influence of noisy tasks on the meta-model's updates across both Rand and Flip scenarios, maintaining this protective measure throughout the training duration.

## 6 Conclusion

In this paper, we propose two fully first-order algorithms designed to address the challenges posed by multi-block minimax bilevel optimization problems: a fully single-loop algorithm, FOSL, and a

memory-efficient double-loop algorithm with cold-start initialization, MemCS. We show that our methods can achieve comparative and even better per-block sample complexities than other methods with the same type in standard bilevel optimization. The experimental results indicate that our methods consistently demonstrate superior performance and robustness in applications on deep AUC maximization and robust meta-learning.

## Acknowledgement

Yifan Yang and Kaiyi Ji were partially supported by NSF grants CCF-2311274 and ECCS-2326592. Zhaofeng Si and Siwei Lyu were partially supported by an NSF research grant IIS-2008532.

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

# A  Specifications of Applications

In this section, we provide a detailed introduction to the formulation utilized in Section 5.

## A.1  Deep AUC Maximization (DAM)

For a classifier model $f(\mathbf{w})$, we have the AUC score function as

$$AUC\big(f(\mathbf{w})\big) = Pr\big(f(\mathbf{w}; x) \geq f(\mathbf{w}; x')|y = 1, y' = -1\big),$$

where $Pr(X)$ denote probability of an event $X$ being true. One of the surrogate loss (AUC square loss [9]) is given by:

$$\min_{\mathbf{w}} \frac{1}{n_+ n_-} \sum_{y_i=1} \sum_{y_j=-1} \Big(c - \big(f(\mathbf{w}; x_i) - f(\mathbf{w}; x_j)\big)\Big)^2,$$

where $n_+$ and $n_-$ are the numbers of positive examples and negative examples, respectively, and $c$ is the margin parameter. One can transfer this problem into an equivalent minimax optimization problem according to Proposition 1 in [37] by:

$$\min_{\mathbf{w},a,b} \max_{\alpha} \Phi(\mathbf{w}, a, b, \alpha) := \frac{1}{n} \sum_{i=1}^{n} \phi(\mathbf{w}, a, b, \alpha; x_i, y_i),$$

where

$$\phi(\mathbf{w}, a, b, \alpha; x_i, y_i) = (1-p)\big(f(\mathbf{w}; x_i) - a\big)^2 \cdot \mathbb{I}_{y_i=1} + p\big(f(\mathbf{w}; x_i) - b\big)^2 \cdot \mathbb{I}_{y_i=-1} - p(1-p)\alpha^2$$
$$+ 2\alpha\big(p(1-p)c + pf(\mathbf{w}; x_i) \cdot \mathbb{I}_{y_i=-1} - (1-p)f(\mathbf{w}; x_i) \cdot \mathbb{I}_{y_i=1}\big),$$

where $a, b$ are margin parameters, $p = n_+/n$. This formulation decomposed the individual examples, which is more favorable in stochastic scenarios. [59] proposed a compositional training algorithm for this problem. The compositional objective function is formulated as:

$$\min_{\mathbf{w},a,b} \max_{\alpha} \Phi(\mathbf{w} - \alpha \nabla L_{AVG}(\mathbf{w}), a, b, \alpha),$$

where $L_{AVG} = \frac{1}{n} \sum_{i=1}^{n} \ell(\mathbf{w}; x_i, y_i)$, $\ell$ denotes task loss, e.g. cross-entropy in classification tasks. Consider a multi-block problem with $m$ tasks. This problem can be reformulated as a multi-block minimax bi-level optimization problem [24]:

$$\min_{\mathbf{w},a,b} \max_{\alpha} \sum_{j=1}^{m} \Phi_j\big(\mathbf{u}_j^*(\mathbf{w}_j), a_j, b_j, \alpha_j\big) \qquad s.t. \ \mathbf{u}_j^*(\mathbf{w}_j) = \arg\min_{\mathbf{u}_j} g_j(\mathbf{u}_j, \mathbf{w}_j),$$

where $g_j(\mathbf{u}_j, \mathbf{w}_j) := \frac{1}{2}\big\|\mathbf{u}_j - \big(\mathbf{w}_j - \tilde{\eta}\nabla L_{AVG}(\mathbf{w}_j)\big)\big\|^2$.

## A.2  Robust Meta-learning

Consider the formulation of Robust Meta-learning in Section 5.2:

$$\min_{\phi} F(\phi, \mathbf{w}^*) := \frac{1}{k} \sum_{i=n-k+1}^{n} g_{[i]}\big(\phi, w_i^*(\phi)\big) \quad s.t. \ w_i^*(\phi) = \arg\min_{w_i} g_i(\phi, w_i),$$

where $g_{[n]}\big(\phi, w_n^*(\phi)\big) \leq g_{[n-1]}\big(\phi, w_{n-1}^*(\phi)\big) \leq ... \leq g_{[1]}\big(\phi, w_1^*(\phi)\big)$ denotes task-specific losses. We define $g_i^*(\phi) := g_i\big(\phi, w_i^*(\phi)\big)$ for simplicity in later formulations. The summation of the bottom-k losses is equivalent to the sum of all task losses subtracted by the sum of the top-(n-k) losses:

$$F(\phi, \mathbf{w}^*) = \frac{1}{k}\bigg(\sum_{i=1}^{n} g_i^*(\phi) - \sum_{i=1}^{n-k} g_{[i]}^*(\phi)\bigg).$$

With the Lemma 1 in [42], we have:

$$\sum_{i=1}^{n-k} g_{[i]}^*(\phi) = \min_{\gamma} \Big\{ (n-k)\gamma + \sum_{i=1}^{n} [g_i^*(\phi) - \gamma]_+ \Big\}.$$

Now we can convert the original upper-level problem to:

$$\min_{\phi} \max_{\gamma} \hat{F}(\phi, \mathbf{w}^*, \gamma) := \frac{1}{k} \sum_{i=1}^{n} \left\{ g_i^*(\phi) - [g_i^*(\phi) - \gamma]_+ - \frac{n-k}{n} \gamma \right\}.$$

The problem of robust meta-learning is then formulated as:

$$\min_{\phi} \max_{\gamma} \hat{F}(\phi, \mathbf{w}^*, \gamma) = \frac{1}{k} \sum_{i=1}^{n} \left\{ f_i(\phi, w_i^*, \gamma) := g_i^*(\phi) - [g_i^*(\phi) - \gamma]_+ - \frac{n-k}{n} \gamma \right\}$$

$$s.t. \ \ w_i^*(\phi) = \arg\min_{w_i} g_i(\phi, w_i),$$

where $\phi$ is the parameter of meta-model, and $\mathbf{w} = [w_i, ..., w_n]^T$ is the vector of task specific parameters.

Inspired by [10], we introduce a smoothed version of the upper-level loss function by incorporating Gaussian noise into the indicator function for alignment with the assumption of our MemCS algorithm:

$$\tilde{F}(\phi, \mathbf{w}^*, \gamma) = \frac{1}{k} \sum_{i=1}^{n} \left\{ f_i(\phi, w_i^*, \gamma) := g_i^*(\phi) - [g_i^*(\phi) - \gamma + \epsilon Z]_+ - \frac{n-k}{n} \gamma \right\},$$

where $\epsilon > 0$ is the smoothing parameter, and $Z \sim \mathcal{N}(0, 1)$ is standard normal random variable.

# B   Implementation Details and Extra Experimental Results

## B.1   Datasets Description

**Deep AUC Maximization.** We assess our methodology using four datasets. The first dataset, *CIFAR100* [29], serves primarily for classification endeavors. Within the context of the multi-block deep AUC maximization challenge, we treat each of the 100 categories as an individual block, with samples belonging to a specific category considered positive for that block. This dataset comprises $60,000$ images, each measuring $32 \times 32$ pixels, divided into $50,000$ training and $10,000$ testing images. The *CelebA* [40] dataset encompasses 202,599 facial images, each annotated with a diverse set of attributes from $40$ different features. The *CheXpert* [26] dataset includes $224,316$ chest radiograph images from $65,240$ patients, marked for $14$ distinct observations. Adhering to the methodology proposed in [24], we employ a simplified version of CheXpert with a reduced image resolution and omit the *Fracture* observation due to insufficient positive samples. Lastly, the *OGBG-MolPCBA* [25] dataset is employed to predict molecular properties, representing each molecule as a graph with atoms as nodes and chemical bonds as edges, featuring $437,929$ such graphs annotated across 128 properties.

**Robust Meta-learning.** Our experiments are conducted over two popular datasets for few-shot learning: Mini-ImageNet [53] and Tiered-ImageNet [47]. Both datasets are subsets of the ILSVRC-12 dataset. Mini-ImageNet comprises 100 classes, each containing 600 images with dimensions of $84 \times 84$ pixels. The 100 classes are distributed among training, validation, and testing sets with a ratio of 64:16:20, respectively. In contrast, Tiered-ImageNet is a more extensive and challenging dataset, featuring 779,165 images annotated across 608 classes. These classes are further organized into 34 categories, with 20 categories designated for training, 6 for validation, and 8 for testing.

## B.2   Implementation Details

**Deep AUC Maximization.** For the CIFAR100 and the CelebA datasets, we use the ResNet18 architecture. For the large-scale CheXpert dataset, we opt for the DenseNet121 model pre-trained on ImageNet. When dealing with the OGBG-MolPCBA graph dataset, the Graph Isomorphism Network (GIN) is used for training. All experimental runs are performed using a single NVIDIA RTX 6000 GPU. Regarding hyperparameters, we set the total training epoch to 2000 for the CIFAR100 and 100 for the OGBG-MolPCBA datasets, adjust it to $40$ for CelebA, and reduce it to 6 for CheXpert. The learning rate for the optimal approximator $\mathbf{v}$ is uniformly set to $\eta_{\mathbf{v}} = 0.1$ across all experiments, with $\eta_{\mathbf{w}} = \eta_{\mathbf{u}} = \eta_{\mathbf{v}}/\lambda$ to maintain gradient magnitude consistency between $\mathbf{u}$ and $\mathbf{v}$. This consistency is vital due to the influence of the $\lambda$ parameter in the Lagrangian function on the update process for $\mathbf{u}$.

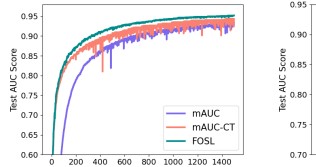 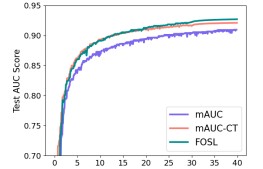

Table A1: Comparison of average iteration time and total training time of our method and AUC-CT[24] on small scale dataset (CelebA) and large scale dataset(CheXpert).

Figure A1: Test AUC score along with training epochs on the CIFAR100 (left) and the CelebA (right) datasets.

| Method | CelebA | CheXpert |
|---|---|---|
| FOSL | 0.55s/8.2h | 0.78s/7.3h |
| AUC-CT[24] | 0.69s/10.3h | 0.83s/7.7h |

Table A2: Comparison between FOSL and MemCS on robust meta-learning task.

| Algorithm | Best Test Accuracy | Model Parameter Size | Average Iteration Time |
|---|---|---|---|
| MemCS | 69.25 | 0.433MB | 3.15s |
| FOSL | 67.75 | 611.167MB | 1.42s |

Learning rate decay is applied to CelebA starting at the 30th epoch and to CheXpert at the 4th epoch, whereas no decay strategy is applied for CIFAR100 and OGBG-MolPCBA.

**Robust Meta-learning.** We adopt the Adam optimizer to update the meta-model in the context of MAML. For the hyper-parameters of MAML, we set the learning rate of meta-model update as $\eta_{meta} = 0.02$, and set the learning rate of fast adaptation as $\eta_{adapt} = 0.02$, with an adaptation step of 15. To be consistent with the DAM experiments, we configure the learning rate of the optimal approximator as $\eta_{\mathbf{v}} = \eta_{adapt} = 0.02$, and the learning rate of $\mathbf{w}$ and meta-model parameter $\phi$ as $\eta_{\phi} = \eta_{\mathbf{w}} = \eta_{\mathbf{v}}/\lambda$ in the implementation. In practice, setting $\lambda$ to 3 results in favorable performance. All experiments are conducted on a single NVIDIA RTX 6000 GPU using a widely used lightweight model featuring 4 convolutional layers (CNN4).

## B.3 Extra Results on Deep AUC Maximization

This section presents the visualization of statistics throughout the training process and compares the computational costs with our method and AUC-CT [24]. To better grasp the training dynamics, we charted the test AUC scores across various training epochs for both the CIFAR100 and CelebA datasets, as shown in Figure A1. The findings demonstrate that our method not only exhibits enhanced generalization capabilities but also greater stability.

Moreover, to assess efficiency across varying dataset scales, we examined the average iteration times and total training time of our FOSL algorithm and AUC-CT [24] on different-sized datasets ($32 \times 32$ in CIFAR100 vs. $224 \times 224$ in CheXpert) as detailed in Table A1. Note that the training time largely depends on the implementation and hyperparameters, such as the number of sampled tasks and batch sizes per task, suggesting that computational costs can be adjusted by modifying these hyperparameters according to the dataset. The result in Table A1 shows the ability to control training costs for datasets of various scales, which is indicated by the small gap between the total training time on CelebA and CheXpert datasets. Additionally, our method demonstrated a faster training pace compared to AUC-CT [24] under the same training settings. Note that the implementation of AUC-CT [24] avoided the calculation of second-order matrices so that the computational cost is more controllable with an increased dataset scale.

## B.4 Comparison between FOSL and MemCS

In this section, we compare our two proposed methods within the same experimental setting, i.e. robust meta-learning. To make it compatible with our FOSL algorithm, we slightly adjusted our training setting so that the number of training tasks in the dataset is known (20000 tasks) while maintaining the same testing procedures as those used with the MemCS algorithm. The results, including test accuracy, memory cost, and computational cost, are detailed in Table A2. The result shows that using FOSL in a robust meta-learning setting can introduce a greatly increased memory cost, which is especially significant for a small model. However, the single-loop nature of FOSL can drastically shorten the average iteration time during training. This makes the FOSL algorithm a potentially advantageous choice in scenarios involving larger base models and fewer blocks.

## C Notations

The original problem we solve here is

$$\min_{x \in \mathbb{R}^{d_x}} \max_{y \in \mathbb{R}^{d_y}} F\big(x, y, \mathbf{z}^*(x)\big) := \frac{1}{n} \sum_{i=1}^{n} f_i\big(x, y, z_i^*(x)\big)$$

$$= \frac{1}{n} \sum_{i=1}^{n} \mathbb{E}_\xi \big[ f_i\big(x, y, z_i^*(x); \xi_i\big) \big]$$

$$\text{s.t. } z_i^*(x) = \arg \min_{z \in \mathbb{R}^{d_z}} g_i(x, z) = \mathbb{E}_\zeta \big[ g_i(x, z; \zeta_i) \big].$$

Moreover, we define $z_{\lambda,i}^*(x) = \arg \min_z \mathcal{L}_i(x, y^*(x), z, v)$ and $y^*(x) = \arg \max_y F\big(x, y, \mathbf{z}^*(x)\big)$. For the convenience of proof, we also define

$$\Phi(x) = F\big(x, y^*(x), \mathbf{z}^*(x)\big) = \frac{1}{n} \sum_{i=1}^{n} f_i\big(x, y^*(x), z_i^*(x)\big).$$

For the notational convenience of FOSL, we define the estimators of client set $I_t$ as

$$h_x^t := \frac{1}{|I_t|} \sum_{i \in I_t} \Big[ h_{x,i}^t := \nabla_x \mathcal{L}_i(x_t, y_t, z_{i,t}, v_{i,t}; \xi_{x,i}^t) \Big],$$

$$h_y^t := \frac{1}{|I_t|} \sum_{i \in I_t} \Big[ h_{y,i}^t := \nabla_y f_i(x_t, y_t, v_{i,t}; \xi_{y,i}^t) \Big],$$

$$h_z^t := \frac{1}{|I_t|} \sum_{i \in I_t} \Big[ h_{y,i}^t := \nabla_z \mathcal{L}_i(x_t, y_t, z_{i,t}; \xi_{z,i}^t) \Big],$$

$$h_v^t := \frac{1}{|I_t|} \sum_{i \in I_t} \Big[ h_{v,i}^t := \nabla_z g_i(x_t, v_{i,t}; \xi_{v,i}^t) \Big]. \tag{4}$$

Since we sample tasks without replacement and our estimators are unbiased, we have the expectations of estimators as

$$\widetilde{h}_x^t := \mathbb{E}[h_x^t] = \frac{1}{n} \sum_{i=1}^{n} \Big[ \widetilde{h}_{x,i}^t := \nabla_x \mathcal{L}_i(x_t, y_t, z_{i,t}, v_{i,t}) \Big],$$

$$\widetilde{h}_y^t := \mathbb{E}[h_y^t] = \frac{1}{n} \sum_{i=1}^{n} \Big[ \widetilde{h}_{y,i}^t := \nabla_y f(x_t, y_t, v_{i,t}) \Big],$$

$$\widetilde{h}_z^t := \mathbb{E}[h_z^t] = \frac{1}{n} \sum_{i=1}^{n} \Big[ \widetilde{h}_{z,i}^t := \nabla_z \mathcal{L}_i(x_t, y_t, z_{i,t}, v_{i,t}) \Big],$$

$$\widetilde{h}_v^t := \mathbb{E}[h_v^t] = \frac{1}{n} \sum_{i=1}^{n} \Big[ \widetilde{h}_{v,i}^t := \nabla_z g(x_t, v_{i,t}) \Big]. \tag{5}$$

We also define the optimal Lagrangian estimator of $x$ and its gradients as

$$\mathcal{L}^*(x) := \frac{1}{n} \sum_{i=1}^{n} \mathcal{L}_i\big(x, y^*(x), z_{\lambda,i}^*(x), z_i^*(x)\big),$$

$$\mathcal{H}^*(x) := \frac{1}{n} \sum_{i=1}^{n} \Big[ \mathcal{H}_i^*(x) := \nabla_x \mathcal{L}\big(x, y^*(x), z_{\lambda,i}^*(x), z_i^*(x)\big) \Big]$$

$$= \frac{1}{n} \sum_{i=1}^{n} \Big[ \nabla_x f_i\big(x, y^*(x), z_{\lambda,i}^*(x)\big) + \lambda \Big( \nabla_x g_i\big(x, z_{\lambda,i}^*(x)\big) - \nabla_x g_i\big(x, z_i^*(x)\big) \Big) \Big]. \tag{6}$$

# D Proofs of Preliminary Lemmas

## D.1 Some basic properties

**Lemma D.1.** *Under Assumptions 4.3, 4.4, for $\forall \lambda \geq \frac{2L_{f,1}}{\mu_g}$, both $\mathcal{L}_i(x,y,z,v)$ and $\mathcal{L}(x,y,z,v)$ are $(\frac{\lambda \mu_g}{2})$-strongly convex in $z$ and $L_{\lambda,1}$-smooth in $(x,y,z)$, where $L_{\lambda,1} := 3\lambda L_{g,1}$.*

*Proof.* Since $\lambda \geq \frac{2L_{f,1}}{\mu_g} \geq \frac{2L_{f,1}}{L_{g,1}}$, we have that

$$\|\nabla^2_{zz}\mathcal{L}_i(x,y,z,v)\| = \|\nabla^2_{zz}f_i(x,y,z) + \lambda\nabla^2_{zz}g_i(x,z)\| \geq \|\lambda\nabla^2_{zz}g_i(x,z)\| - \|\nabla^2_{zz}f_i(x,y,z)\| \geq \frac{\lambda\mu_g}{2},$$

$$\|\nabla^2_{zz}\mathcal{L}(x,y,z,v)\| = \|\nabla^2_{zz}F(x,y,z) + \lambda\nabla^2_{zz}G(x,z)\| \geq \|\lambda\nabla^2_{zz}G(x,z)\| - \|\nabla^2_{zz}F(x,y,z)\| \geq \frac{\lambda\mu_g}{2};$$

$$\|\nabla^2\mathcal{L}_i(x,y,z,v)\| = \|\nabla^2 f_i(x,y,z) + \lambda\nabla^2 g_i(x,z) - \lambda\nabla^2 g_i(x,v)\| \leq L_{f_1} + 2\lambda L_{g,1} \leq 3\lambda L_{g,1} =: L_{\lambda,1},$$

$$\|\nabla^2\mathcal{L}(x,y,z,v)\| = \|\nabla^2 F(x,y,z) + \lambda\nabla^2 G(x,z) - \lambda\nabla^2 G(x,v)\| \leq L_{f_1} + 2\lambda L_{g,1} \leq 3\lambda L_{g,1} =: L_{\lambda,1}.$$

Then the proof is complete. $\qquad\square$

**Lemma D.2.** *Under Assumptions 4.3, 4.4, for $\lambda \geq \max\left\{\frac{2L_{f,1}}{\mu_g}, (1 + \frac{L_{g,1}}{\mu_g})\frac{L^2_{f,1}}{3\mu_f L_{g,1}}\right\}$, we have $\|\nabla z_i^*(x)\| \leq \frac{L_{g,1}}{\mu_g}$, $\|\nabla z_{\lambda,i}^*(x)\| \leq \frac{12L_{g,1}}{\mu_g}$ and $\|\nabla y^*(x)\| \leq \left(1 + \frac{L_{g,1}}{\mu_g}\right)\frac{L_{f,1}}{\mu_f}$.*

*Proof.* Recall that we define $z_i^*(x) := \arg\min_z g_i(x,z)$ and $z_{\lambda,i}^*(x) := \arg\min_z \mathcal{L}_i(x, y^*(x), z, v)$. Then we have $\nabla_z g_i(x, z_i^*(x)) = \mathbf{0}$ and $\nabla_z \mathcal{L}_i(x, y^*(x), z_{\lambda,i}^*(x), v) = \mathbf{0}$. Via implicit function theorem, we obtain

$$\nabla^2_{xz}g_i(x, z_i^*(x)) + (\nabla z_i^*(x))^T \nabla^2_{zz}g_i(x, z_i^*(x)) = \mathbf{0},$$

$$\nabla^2_{xz}\mathcal{L}_i(x, y^*(x), z_{\lambda,i}^*(x), v) + (\nabla y^*(x))^T \nabla^2_{yz}\mathcal{L}_i(x, y^*(x), z_{\lambda,i}^*(x), v)$$
$$+ (\nabla z_{\lambda,i}^*(x))^T \nabla^2_{zz}\mathcal{L}_i(x, y^*(x), z_{\lambda,i}^*(x), v) = \mathbf{0}. \tag{7}$$

To measure that Lipschitz continuity of $z_i^*(x)$ and $z_{\lambda,i}^*(x)$ w.r.t. x, we take spectral norm of $\nabla z_i^*(x)$ and $\nabla z_{\lambda,i}^*(x)$ as

$$\|\nabla z_i^*(x)\| = \| - \nabla^2_{xz}g_i(x, z_i^*(x))[\nabla^2_{zz}g_i(x, z_i^*(x))]^{-1}\| \overset{(a)}{\leq} \frac{L_{g,1}}{\mu_g},$$

$$\|\nabla z_{\lambda,i}^*(x)\| = \| - \nabla^2_{xz}\mathcal{L}_i(x, y^*(x), z_{\lambda,i}^*(x), v)[\nabla^2_{zz}\mathcal{L}_i(x, y^*(x), z_{\lambda,i}^*(x), v)]^{-1}$$
$$- (\nabla y^*(x))^T \nabla^2_{yz}\mathcal{L}_i(x, y, z_{\lambda,i}^*(x), v)[\nabla^2_{zz}\mathcal{L}_i(x, y^*(x), z_{\lambda,i}^*(x), v)]^{-1}\|, \tag{8}$$

where (a) uses Assumption 4.4. Similarly, for $y^*(x)$, we have $\nabla_y F(x, y^*(x), \mathbf{z}^*(x)) = \mathbf{0}$. Via implicit function theorem, we have

$$\frac{1}{n}\sum_{i=1}^n \left[\nabla^2_{xy}f_i(x, y^*(x), z_i^*(x)) + (\nabla y^*(x))^T \nabla^2_{yy}f_i(x, y^*(x), z_i^*(x))\right.$$
$$\left. + (\nabla z_i^*(x))^T \nabla^2_{zy}f_i(x, y^*(x), z_i^*(x))\right] = \mathbf{0}, \tag{9}$$

which indicates

$$\|\nabla y^*(x_t)\| \leq \left\|\frac{1}{n}\sum_{i=1}^n \left[\nabla^2_{xy}f_i(x, y^*(x), z_i^*(x)) + (\nabla z_i^*(x))^T \nabla^2_{yz}f_i(x, y^*(x), z_i^*(x))\right]\right\|$$
$$\cdot \|[\nabla^2_{yy}F(x, y^*(x), \mathbf{z}^*(x))]^{-1}\|$$
$$\leq \left(\frac{1}{n}\sum_{i=1}^n \left\|[\nabla^2_{xy}f_i(x, y^*(x), z_i^*(x)) + (\nabla z_i^*(x))^T \nabla^2_{yz}f_i(x, y^*(x), z_i^*(x))]\right\|\right)$$

$$\cdot \left\| \left[ \nabla_{yy}^2 F\big(x, y^*(x), \mathbf{z}^*(x)\big) \right]^{-1} \right\|$$

$$\overset{(a)}{\leq} \left( 1 + \frac{L_{g,1}}{\mu_g} \right) \frac{L_{f,1}}{\mu_f},$$

where (a) uses Assumption 4.3 and Assumption 4.4. Back to the second equation in eq. (8), with $\|\nabla y^*(x_t)\| \leq (1 + \frac{L_{g,1}}{\mu_g})\frac{L_{f,1}}{\mu_f}$, we have

$$
\begin{aligned}
\|\nabla z_{\lambda,i}^*(x)\| \leq & \left\| \nabla_{xz}^2 \mathcal{L}_i\big(x, y^*(x), z_{\lambda,i}^*(x), v\big) + \big(\nabla y^*(x)\big)^T \nabla_{yz}^2 \mathcal{L}_i\big(x, y, z_{\lambda,i}^*(x), v\big) \right\| \\
& \cdot \left\| \left[ \nabla_{zz}^2 \mathcal{L}_i\big(x, y^*(x), z_{\lambda,i}^*(x), v\big) \right]^{-1} \right\| \\
\overset{(a)}{\leq} & \left[ 3\lambda L_{g,1} + \left(1 + \frac{L_{g,1}}{\mu_g}\right) \frac{L_{f,1}^2}{\lambda \mu_g} \right] \frac{2}{\lambda \mu_g} \\
\overset{(b)}{\leq} & \frac{12 L_{g,1}}{\mu_g},
\end{aligned}
$$

where (a) uses Lemma D.1 and (b) uses $\lambda \geq (1 + \frac{L_{g,1}}{\mu_g})\frac{L_{f,1}^2}{3\mu_f L_{g,1}}$. Then the proof is complete. $\qquad \square$

**Lemma D.3.** *Under Assumptions 4.3, 4.4, the optimal solutions $z_i^*(x)$, $z_{\lambda,i}^*(x)$ and $y_i^*(x)$ are $L_{*,z}$-, $L_{*,z_\lambda}$- and $L_{*,y}$-smooth respectively, where we define $L_{*,z}$, $L_{*,z_\lambda}$ and $L_{*,y}$ as*

$$L_{*,z} := \frac{L_{g,2}}{\mu_g}\left(1 + \frac{L_{g,1}}{\mu_g}\right)^2, \quad L_{*,z_\lambda} := \left(1 + \left(1 + \frac{L_{g,1}}{\mu_g}\right)\frac{L_{f,1}}{\mu_f} + \frac{12 L_{g,1}}{\mu_g}\right),$$

$$L_{*,y} := \left(1 + \frac{L_{g,1}}{\mu_g} + \left(1 + \frac{L_{g,1}}{\mu_g}\right)\frac{L_{f,1}}{\mu_f}\right)^2 \frac{L_{f,2}}{\mu_f} + \left(1 + \frac{L_{g,1}}{\mu_g}\right)^2 \frac{L_{f,1} L_{g,2}}{\mu_f \mu_g}$$

*for any $i \in \{1, ..., n\}$, where we assume $\lambda \geq \{2L_{f,1}/\mu_g, (1 + \frac{L_{g,1}}{\mu_g})\frac{L_{f,1}^2}{3\mu_f L_{g,1}}, (1 + \frac{L_{g,1}}{\mu_g})\frac{L_{f,1} L_{f,2}}{3\mu_f L_{g,1}}, \frac{L_{f,1} L_{*,y}}{6 L_{g,1}}(1 + (1 + \frac{L_{g,1}}{\mu_g})\frac{L_{f,1}}{\mu_f} + \frac{12 L_{g,1}}{\mu_g})^{-1}, ((1 + \frac{L_{g,1}}{\mu_g})\frac{L_{f,1}}{\mu_f} + 1)\frac{L_{f,1}}{L_{g,1}}\}$.*

*Proof.* Since $z_i^*(x) = \arg\min_z g_i(x, z)$, we have $\nabla_z g_i\big(x, z^*(x)\big) = \mathbf{0}$, which indicates that

$$\nabla_{xz}^2 g_i\big(x, z^*(x)\big) + \nabla z^*(x) \nabla_{zz}^2 g_i\big(x, z^*(x)\big) = \mathbf{0}.$$

For any $x_1, x_2 \in \mathbb{R}^{d_x}$, we have

$$
\begin{aligned}
& \|\nabla z_i^*(x_1) - \nabla z_i^*(x_2)\| \\
= & \left\| \nabla_{xz}^2 g_i\big(x_1, z_i^*(x_1)\big) \left[ \nabla_{zz}^2 g_i\big(x_1, z_i^*(x_1)\big) \right]^{-1} - \nabla_{xz}^2 g_i\big(x_2, z_i^*(x_2)\big) \left[ \nabla_{zz}^2 g_i\big(x_2, z_i^*(x_2)\big) \right]^{-1} \right\| \\
\leq & \left\| \left[ \nabla_{xz}^2 g_i\big(x_1, z_i^*(x_1)\big) - \nabla_{xz}^2 g_i\big(x_2, z_i^*(x_2)\big) \right] \left[ \nabla_{zz}^2 g_i\big(x_1, z_i^*(x_1)\big) \right]^{-1} \right\| \\
& + \left\| \nabla_{xz}^2 g_i\big(x_2, z_i^*(x_2)\big) \left( \left[ \nabla_{zz}^2 g_i\big(x_1, z_i^*(x_1)\big) \right]^{-1} - \left[ \nabla_{zz}^2 g_i\big(x_2, z_i^*(x_2)\big) \right]^{-1} \right) \right\| \\
\leq & \left\| \nabla_{xz}^2 g_i\big(x_1, z_i^*(x_1)\big) - \nabla_{xz}^2 g_i\big(x_2, z_i^*(x_2)\big) \right\| \cdot \left\| \left[ \nabla_{zz}^2 g_i\big(x_1, z_i^*(x_1)\big) \right]^{-1} \right\| \\
& + \left\| \nabla_{xz}^2 g_i\big(x_2, z_i^*(x_2)\big) \right\| \cdot \left\| \left[ \nabla_{zz}^2 g_i\big(x_1, z_i^*(x_1)\big) \right]^{-1} - \left[ \nabla_{zz}^2 g_i\big(x_2, z_i^*(x_2)\big) \right]^{-1} \right\| \\
\overset{(a)}{\leq} & \frac{1}{\mu_g} \left\| \nabla_{xz}^2 g_i\big(x_1, z_i^*(x_1)\big) - \nabla_{xz}^2 g_i\big(x_2, z_i^*(x_2)\big) \right\| \\
& + \frac{L_{g,1}}{\mu_g^2} \left\| \nabla_{zz}^2 g_i\big(x_2, z_i^*(x_2)\big) - \nabla_{zz}^2 g_i\big(x_1, z_i^*(x_1)\big) \right\| \\
\overset{(a)}{\leq} & \frac{L_{g,2}}{\mu_g}\left(1 + \frac{L_{g,1}}{\mu_g}\right)\left(\|x_1 - x_2\| + \|z_i^*(x_1) - z_i^*(x_2)\|\right) \\
\overset{(b)}{\leq} & \frac{L_{g,2}}{\mu_g}\left(1 + \frac{L_{g,1}}{\mu_g}\right)^2 \|x_1 - x_2\|,
\end{aligned}
\tag{10}
$$

where (a) uses Assumption 4.3, 4.4 and $(A^{-1} - B^{-1}) = A^{-1}(B - A)B^{-1}$; (b) follows from Lemma D.2. Next, plug $x = x_1$ and $x = x_2$ into eq. (9) and differentiate these two equations, then we get

$$\left(\nabla y^*(x_1)\right)^T \nabla_{yy}^2 F\left(x_1, y^*(x_1), z_i^*(x_1)\right) - \left(\nabla y^*(x_2)\right)^T \nabla_{yy}^2 F\left(x_2, y^*(x_2), z_i^*(x_2)\right)$$

$$= \left(\nabla y^*(x_1) - \nabla y^*(x_2)\right)^T \nabla_{yy}^2 F\left(x_1, y^*(x_1), z_i^*(x_1)\right)$$

$$+ \left(\nabla y^*(x_2)\right)^T \left(\nabla_{yy}^2 F\left(x_1, y^*(x_1), z_i^*(x_1)\right) - \nabla_{yy}^2 F\left(x_2, y^*(x_2), z_i^*(x_2)\right)\right), \qquad (11)$$

and by using eq. (9), we get

$$\frac{1}{n}\sum_{i=1}^{n}\left[\left(\nabla y^*(x_1)\right)^T \nabla_{yy}^2 f_i\left(x_1, y^*(x_1), z_i^*(x_1)\right) - \left(\nabla y^*(x_2)\right)^T \nabla_{yy}^2 f_i\left(x_2, y^*(x_2), z_i^*(x_2)\right)\right]$$

$$=\frac{1}{n}\sum_{i=1}^{n}\left[\nabla_{xy}^2 f_i\left(x_1, y^*(x_1), z_i^*(x_1)\right) - \nabla_{xy}^2 f_i\left(x_2, y^*(x_2), z_i^*(x_2)\right)\right.$$

$$+ \left(\nabla z_i^*(x_1) - \nabla z_i^*(x_2)\right)^T \nabla_{zy}^2 f_i\left(x_1, y^*(x_1), z_i^*(x_1)\right)$$

$$\left. + \left(\nabla z_i^*(x_1)\right)^T \left(\nabla_{zy}^2 f_i\left(x_1, y^*(x_1), z_i^*(x_1)\right) - \nabla_{zy}^2 f_i\left(x_1, y^*(x_1), z_i^*(x_1)\right)\right)\right]. \quad (12)$$

By combining eq. (11), eq. (12) and taking norm, we have

$$\left\|\nabla y^*(x_1) - \nabla y^*(x_2)\right\|$$

$$\leq \left\|\left[\nabla_{yy}^2 F\left(x_1, y^*(x_1), z_i^*(x_1)\right)\right]^{-1}\right\|$$

$$\cdot \left(\left\|\frac{1}{n}\sum_{i=1}^{n} \nabla_{xy}^2 f_i\left(x_1, y^*(x_1), z_i^*(x_1)\right) - \nabla_{xy}^2 f_i\left(x_2, y^*(x_2), z_i^*(x_2)\right)\right\|\right.$$

$$+ \left\|\frac{1}{n}\sum_{i=1}^{n}\left(\nabla z_i^*(x_1) - \nabla z_i^*(x_2)\right)^T \nabla_{zy}^2 f_i\left(x_1, y^*(x_1), z_i^*(x_1)\right)\right\|$$

$$+ \left\|\frac{1}{n}\sum_{i=1}^{n}\left(\nabla z_i^*(x_1)\right)^T \left(\nabla_{zy}^2 f_i\left(x_1, y^*(x_1), z_i^*(x_1)\right) - \nabla_{zy}^2 f_i\left(x_1, y^*(x_1), z_i^*(x_1)\right)\right)\right\|$$

$$\left.+ \left\|\nabla y^*(x_2)\right\| \cdot \left\|\frac{1}{n}\sum_{i=1}^{n} \nabla_{yy}^2 f_i\left(x_1, y^*(x_1), z_i^*(x_1)\right) - \nabla_{yy}^2 f_i\left(x_2, y^*(x_2), z_i^*(x_2)\right)\right\|\right)$$

$$\overset{(a)}{\leq} \frac{1}{\mu_f}\left(1 + \frac{L_{g,1}}{\mu_g} + \left(1 + \frac{L_{g,1}}{\mu_g}\right)\frac{L_{f,1}}{\mu_f}\right)L_{f,2}$$

$$\cdot \left(\|x_1 - x_2\| + \|y^*(x_1) - y^*(x_2)\| + \frac{1}{n}\sum_{i=1}^{n}\|z_i^*(x_1) - z_i^*(x_2)\|\right)$$

$$+ \frac{L_{f,1}}{\mu_f}\left\|\frac{1}{n}\sum_{i=1}^{n} \nabla z_i^*(x_1) - \nabla z_i^*(x_2)\right\|$$

$$\overset{(b)}{\leq} \left[\left(1 + \frac{L_{g,1}}{\mu_g} + \left(1 + \frac{L_{g,1}}{\mu_g}\right)\frac{L_{f,1}}{\mu_f}\right)^2 \frac{L_{f,2}}{\mu_f} + \left(1 + \frac{L_{g,1}}{\mu_g}\right)^2 \frac{L_{f,1}L_{g,2}}{\mu_f\mu_g}\right]\|x_1 - x_2\|,$$

where (a) uses Assumption 4.4 and Lemma D.2; (b) follows from Assumption 4.4, Lemma D.2, and eq. (10). Similarly to eq. (10), from eq. (7), if we simplify the notation as

$$A_1 = \nabla_{xz}^2 \mathcal{L}_i\left(x_1, y^*(x_1), z_{\lambda,i}^*(x_1), v_1\right) + \left(\nabla y^*(x_1)\right)^T \nabla_{yz}^2 \mathcal{L}_i\left(x_1, y^*(x_1), z_{\lambda,i}^*(x), v_1\right),$$

$$B_1 = \nabla_{zz}^2 \mathcal{L}_i\left(x_1, y^*(x_1), z_{\lambda,i}^*(x_1), v_1\right)$$

$$A_2 = \nabla_{xz}^2 \mathcal{L}_i\left(x_2, y^*(x_2), z_{\lambda,i}^*(x_2), v_2\right) + \left(\nabla y^*(x_2)\right)^T \nabla_{yz}^2 \mathcal{L}_i\left(x_2, y^*(x_1), z_{\lambda,i}^*(x), v_2\right)$$

$$B_2 = \nabla_{zz}^2 \mathcal{L}_i\left(x_2, y^*(x_2), z_{\lambda,i}^*(x_2), v_2\right),$$

then we have

$$\left\|\nabla z_{\lambda,i}^*(x_1) - \nabla z_{\lambda,i}^*(x_2)\right\| = \left\|A_1 B_1^{-1} - A_2 B_2^{-1}\right\|$$

$$\leq \|(A_1 - A_2)B_1^{-1}\| + \|A_2(B_1^{-1} - B_2^{-1})\|$$
$$\leq \|A_1 - A_2\| \cdot \|B_1^{-1}\| + \|A_2\| \cdot \|B_1^{-1} - B_2^{-1}\|$$
$$\leq \|A_1 - A_2\| \cdot \|B_1^{-1}\| + \|A_2\| \cdot \|B_1^{-1}\| \cdot \|B_2^{-1}\| \cdot \|B_1 - B_2\|. \quad (13)$$

For the first term in eq. (13), via Lemma D.1, we have

$$\|A_1 - A_2\|$$
$$\leq 3\lambda L_{g,1}\big(\|x_1 - x_2\| + \|y^*(x_1) - y^*(x_2)\| + \|z_{\lambda,i}^*(x_1) - z_{\lambda,i}^*(x_2)\|\big)$$
$$\quad + L_{f,1}\|\nabla y^*(x_1) - \nabla y^*(x_2)\|$$
$$\quad + L_{f,2}\|\nabla y^*(x_2)\| \cdot \big(\|x_1 - x_2\| + \|y^*(x_1) - y^*(x_2)\| + \|z_{\lambda,i}^*(x_1) - z_{\lambda,i}^*(x_2)\|\big)$$
$$\overset{(a)}{\leq} \left(3\lambda L_{g,1} + \left(1 + \frac{L_{g,1}}{\mu_g}\right)\frac{L_{f,1}L_{f,2}}{\mu_f}\right)\big(\|x_1 - x_2\| + \|y^*(x_1) - y^*(x_2)\| + \|z_{\lambda,i}^*(x_1) - z_{\lambda,i}^*(x_2)\|\big)$$
$$\quad + L_{f,1}\|\nabla y^*(x_1) - \nabla y^*(x_2)\|$$
$$\overset{(b)}{\leq} 6\lambda L_{g,1}\left(1 + \left(1 + \frac{L_{g,1}}{\mu_g}\right)\frac{L_{f,1}}{\mu_f} + \frac{12L_{g,1}}{\mu_g}\right)\|x_1 - x_2\| + L_{f,1}L_{*,y}\|x_1 - x_2\|$$
$$\overset{(c)}{\leq} 9\lambda L_{g,1}\left(1 + \left(1 + \frac{L_{g,1}}{\mu_g}\right)\frac{L_{f,1}}{\mu_f} + \frac{12L_{g,1}}{\mu_g}\right)\|x_1 - x_2\| \quad (14)$$

where (a) uses Lemma D.2; (b) follows from Lemmas D.2, D.3 and $\lambda \geq (1 + \frac{L_{g,1}}{\mu_g})\frac{L_{f,1}L_{f,2}}{3\mu_f L_{g,1}}$; (c) uses $\lambda \geq \frac{L_{f,1}L_{*,y}}{6L_{g,1}}\big[1 + (1 + \frac{L_{g,1}}{\mu_g})\frac{L_{f,1}}{\mu_f} + \frac{12L_{g,1}}{\mu_g}\big]^{-1}$. By using Lemma D.1, we have $\|B_1^{-1}\| \leq \frac{2}{\lambda\mu_g}$, $\|B_2^{-1}\| \leq \frac{2}{\lambda\mu_g}$, $\|B_1 - B_2\| \leq 3\lambda L_{g,1}\|x_1 - \|$ and

$$\|A_2\| = \big\|\nabla_{xz}^2\mathcal{L}_i\big(x_2, y^*(x_2), z_{\lambda,i}^*(x_2), v_2\big) + \big(\nabla y^*(x_2)\big)^T\nabla_{yz}^2\mathcal{L}_i\big(x_2, y^*(x_1), z_{\lambda,i}^*(x), v_2\big)\big\|$$
$$\leq \big\|\nabla_{xz}^2\mathcal{L}_i\big(x_2, y^*(x_2), z_{\lambda,i}^*(x_2), v_2\big)\big\| + \|\nabla y^*(x_2)\| \cdot \big\|\nabla_{yz}^2 f_i\big(x_2, y^*(x_1), z_{\lambda,i}^*(x)\big)\big\|$$
$$\overset{(a)}{\leq} (L_{f,1} + \lambda L_{g,1}) + \left(1 + \frac{L_{g,1}}{\mu_g}\right)\frac{L_{f,1}^2}{\mu_f} \overset{(b)}{\leq} 2\lambda L_{g,1}, \quad (15)$$

where (a) uses Assumption 4.4 and Lemma D.2; (b) uses $\lambda \geq \big((1 + \frac{L_{g,1}}{\mu_g})\frac{L_{f,1}}{\mu_f} + 1\big)\frac{L_{f,1}}{L_{g,1}}$. We also have

$$\|B_1 - B_2\| = 3\lambda L_{g,1}\big(\|x_1 - x_2\| + \|y^*(x_1) - y^*(x_2)\| + \|z_{\lambda,i}^*(x_1) - z_{\lambda,i}^*(x_2)\|\big)$$
$$\overset{(a)}{\leq} 3\lambda L_{g,1}\left(1 + \left(1 + \frac{L_{g,1}}{\mu_g}\right)\frac{L_{f,1}}{\mu_f} + \frac{12L_{g,1}}{\mu_g}\right)\|x_1 - x_2\|, \quad (16)$$

where (a) uses Lemma D.2. Combining eq. (14), eq. (15), eq. (16) with the results in Lemma D.1, we have

$$\|\nabla z_{\lambda,i}^*(x_1) - \nabla z_{\lambda,i}^*(x_2)\| \leq \|A_1 - A_2\| \cdot \|B_1^{-1}\| + \|A_2\| \cdot \|B_1^{-1}\| \cdot \|B_2^{-1}\| \cdot \|B_1 - B_2\|$$
$$\leq \left(\frac{18L_{g,1}}{\mu_g} + \frac{24L_{g,1}^2}{\mu_g^2}\right)\left(1 + \left(1 + \frac{L_{g,1}}{\mu_g}\right)\frac{L_{f,1}}{\mu_f} + \frac{12L_{g,1}}{\mu_g}\right)\|x_1 - x_2\|.$$

Thus, the proof is complete. $\qquad\square$

### D.2   Gap of Lower-level Optimal Points

**Lemma D.4.** *Under Assumptions 4.3, 4.4, for any given $x$ and , the gap between the optimal solutions of the lower-level problem $z_i^*(x)$ and the surrogate minimax problem $z_{\lambda,i}^*(x)$ can be bounded as*

$$\|z_{\lambda,i}^*(x) - z_i^*(x)\| \leq \frac{L_{f,0}}{\mu_g\lambda},$$
$$\|\nabla z_{\lambda,i}^*(x) - \nabla z_i^*(x)\| \leq \frac{1}{\lambda} \cdot \left[\frac{1}{\mu_g}\left(\frac{L_{f,0}L_{g,2}}{\mu_g} + L_{f,1}\left(1 + \left(1 + \frac{L_{g,1}}{\mu_g}\right)\frac{L_{f,1}}{\mu_f}\right)\right)\right.$$
$$\left. + \frac{6L_{g,1}}{\mu_g^2}\left(1 + \left(1 + \frac{L_{g,1}}{\mu_g}\right)\right) \cdot \left(L_{f,1} + \frac{L_{f,0}L_{g,2}}{\mu_g}\right)\right],$$

*for any $i \in \{1, ..., n\}$, where we assume $\lambda \geq \max\big\{\frac{2L_{f,1}}{\mu_g}, (1 + \frac{L_{g,1}}{\mu_g})\frac{L_{f,1}^2}{3\mu_f L_{g,1}}\big\}$.*

*Proof.* For each block, we can have that

$$\|z_{\lambda,i}^*(x) - z_i^*(x)\| \overset{(a)}{\leq} \frac{1}{\mu_g}\|\nabla_z g_i\big(x, z_{\lambda,i}^*(x)\big) - \nabla_z g_i\big(x, z_i^*(x)\big)\|$$

$$\overset{(b)}{\leq} \frac{1}{\mu_g \lambda}\|\nabla_z f_i(x, y^*(x), z_{\lambda,i}^*(x))\| \overset{(c)}{\leq} \frac{L_{f,0}}{\mu_g \lambda},$$

where (a) uses Assumption 4.3; (b) follows from the definition of $z_i^*(x)$ and $z_{\lambda,i}^*(x)$; (c) uses Assumption 4.4. For the second part, since $\nabla_z g_i\big(x, z_i^*(x)\big) = 0$, $\nabla_z \mathcal{L}_i\big(x, y^*(x), z_{\lambda,i}^*(x)\big) = 0$, we have

$$\nabla_{xz}^2 g_i\big(x, z_i^*(x)\big) + \nabla z_i^*(x)\nabla_{zz}^2 g_i\big(x, z_i^*(x)\big) = 0,$$

$$\nabla_{xz}^2 \mathcal{L}_i\big(x, y, z_{\lambda,i}^*(x)\big) + \nabla y^*(x)\nabla_{yz}^2 \mathcal{L}_i\big(x, y, z_{\lambda,i}^*(x)\big) + \nabla z_{\lambda,i}^*(x)\nabla_{zz}^2 \mathcal{L}_i\big(x, y, z_{\lambda,i}^*(x)\big) = 0,$$

which indicates that

$$\nabla z_i^*(x) = -\nabla_{xz}^2 g_i\big(x, z_i^*(x)\big)\big[\nabla_{zz}^2 g_i\big(x, z_i^*(x)\big)\big]^{-1},$$

$$\nabla z_{\lambda,i}^*(x) = -\big[\nabla_{xz}^2 \mathcal{L}_i\big(x, y^*(x), z_{\lambda,i}^*(x)\big) + \nabla y^*(x)\nabla_{yz}^2 \mathcal{L}_i\big(x, y, z_{\lambda,i}^*(x)\big)\big]\big[\nabla_{zz}^2 \mathcal{L}_i\big(x, y^*(x), z_{\lambda,i}^*(x)\big)\big]^{-1}$$

$$= -\frac{\big[\nabla_{xz}^2 \mathcal{L}_i\big(x, y^*(x), z_{\lambda,i}^*(x)\big) + \nabla y^*(x)\nabla_{yz}^2 \mathcal{L}_i\big(x, y, z_{\lambda,i}^*(x)\big)\big]}{\lambda}\left[\frac{\nabla_{zz}^2 \mathcal{L}_i\big(x, y^*(x), z_{\lambda,i}^*(x)\big)}{\lambda}\right]^{-1}.$$

Then the gap can be displayed as

$$\|\nabla z_{\lambda,i}^*(x) - \nabla z_i^*(x)\| \leq \left\|\nabla_{xz}^2 g_i\big(x, z_i^*(x)\big) - \frac{\nabla_{xz}^2 \mathcal{L}_i\big(x, y^*(x), z_{\lambda,i}^*(x)\big) + \nabla y^*(x)\nabla_{yz}^2 \mathcal{L}_i\big(x, y, z_{\lambda,i}^*(x)\big)}{\lambda}\right\|$$

$$\cdot\left\|\big[\nabla_{zz}^2 g_i\big(x, z_i^*(x)\big)\big]^{-1}\right\|$$

$$+ \left\|\frac{\nabla_{xz}^2 \mathcal{L}_i\big(x, y^*(x), z_{\lambda,i}^*(x)\big) + \nabla y^*(x)\nabla_{yz}^2 \mathcal{L}_i\big(x, y, z_{\lambda,i}^*(x)\big)}{\lambda}\right\|$$

$$\cdot\left\|\big[\nabla_{zz}^2 g_i\big(x, z_i^*(x)\big)\big]^{-1} - \left[\frac{\nabla_{zz}^2 \mathcal{L}_i\big(x, y^*(x), z_{\lambda,i}^*(x)\big)}{\lambda}\right]^{-1}\right\|$$

$$\overset{(a)}{\leq} \frac{1}{\mu_g}\left[\|\nabla_{xz}^2 g_i\big(x, z_i^*(x)\big) - \nabla_{xz}^2 g_i\big(x, z_{\lambda,i}^*(x)\big)\|\right.$$

$$\left. + \left\|\frac{\nabla_{xz}^2 f_i\big(x, y^*(x), z_{\lambda,i}^*(x)\big) + \nabla y^*(x)\nabla_{yz}^2 f_i\big(x, y, z_{\lambda,i}^*(x)\big)}{\lambda}\right\|\right]$$

$$+ 3L_{g,1}\Big(1 + \big(1 + \frac{L_{g,1}}{\mu_g}\big)\Big) \cdot \frac{2}{\mu_g^2}\Big(L_{f,1} + \frac{L_{f,0}L_{g,2}}{\mu_g}\Big)\frac{1}{\lambda}$$

$$\leq \frac{1}{\mu_g}\left[L_{g,2}\|z_{\lambda,i}^*(x) - z_i^*(x)\| + \frac{1}{\lambda}\cdot L_{f,1}\Big(1 + \big(1 + \frac{L_{g,1}}{\mu_g}\big)\frac{L_{f,1}}{\mu_f}\Big)\right]$$

$$+ \frac{1}{\lambda}\cdot\frac{6L_{g,1}}{\mu_g^2}\Big(1 + \big(1 + \frac{L_{g,1}}{\mu_g}\big)\Big) \cdot \Big(L_{f,1} + \frac{L_{f,0}L_{g,2}}{\mu_g}\Big)$$

$$\leq \frac{1}{\lambda}\cdot\frac{1}{\mu_g}\left[\frac{L_{f,0}L_{g,2}}{\mu_g} + L_{f,1}\Big(1 + \big(1 + \frac{L_{g,1}}{\mu_g}\big)\frac{L_{f,1}}{\mu_f}\Big)\right]$$

$$+ \frac{1}{\lambda}\cdot\frac{6L_{g,1}}{\mu_g^2}\Big(1 + \big(1 + \frac{L_{g,1}}{\mu_g}\big)\Big) \cdot \Big(L_{f,1} + \frac{L_{f,0}L_{g,2}}{\mu_g}\Big),$$

where (a) can be satisfied because

$$\left\|\big[\nabla_{zz}^2 g_i\big(x, z_i^*(x)\big)\big]^{-1} - \left[\frac{\nabla_{zz}^2 \mathcal{L}_i\big(x, y^*(x), z_{\lambda,i}^*(x)\big)}{\lambda}\right]^{-1}\right\|$$

$$= \left\|\big[\nabla_{zz}^2 g_i\big(x, z_i^*(x)\big)\big]^{-1}\right\| \cdot \left\|\frac{\nabla_{zz}^2 \mathcal{L}_i\big(x, y^*(x), z_{\lambda,i}^*(x)\big)}{\lambda} - \nabla_{zz}^2 g_i\big(x, z_i^*(x)\big)\right\|$$

$$\cdot\left\|\left[\frac{\nabla_{zz}^2 \mathcal{L}_i\big(x, y^*(x), z_{\lambda,i}^*(x)\big)}{\lambda}\right]^{-1}\right\|$$

$$\overset{(a.1)}{\leq} \frac{2}{\mu_g^2}\left(\left\|\frac{\nabla_{zz}^2 f_i\big(x,y^*(x),z_{\lambda,i}^*(x)\big)}{\lambda}\right\| + \left\|\nabla_{zz}^2 g_i\big(x,z_{\lambda,i}^*(x)\big) - \nabla_{zz}^2 g_i\big(x,z_i^*(x)\big)\right\|\right)$$

$$\leq \frac{2}{\mu_g^2}\left(\frac{L_{f,1}}{\lambda} + L_{g,2}\big\|z_{\lambda,i}^*(x) - z_i^*(x)\big\|\right)$$

$$\leq \frac{2}{\mu_g^2}\left(L_{f,1} + \frac{L_{f,0}L_{g,2}}{\mu_g}\right)\frac{1}{\lambda},$$

where (a.1) uses Assumption 4.3 and Lemma D.1. Then, the proof is complete. $\qquad\square$

**Lemma D.5.** *Under Assumptions 4.3, 4.4, the gap between $\nabla\Phi(x)$ and $\mathcal{H}^*(x)$ can be bounded as*

$$\big\|\nabla\Phi(x) - \mathcal{H}^*(x)\big\|^2 \leq \frac{C_{gap}}{\lambda^2},$$

*where $C_{gap} := 3\Big(1 + \frac{L_{f,1}^2}{\mu_g^2}\Big)L_{f,1}^2\Big(\frac{L_{f,0}}{\mu_g}\Big)^2 + \Big(1 + \frac{L_{g,1}^2}{\mu_g^2}\Big)\frac{3L_{g,1}^2}{2}\Big(\frac{L_{f,0}}{\mu_g}\Big)^4$ and we assume $\lambda \geq \max\Big\{\frac{2L_{f,1}}{\mu_g}, (1 + \frac{L_{g,1}}{\mu_g})\frac{L_{f,1}^2}{3\mu_f L_{g,1}}\Big\}.$*

*Proof.* By the definitions of $\nabla F\big(x,y^*(x),\mathbf{z}^*(x)\big)$ and $\mathcal{H}^*(x)$, we have

$$\big\|\nabla F\big(x,y^*(x),\mathbf{z}^*(x)\big) - \mathcal{H}^*(x)\big\|^2$$

$$= \left\|\frac{1}{n}\sum_{i=1}^n \nabla f_i\big(x,y^*(x),z_i^*(x)\big) - \nabla\mathcal{L}_i\big(x,y^*(x),z_{\lambda,i}^*(x),z_i^*(x)\big)\right\|^2$$

$$\leq \frac{1}{n}\sum_{i=1}^n \big\|\nabla f_i\big(x,y^*(x),z_i^*(x)\big) - \nabla\mathcal{L}_i\big(x,y^*(x),z_{\lambda,i}^*(x),z_i^*(x)\big)\big\|^2. \qquad (17)$$

For any $i \in \{1,...,n\}$, we have

$$\big\|\nabla f_i\big(x,y^*(x),z_i^*(x)\big) - \nabla\mathcal{L}_i\big(x,y^*(x),z_{\lambda,i}^*(x),z_i^*(x)\big)\big\|^2$$

$$= \big\|\nabla_x f_i\big(x,y^*(x),z_i^*(x)\big) - \nabla_{xz}^2 g_i\big(x,z_i^*(x)\big)[\nabla_{zz}^2 g_i\big(x,z_i^*(x)\big)]^{-1}\nabla_z f_i\big(x,y^*(x),z_i^*(x)\big)$$

$$\quad - \nabla_x f_i\big(x,y^*(x),z_{\lambda,i}^*(x)\big) - \lambda\big(\nabla_x g_i\big(x,z_{\lambda,i}^*(x)\big) - \nabla_x g_i\big(x,z_i^*(x)\big)\big)\big\|^2$$

$$\leq 3\big\|\nabla_x f_i\big(x,y^*(x),z_i^*(x)\big) - \nabla_x f_i\big(x,y^*(x),z_{\lambda,i}^*(x)\big)\big\|^2$$

$$\quad + 3\big\|\nabla_{xz}^2 g_i\big(x,z_i^*(x)\big)[\nabla_{zz}^2 g_i\big(x,z_i^*(x)\big)]^{-1}\big(\nabla_z f_i\big(x,y^*(x),z_{\lambda,i}^*(x)\big) - \nabla_z f_i\big(x,y^*(x),z_i^*(x)\big)\big)\big\|^2$$

$$\quad + 3\big\| - \nabla_{xz}^2 g_i\big(x,z_i^*(x)\big)[\nabla_{zz}^2 g_i\big(x,z_i^*(x)\big)]^{-1}\nabla_z f_i\big(x,y^*(x),z_{\lambda,i}^*(x)\big)$$

$$\quad - \lambda\big(\nabla_x g_i\big(x,z_{\lambda,i}^*(x)\big) - \nabla_x g_i\big(x,z_i^*(x)\big)\big)\big\|^2$$

$$\leq 3\big\|\nabla_x f_i\big(x,y^*(x),z_i^*(x)\big) - \nabla_x f_i\big(x,y^*(x),z_{\lambda,i}^*(x)\big)\big\|^2$$

$$\quad + 3\big\|\nabla_{xz}^2 g_i\big(x,z_i^*(x)\big)[\nabla_{zz}^2 g_i\big(x,z_i^*(x)\big)]^{-1}\big(\nabla_z f_i\big(x,y^*(x),z_{\lambda,i}^*(x)\big) - \nabla_z f_i\big(x,y^*(x),z_i^*(x)\big)\big)\big\|^2$$

$$\quad + 3\big\|\lambda\nabla_{xz}^2 g_i\big(x,z_i^*(x)\big)[\nabla_{zz}^2 g_i\big(x,z_i^*(x)\big)]^{-1}\nabla_z g_i\big(x,z_{\lambda,i}^*(x)\big)$$

$$\quad - \lambda\big(\nabla_x g_i\big(x,z_{\lambda,i}^*(x)\big) - \nabla_x g_i\big(x,z_i^*(x)\big)\big)\big\|^2$$

$$\overset{(a)}{\leq} 3\Big(1 + \frac{L_{g,1}^2}{\mu_g^2}\Big)L_{f,1}^2\big\|z_{\lambda,i}^*(x) - z_i^*(x)\big\|^2$$

$$\quad + 6\lambda^2\big\|\nabla_{xz}^2 g_i\big(x,z_i^*(x)\big)[\nabla_{zz}^2 g_i\big(x,z_i^*(x)\big)]^{-1}$$

$$\qquad \cdot \big[\nabla_z g_i\big(x,z_{\lambda,i}^*(x)\big) - \nabla_z g_i\big(x,z_i^*(x)\big) - \nabla_{zz}^2 g_i\big(x,z_{\lambda,i}^*(x)\big)\big(z_{\lambda,i}^*(x) - z_i^*(x)\big)\big]\big\|^2$$

$$\quad + 6\lambda^2\big\|\nabla_{xz}^2 g_i\big(x,z_{\lambda,i}^*(x)\big)\big(z_{\lambda,i}^*(x) - z_i^*(x)\big) - \nabla_x g_i\big(x,z_{\lambda,i}^*(x)\big) - \nabla_x g_i\big(x,z_i^*(x)\big)\big\|^2$$

$$\overset{(b)}{\leq} 3\Big(1 + \frac{L_{f,1}^2}{\mu_g^2}\Big)L_{f,1}^2\big\|z_{\lambda,i}^*(x) - z_i^*(x)\big\|^2 + 6\lambda^2\Big(1 + \frac{L_{g,1}^2}{\mu_g^2}\Big)\Big(\frac{L_{g,1}}{2}\Big)^2\big\|z_{\lambda,i}^*(x) - z_i^*(x)\big\|^4$$

$$\overset{(c)}{\leq} \left[3\left(1+\frac{L_{f,1}^2}{\mu_g^2}\right)L_{f,1}^2\left(\frac{L_{f,0}}{\mu_g}\right)^2 + \left(1+\frac{L_{g,1}^2}{\mu_g^2}\right)\frac{3L_{g,1}^2}{2}\left(\frac{L_{f,0}}{\mu_g}\right)^4\right]\frac{1}{\lambda^2},\tag{18}$$

where (a) follows from Assumption 4.3, 4.4 and eq. (7); (b) follows from Assumption 4.3, 4.4 and Lemma 1 in [44]; (c) uses Lemma D.4. The proof is finished by substituting eq. (18) into eq. (17). $\square$

**Lemma D.6.** *Under Assumptions 4.3, 4.4, the gradient of Lagrangian function with optimal solutions* $\mathcal{H}^*(x)$ *is* $L_{*,1}$*-Lipschitz continuous in* $x$*, where we define* $L_{*,1} := \left(1+\frac{12L_{g,1}}{\mu_g}+\left(1+\frac{L_{g,1}}{\mu_g}\right)\frac{L_{f,1}}{\mu_f}\right)L_{f,1}+$
$\left(1+\frac{L_{g,1}}{\mu_g}\right)\frac{L_{f,0}L_{g,2}}{\mu_g}+L_{g,2}\left[\frac{1}{\mu_g}\left(\frac{L_{f,0}L_{g,2}}{\mu_g}+L_{f,1}\left(1+(1+\frac{L_{g,1}}{\mu_g})\frac{L_{f,1}}{\mu_f}\right)\right)+\frac{6L_{g,1}}{\mu_g^2}\left(1+(1+\frac{L_{g,1}}{\mu_g})\right)\left(L_{f,1}+\right.$
$\left.\frac{L_{f,0}L_{g,2}}{\mu_g}\right)\right]$ *and we assume* $\lambda \geq \left\{2L_{f,1}/\mu_g, (1+\frac{L_{g,1}}{\mu_g})\frac{L_{f,1}^2}{3\mu_f L_{g,1}}, (1+\frac{L_{g,1}}{\mu_g})\frac{L_{f,1}L_{f,2}}{3\mu_f L_{g,1}}, \frac{L_{f,1}L_{*,y}}{6L_{g,1}}\left(1+(1+\right.\right.$
$\left.\left.\frac{L_{g,1}}{\mu_g})\frac{L_{f,1}}{\mu_f}+\frac{12L_{g,1}}{\mu_g}\right)^{-1}, \left((1+\frac{L_{g,1}}{\mu_g})\frac{L_{f,1}}{\mu_f}+1\right)\frac{L_{f,1}}{L_{g,1}}\right\}.$

*Proof.* Recall that in eq. (6),

$$\mathcal{H}^*(x) = \frac{1}{n}\sum_{i=1}^n \left[\nabla_x f_i\big(x, y^*(x), z_{\lambda,i}^*(x)\big) + \lambda\Big(\nabla_x g_i\big(x, z_{\lambda,i}^*(x)\big) - \nabla_x g_i\big(x, z_i^*(x)\big)\Big)\right].$$

Then we have

$$\nabla\mathcal{H}^*(x) \overset{(a)}{=} \frac{1}{n}\sum_{i=1}^n \nabla_{xx}^2 f_i\big(x, y^*(x), z_{\lambda,i}^*(x)\big) + \big(\nabla y^*(x)\big)^T \nabla_{yx}^2 f_i\big(x, y^*(x), z_{\lambda,i}^*(x)\big)$$
$$+ \big(\nabla z_{\lambda,i}^*(x)\big)^T \nabla_{zx}^2 f_i\big(x, y^*(x), z_{\lambda,i}^*(x)\big) + \lambda\big(\nabla_{xx}^2 g_i(x, z_{\lambda,i}^*(x) - \nabla_{xx}^2 g_i(x, z_i^*(x))\big)$$
$$+ \lambda\Big(\big(\nabla z_{\lambda,i}^*(x)\big)^T \nabla_{zx}^2 g_i\big(x, z_{\lambda,i}^*(x)\big) - \big(\nabla z_i^*(x)\big)^T \nabla_{zx}^2 g_i\big(x, z_i^*(x)\big)\Big).\tag{19}$$

By taking norm, we have

$$\|\nabla\mathcal{H}^*(x)\| \leq \frac{1}{n}\sum_{i=1}^n \big\|\nabla_{xx}^2 f_i\big(x, y^*(x), z_{\lambda,i}^*(x)\big)\big\| + \frac{1}{n}\sum_{i=1}^n \big\|\nabla y^*(x)\big\|\big\|\nabla_{xy}^2 f_i\big(x, y^*(x), z_{\lambda,i}^*(x)\big)\big\|$$
$$+ \frac{1}{n}\sum_{i=1}^n \|\nabla z_{\lambda,i}^*(x)\|\big\|\nabla_{xz}^2 f_i\big(x, y^*(x), z_{\lambda,i}^*(x)\big)\big\|$$
$$+ \frac{\lambda}{n}\sum_{i=1}^n \Big[\big\|\nabla_{xx}^2 g_i\big(x, z_{\lambda,i}^*(x)\big) - \nabla_{xx}^2 g_i\big(x, z_i^*(x)\big)\big\|$$
$$+ \|\nabla z_i^*(x)\| \cdot \big\|\nabla_{xz}^2 g_i\big(x, z_{\lambda,i}^*(x)\big) - \nabla_{xz}^2 g_i\big(x, z_i^*(x)\big)\big\|$$
$$+ \|\nabla z_{\lambda,i}^*(x) - \nabla z_i^*(x)\| \cdot \big\|\nabla_{xz}^2 g_i\big(x, z_{\lambda,i}^*(x)\big)\big\|\Big]$$
$$\overset{(a)}{\leq} \left(1 + \frac{12L_{g,1}}{\mu_g} + \left(1 + \frac{L_{g,1}}{\mu_g}\right)\frac{L_{f,1}}{\mu_f}\right)L_{f,1} + \lambda\left(1 + \frac{L_{g,1}}{\mu_g}\right)L_{g,2}\|z_{\lambda,i}^*(x) - z_i^*(x)\|$$
$$+ \lambda L_{g,1}\|\nabla z_{\lambda,i}^*(x) - \nabla z_i^*(x)\|$$
$$\overset{(b)}{\leq} \left(1 + \frac{12L_{g,1}}{\mu_g} + \left(1 + \frac{L_{g,1}}{\mu_g}\right)\frac{L_{f,1}}{\mu_f}\right)L_{f,1} + \left(1 + \frac{L_{g,1}}{\mu_g}\right)\frac{L_{f,0}L_{g,2}}{\mu_g}$$
$$+ L_{g,2}\left[\frac{1}{\mu_g}\left(\frac{L_{f,0}L_{g,2}}{\mu_g} + L_{f,1}\left(1 + \left(1 + \frac{L_{g,1}}{\mu_g}\right)\frac{L_{f,1}}{\mu_f}\right)\right)\right.$$
$$\left.+ \frac{6L_{g,1}}{\mu_g^2}\left(1 + \left(1 + \frac{L_{g,1}}{\mu_g}\right)\right) \cdot \left(L_{f,1} + \frac{L_{f,0}L_{g,2}}{\mu_g}\right)\right],$$

where (a) uses Assumption 4.4 and Lemma D.2; (b) follows from Lemma D.4 Then, the proof is complete. $\square$

# E Proofs of Theorem 4.10 and and Corollary 4.11

## E.1 Descent in Objective Function

**Lemma E.1.** *Under Assumptions 4.3, 4.4, 4.5 and Lemma D.6, for $L_{*,1}$-smooth $\mathcal{L}^*(x)$, the consecutive iterates of Algorithm 1 satisfy:*

$$
\mathbb{E}\big[\mathcal{L}^*(x_{t+1}) - \mathcal{L}^*(x_t)\big]
$$

$$
\leq -\frac{\eta_x}{2}\mathbb{E}\|\mathcal{H}^*(x_t)\|^2 - \frac{\eta_x}{2}\mathbb{E}\|\widetilde{h}_x^t\|^2 + \frac{\eta_x^2 L_{*,1}}{2}\mathbb{E}\|h_x^t\|^2 + \frac{3\eta_x L_{f,1}^2}{2}\mathbb{E}\big\|y_t - y^*(x_t)\big\|^2
$$

$$
+ \frac{3\eta_x L_{\lambda,1}^2}{2}\mathbb{E}\left[\frac{1}{n}\sum_{i=1}^n \big\|z_{i,t} - z_{\lambda,i}^*(x_t)\big\|^2 + \frac{1}{n}\sum_{i=1}^n \big\|v_{i,t} - z_i^*(x_t)\big\|^2\right]
$$

*for all $t \in \{0, 1, ..., T-1\}$, where we assume $\lambda \geq \frac{2L_{f,1}}{\mu_g}$.*

*Proof.* Recall the definitions of $\mathcal{L}^*(x)$ and $\mathcal{H}^*(x)$ in eq. (6). By using the smoothness of $\mathcal{L}^*(x_t)$ in Lemma D.6, we have that

$$
\mathbb{E}\big[\mathcal{L}^*(x_{t+1})\big]
$$

$$
\leq \mathbb{E}\big[\mathcal{L}^*(x_t)\big] + \mathbb{E}\langle\mathcal{H}^*(x_t), x_{t+1} - x_t\rangle + \frac{L_{*,1}}{2}\mathbb{E}\|x_{t+1} - x_t\|^2
$$

$$
= \mathbb{E}\big[\mathcal{L}^*(x_t)\big] - \eta_x\mathbb{E}\langle\mathcal{H}^*(x_t), h_x^t\rangle + \frac{\eta_x^2 L_{*,1}}{2}\mathbb{E}\|h_x^t\|^2
$$

$$
= \mathbb{E}\big[\mathcal{L}^*(x_t)\big] - \eta_x\langle\mathcal{H}^*(x_t), \widetilde{h}_x^t\rangle + \frac{\eta_x^2 L_{*,1}}{2}\mathbb{E}\|h_x^t\|^2
$$

$$
= \mathbb{E}\big[\mathcal{L}^*(x_t)\big] - \frac{\eta_x}{2}\mathbb{E}\|\mathcal{H}^*(x_t)\|^2 - \frac{\eta_x}{2}\mathbb{E}\|\widetilde{h}_x^t\|^2 + \frac{\eta_x}{2}\mathbb{E}\|\mathcal{H}^*(x_t) - \widetilde{h}_x^t\|^2 + \frac{\eta_x^2 L_{*,1}}{2}\mathbb{E}\|h_x^t\|^2
$$

$$
\overset{(a)}{\leq} \mathbb{E}\big[\mathcal{L}^*(x_t)\big] - \frac{\eta_x}{2}\mathbb{E}\|\mathcal{H}^*(x_t)\|^2 - \frac{\eta_x}{2}\mathbb{E}\|\widetilde{h}_x^t\|^2 + \frac{\eta_x^2 L_{*,1}}{2}\mathbb{E}\|h_x^t\|^2 + \frac{3\eta_x L_{f,1}^2}{2}\mathbb{E}\big\|y_t - y^*(x_t)\big\|
$$

$$
+ \frac{3\eta_x L_{\lambda,1}^2}{2}\mathbb{E}\left[\frac{1}{n}\sum_{i=1}^n \big\|z_{i,t} - z_{\lambda,i}^*(x_t)\big\| + \frac{1}{n}\sum_{i=1}^n \big\|v_{i,t} - z_i^*(x_t)\big\|\right],
$$

where (a) follows from

$$
\mathbb{E}\|\mathcal{H}^*(x_t) - \widetilde{h}_x^t\|^2
$$

$$
= \mathbb{E}\left\|\frac{1}{n}\sum_{i=1}^n \nabla_x\mathcal{L}_i\big(x_t, y^*(x_t), z_{\lambda,i}^*(x_t), z_i^*(x_t)\big) - \nabla_x\mathcal{L}_i\big(x_t, y_t, z_{i,t}, v_{i,t}\big)\right\|^2
$$

$$
\overset{(a.1)}{\leq} \frac{1}{n}\sum_{i=1}^n \mathbb{E}\big\|\nabla_x\mathcal{L}_i\big(x_t, y^*(x_t), z_{\lambda,i}^*(x_t), z_i^*(x_t)\big) - \nabla_x\mathcal{L}_i\big(x_t, y_t, z_{i,t}, v_{i,t}\big)\big\|^2
$$

$$
\overset{(a.2)}{\leq} 3L_{f,1}^2\mathbb{E}\big\|y_t - y^*(x_t)\big\|^2 + 3L_{\lambda,1}^2\mathbb{E}\left[\frac{1}{n}\sum_{i=1}^n \big\|z_{i,t} - z_{\lambda,i}^*(x_t)\big\|^2 + \frac{1}{n}\sum_{i=1}^n \big\|v_{i,t} - z_i^*(x_t)\big\|^2\right],
$$

and (a.1) uses Jensen inequality; (a.2) follows from Assumption 4.4 and Lemma D.1. Then, the proof is complete. $\qquad\square$

## E.2 Bounds of Estimators

**Lemma E.2.** *Under Assumptions 4.3, 4.4, 4.5, 4.6, the estimators of $v_i$, $z_i$, $y$ and $x$ can be bounded as*

$$
\mathbb{E}\|h_{v,i}^t\|^2 \leq 2L_{g,1}^2\mathbb{E}\|v_{i,t} - z_i^*(x_t)\|^2 + 2\sigma_g^2,
$$

$$
\mathbb{E}\|h_{z,i}^t\|^2 \leq 4(L_{f,1}^2 + \lambda^2 L_{g,1}^2)\Big(\mathbb{E}\|y_t - y^*(x_t)\|^2 + \mathbb{E}\|z_{i,t} - z_{\lambda,i}^*(x_t)\|^2\Big) + 4(\sigma_f^2 + \lambda^2\sigma_g^2),
$$

$$\mathbb{E}\|h_y^t\|^2 \leq \frac{\sigma_f^2}{|I_t|} + \frac{(n-|I_t|)\sigma_{th}^2}{(n-1)|I_t|} + \left(1 + \frac{\beta_{th}^2}{|I_t|}\right) L_{f,1}^2 \left(\mathbb{E}\|y_t - y^*(x_t)\|^2 + \frac{1}{n}\sum_{i=1}^n \mathbb{E}\|v_{i,t} - z_i^*(x_t)\|^2\right),$$

$$\mathbb{E}\|h_x^t\|^2 \leq \mathbb{E}\|\widetilde{h}_x^t\|^2 + \frac{3(n-|I_t|)}{(n-1)|I_t|}(L_{f,0}^2 + 2\lambda^2 L_{g,0}^2) + \frac{3}{|I_t|}(\sigma_f^2 + 2\lambda^2\sigma_g^2).$$

*for any $i \in \{1,...,n\}$ and $t \in \{0,1,...,T-1\}$.*

*Proof.* By using the definition of $z_i^*(x_t)$, we can have that

$$\mathbb{E}\|h_{v,i}^t\|^2 \leq 2\mathbb{E}\|\nabla_z g_i(x_t, v_{i,t}; \xi_{i,t}^v) - \nabla_z g_i(x_t, v_{i,t})\|^2 + 2\mathbb{E}\|\nabla_z g_i(x_t, v_{i,t}) - \nabla_z g_i(x_t, z_i^*(x_t))\|^2$$
$$\leq 2L_{g,1}^2 \mathbb{E}\|v_{i,t} - z_i^*(x_t)\|^2 + 2\sigma_g^2.$$

Similarly, we have

$$\mathbb{E}\|h_{z,i}^t\|^2 \leq 2\mathbb{E}\|\nabla_z \mathcal{L}_i(x_t, y_t, z_{i,t}, v_{i,t}) - \nabla_z \mathcal{L}_i(x_t, y_t, z_{i,t}, v_{i,t}; \xi_{i,t}^z)\|^2$$
$$+ 2\mathbb{E}\|\nabla_z \mathcal{L}_i(x_t, y_t, z_{i,t}, v_{i,t}) - \nabla_z \mathcal{L}_i(x_t, y^*(x_t), z_{\lambda,i}^*(x_t), v_{i,t})\|^2$$
$$\leq 4(L_{f,1}^2 + \lambda^2 L_{g,1}^2)\left(\mathbb{E}\|z_{i,t} - z_{\lambda,i}^*(x_t)\|^2 + \mathbb{E}\|y_t - y^*(x_t)\|^2\right) + 4(\sigma_f^2 + \lambda^2\sigma_g^2).$$

Next, for the estimator of $x$, we have

$$\mathbb{E}\|h_x^t\|^2 = \mathbb{E}\left\|\frac{1}{|I_t|}\sum_{i\in I_t}\nabla_x\mathcal{L}_i(x_t, y_t, z_{i,t}, v_{i,t}; \xi_{i,t}^x)\right\|^2$$

$$\overset{(a)}{=} \mathbb{E}\left\|\frac{1}{|I_t|}\sum_{i\in I_t}\left[\nabla_x\mathcal{L}_i(x_t, y_t, z_{i,t}, v_{i,t}; \xi_{i,t}^x) - \nabla_x\mathcal{L}_i(x_t, y_t, z_{i,t}, v_{i,t})\right]\right\|^2$$

$$+ \mathbb{E}\left\|\frac{1}{|I_t|}\sum_{i\in I_t}\nabla_x\mathcal{L}_i(x_t, y_t, z_{i,t}, v_{i,t})\right\|^2. \tag{20}$$

where (a) uses unbiased estimation in Assumption 4.5. For the first part of eq. (20), since tasks are selected without replacement, we have

$$\mathbb{E}\left\|\frac{1}{|I_t|}\sum_{i\in I_t}\left[\nabla_x\mathcal{L}_i(x_t, y_t, z_{i,t}, v_{i,t}; \xi_{i,t}^x) - \nabla_x\mathcal{L}_i(x_t, y_t, z_{i,t}, v_{i,t})\right]\right\|^2$$

$$\overset{(a)}{=} \frac{1}{|I_t|^2}\sum_{i\in I_t}\mathbb{E}\left\|\nabla_x\mathcal{L}_i(x_t, y_t, z_{i,t}, v_{i,t}; \xi_{i,t}^x) - \nabla_x\mathcal{L}_i(x_t, y_t, z_{i,t}, v_{i,t})\right\|^2$$

$$\leq \frac{3}{|I_t|}(\sigma_f^2 + 2\lambda^2\sigma_g^2) \tag{21}$$

where (a) uses the unbiased estimation assumption in Assumption 4.5. For the second part of eq. (20), we have

$$\mathbb{E}\left\|\frac{1}{|I_t|}\sum_{i\in I_t}\nabla_x\mathcal{L}_i(x_t, y_t, z_{i,t}, v_{i,t})\right\|^2$$

$$\overset{(a)}{=} \frac{n(|I_t|-1)}{|I_t|(n-1)}\mathbb{E}\left\|\frac{1}{n}\sum_{i=1}^n\nabla_x\mathcal{L}_i(x_t, y_t, z_{i,t}, v_{i,t})\right\|^2 + \frac{n-|I_t|}{(n-1)|I_t|}\cdot\frac{1}{n}\sum_{i=1}^n\mathbb{E}\left\|\nabla_x\mathcal{L}_i(x_t, y_t, z_{i,t}, v_{i,t})\right\|^2$$

$$\overset{(b)}{\leq} \mathbb{E}\|\widetilde{h}_x^t\|^2 + \frac{3(n-|I_t|)}{(n-1)|I_t|}(L_{f,0}^2 + 2\lambda^2 L_{g,0}^2) \tag{22}$$

where (a) used the Lemma A.1 in [32]; (b) uses Assumption 4.4. By combining eq. (22) with eq. (21), the fourth inequality is proved. Last, for the estimator of $y$, we have

$$\mathbb{E}\|h_y^t\|^2 = \mathbb{E}\left\|\frac{1}{|I_t|}\sum_{i\in I_t}\nabla_y f_i(x_t, y_t, v_{i,t}; \xi_{i,t}^y)\right\|^2$$

$$\overset{(a)}{=}\mathbb{E}\left\|\frac{1}{|I_t|}\sum_{i\in I_t}\nabla_y f_i(x_t,y_t,v_{i,t};\xi_{i,t}^y)-\nabla_y f_i(x_t,y_t,v_{i,t})\right\|^2+\mathbb{E}\left\|\frac{1}{|I_t|}\sum_{i\in I_t}\nabla_y f_i(x_t,y_t,v_{i,t})\right\|^2$$

$$\overset{(b)}{\leq}\frac{1}{|I_t|^2}\sum_{i\in I_t}\mathbb{E}\left\|\nabla_y f_i(x_t,y_t,v_{i,t};\xi_{i,t}^y)-\nabla_y f_i(x_t,y_t,v_{i,t})\right\|^2$$

$$+\frac{n(|I_t|-1)}{|I_t|(n-1)}\mathbb{E}\left\|\frac{1}{n}\sum_{i=1}^n\nabla_y f_i(x_t,y_t,v_{i,t})\right\|^2+\frac{n-|I_t|}{(n-1)|I_t|}\cdot\frac{1}{n}\sum_{i=1}^n\mathbb{E}\left\|\nabla_y f_i(x_t,y_t,v_{i,t})\right\|^2$$

$$\overset{(c)}{\leq}\frac{\sigma_f^2}{|I_t|}+\frac{(n-|I_t|)\sigma_{th}^2}{(n-1)|I_t|}+\frac{n(|I_t|-1)+\beta_{th}^2(n-|I_t|)}{|I_t|(n-1)}\mathbb{E}\left\|\frac{1}{n}\sum_{i=1}^n\nabla_y f_i(x_t,y_t,v_{i,t})\right\|^2$$

$$\overset{(d)}{\leq}\frac{\sigma_f^2}{|I_t|}+\frac{(n-|I_t|)\sigma_{th}^2}{(n-1)|I_t|}$$

$$+\frac{n(|I_t|-1)+\beta_{th}^2(n-|I_t|)}{|I_t|(n-1)}\mathbb{E}\left\|\frac{1}{n}\sum_{i=1}^n\nabla_y f_i(x_t,y_t,v_{i,t})-\nabla_y f_i\big(x_t,y^*(x_t),z_i^*(x_t)\big)\right\|^2$$

$$\overset{(e)}{\leq}\frac{\sigma_f^2}{|I_t|}+\frac{(n-|I_t|)\sigma_{th}^2}{(n-1)|I_t|}$$

$$+\frac{n(|I_t|-1)+\beta_{th}^2(n-|I_t|)}{|I_t|(n-1)}L_{f,1}^2\left(\mathbb{E}\|y_t-y^*(x_t)\|^2+\frac{1}{n}\sum_{i=1}^n\mathbb{E}\|v_{i,t}-z_i^*(x_t)\|^2\right)$$

$$\overset{(f)}{\leq}\frac{\sigma_f^2}{|I_t|}+\frac{(n-|I_t|)\sigma_{th}^2}{(n-1)|I_t|}+\left(1+\frac{\beta_{th}^2}{|I_t|}\right)L_{f,1}^2\left(\mathbb{E}\|y_t-y^*(x_t)\|^2+\frac{1}{n}\sum_{i=1}^n\mathbb{E}\|v_{i,t}-z_i^*(x_t)\|^2\right),$$

where (a) uses Assumption 4.5; (b) follows from the Lemma A.1 in [32]; (c) uses Assumption 4.5, 4.6; (d) follows from definition $y^*(x)=\arg\max_y\frac{1}{n}\sum_i^n f_i\big(x,y,z_i^*(x)\big)$; (e) uses Assumption 4.4 and (f) uses $\beta_{th}\geq 1$. Then, the proof is complete. $\qquad\square$

### E.3 Descent in Approximation Errors

**Lemma E.3.** *Under Assumptions 4.3, 4.4, 4.5, 4.6, there exists $\delta_{v,1},\delta_{z,1},\delta_{y,1}$ such that the iterates of $v_{i,t}$, $z_{i,t}$ and $y_t$ in Algorithm 1 satisfy*

$$\frac{1}{n}\sum_{i=1}^n\left[\mathbb{E}\|v_{i,t+1}-z_i^*(x_{t+1})\|^2-\mathbb{E}\|v_{i,t}-z_i^*(x_t)\|^2\right]$$

$$\leq\left(-\eta_v\mu_g+\delta_v\right)\cdot\frac{1}{n}\sum_{i=1}^n\mathbb{E}\|v_{i,t}-z_i^*(x_t)\|^2+2\eta_v^2(1+\delta_v)\sigma_g^2$$

$$+\frac{\eta_x^2 L_{g,1}^2}{\delta_{v,1}\mu_g^2}\mathbb{E}\|\widetilde{h}_x^t\|^2+\eta_x^2\left(\frac{L_{g,1}^2}{\mu_g^2}+\frac{L_{*,z}}{2}\right)\mathbb{E}\|h_x^t\|^2,$$

$$\frac{1}{n}\sum_{i=1}^n\left[\mathbb{E}\|z_{i,t+1}-z_{\lambda,i}^*(x_{t+1})\|^2-\mathbb{E}\|z_{i,t}-z_{\lambda,i}^*(x_t)\|^2\right]$$

$$\leq\left(-\frac{\eta_z\lambda\mu_g}{4}+\delta_z\right)\frac{1}{n}\sum_{i=1}^n\mathbb{E}\|z_{i,t}-z_{\lambda,i}^*(x_t)\|^2+(1+\delta_z)\left(\frac{2\eta_z L_{f,1}^2}{\lambda\mu_g}+8\eta_z^2\lambda^2 L_{g,1}^2\right)\mathbb{E}\|y_t-y^*(x_t)\|^2$$

$$+\frac{144\eta_x^2 L_{g,1}^2}{\delta_{z,1}\mu_g^2}\mathbb{E}\|\widetilde{h}_x^t\|^2+\left(\frac{144\eta_x^2 L_{g,1}^2}{\mu_g^2}+\frac{\eta_x^2 L_{*,z_\lambda}}{2}\right)\mathbb{E}\|h_x^t\|^2+8(1+\delta_z)\eta_z^2\lambda^2\sigma_g^2,$$

$$\mathbb{E}\|y_{t+1}-y^*(x_{t+1})\|^2-\mathbb{E}\|y_t-y^*(x_t)\|^2$$

$$\leq\left[-\eta_y\mu_f+\eta_y^2(1+\delta_y)\left(1+\frac{\beta_{th}^2}{|I_t|}\right)L_{f,1}^2+\delta_y\right]\mathbb{E}\|y_t-y^*(x_t)\|^2$$

$$+\eta_y^2(1+\delta_y)\left(\frac{\sigma_f^2}{|I_t|}+\frac{(n-|I_t|)\sigma_{th}^2}{(n-1)|I_t|}\right)+\frac{\eta_x^2}{\delta_{y,1}}\left(1+\frac{L_{g,1}}{\mu_g}\right)^2\frac{L_{f,1}^2}{\mu_f^2}\mathbb{E}\|\widetilde{h}_x^t\|^2$$

$$+\left(\frac{\eta_y L_{f,1}^2}{\mu_f}+\eta_y^2\left(1+\frac{\beta_{th}^2}{|I_t|}\right)L_{f,1}^2\right)(1+\delta_y)\cdot\frac{1}{n}\sum_{i=1}^n\mathbb{E}\|v_{i,t}-z_i^*(x_t)\|^2$$

$$+ \eta_x^2 \left( \frac{L_{*,y}}{2} + \left(1 + \frac{L_{g,1}}{\mu_g}\right)^2 \frac{L_{f,1}^2}{\mu_f^2} \right) \mathbb{E} \|h_x^t\|^2,$$

*for all* $t \in \{0, ..., T-1\}$, *where we define* $\delta_v := \delta_{v,1} + \frac{3\eta_x^2 L_{*,z}}{2}(L_{f,0}^2 + 2\lambda^2 L_{g,0}^2)$, $\delta_z := \delta_{v,1} + \frac{3\eta_x^2 L_{*,z_\lambda}}{2}(L_{f,0}^2 + 2\lambda^2 L_{g,0}^2)$, $\delta_y := \delta_{y,1} + \frac{3\eta_x^2 L_{*,y}}{2}(L_{f,0}^2 + 2\lambda^2 L_{g,0}^2)$ *and we assume* $\lambda \geq \{2L_{f,1}/\mu_g, (1 + \frac{L_{g,1}}{\mu_g})\frac{L_{f,1}^2}{3\mu_f L_{g,1}}, (1 + \frac{L_{g,1}}{\mu_g})\frac{L_{f,1}L_{f,2}}{3\mu_f L_{g,1}}, \frac{L_{f,1}L_{*,y}}{6L_{g,1}}(1 + (1 + \frac{L_{g,1}}{\mu_g})\frac{L_{f,1}}{\mu_f} + \frac{12L_{g,1}}{\mu_g})^{-1}, ((1 + \frac{L_{g,1}}{\mu_g})\frac{L_{f,1}}{\mu_f} + 1)\frac{L_{f,1}}{L_{g,1}}\}$, $\eta_v \leq \frac{\mu_g}{2L_{g,1}^2}$, $\eta_z \leq \frac{\mu_g}{32L_{g,1}^2\lambda}$.

*Proof.* For the iterations of the lower-level problem, we have

$$\mathbb{E}\|v_{i,t+1} - z_i^*(x_{t+1})\|^2 = \mathbb{E}\|v_{i,t+1} - z_i^*(x_t)\|^2 + \mathbb{E}\|z_i^*(x_t) - z_i^*(x_{t+1})\|^2$$
$$+ 2\mathbb{E}\langle v_{i,t+1} - z_i^*(x_t), z_i^*(x_t) - z_i^*(x_{t+1})\rangle. \tag{23}$$

For the first term of eq. (23), we have

$$\mathbb{E}\|v_{i,t+1} - z_i^*(x_t)\|^2$$
$$= \mathbb{E}\|v_{i,t} - z_i^*(x_t) - \eta_v h_{v,i}^t\|^2$$
$$= \mathbb{E}\|v_{i,t} - z_i^*(x_t)\|^2 + \eta_v^2\mathbb{E}\|h_{v,i}^t\|^2 - 2\eta_v\mathbb{E}\langle v_{i,t} - z_i^*(x_t), h_{v,i}^t\rangle$$
$$= \mathbb{E}\|v_{i,t} - z_i^*(x_t)\|^2 + \eta_v^2\mathbb{E}\|h_{v,i}^t\|^2 - 2\eta_v\mathbb{E}\langle v_{i,t} - z_i^*(x_t), \nabla_z g_i(x_t, v_{i,t}) - \nabla_z g_i(x_t, z_i^*(x_t))\rangle$$
$$\overset{(a)}{\leq} (1 - 2\eta_v\mu_g)\mathbb{E}\|v_{i,t} - z_i^*(x_t)\|^2 + \eta_v^2\mathbb{E}\|h_{v,i}^t\|^2$$
$$\overset{(b)}{\leq} (1 - 2\eta_v\mu_g + 2\eta_v^2 L_{g,1}^2)\mathbb{E}\|v_{i,t} - z_i^*(x_t)\|^2 + 2\eta_v^2\sigma_g^2$$
$$\overset{(c)}{\leq} (1 - \eta_v\mu_g)\mathbb{E}\|v_{i,t} - z_i^*(x_t)\|^2 + 2\eta_v^2\sigma_g^2, \tag{24}$$

where (a) follows from Assumption 4.3; (b) uses Lemma E.2; (c) results from $\eta_v \leq \frac{\mu_g}{2L_{g,1}^2}$. For the second term of eq. (23), we have

$$\mathbb{E}\|z_i^*(x_t) - z_i^*(x_{t+1})\|^2 \leq \frac{L_{g,1}^2}{\mu_g^2}\mathbb{E}\|x_t - x_{t+1}\|^2 = \frac{\eta_x^2 L_{g,1}^2}{\mu_g^2}\mathbb{E}\|h_x^t\|^2. \tag{25}$$

For the last term of eq. (23), we have

$$2\mathbb{E}\langle v_{i,t+1} - z_i^*(x_t), z_i^*(x_t) - z_i^*(x_{t+1})\rangle$$
$$= -2\mathbb{E}\langle v_{i,t+1} - z_i^*(x_t), \nabla z_i^*(x_t)(x_{t+1} - x_t)\rangle$$
$$\quad - 2\mathbb{E}\langle v_{i,t+1} - z_i^*(x_t), z_i^*(x_{t+1}) - z_i^*(x_t) - \nabla z_i^*(x_t)(x_{t+1} - x_t)\rangle$$
$$= 2\mathbb{E}\langle v_{i,t+1} - z_i^*(x_t), \nabla z_i^*(x_t)\eta_x \widetilde{h}_x^t\rangle$$
$$\quad - 2\mathbb{E}\langle v_{i,t+1} - z_i^*(x_t), z_i^*(x_{t+1}) - z_i^*(x_t) - \nabla z_i^*(x_t)(x_{t+1} - x_t)\rangle$$
$$\leq 2\mathbb{E}\|v_{i,t+1} - z_i^*(x_t)\| \cdot \mathbb{E}\|\nabla z_i^*(x_t)\eta_x \widetilde{h}_x^t\|$$
$$\quad + 2\mathbb{E}\|v_{i,t+1} - z_i^*(x_t)\| \cdot \mathbb{E}\|z_i^*(x_{t+1}) - z_i^*(x_t) - \nabla z_i^*(x_t)(x_{t+1} - x_t)\|$$
$$\overset{(a)}{\leq} 2\mathbb{E}\|v_{i,t+1} - z_i^*(x_t)\| \cdot \mathbb{E}\|\nabla z_i^*(x_t)\eta_x \widetilde{h}_x^t\| + \mathbb{E}\|v_{i,t+1} - z_i^*(x_t)\| \cdot L_{*,z}\mathbb{E}\|x_{t+1} - x_t\|^2$$
$$\leq \delta_{v,1}\mathbb{E}\|v_{i,t+1} - z_i^*(x_t)\|^2 + \frac{\eta_x^2 L_{g,1}^2}{\delta_{v,1}\mu_g^2}\mathbb{E}\|\widetilde{h}_x^t\|^2 + \frac{\eta_x^2 L_{*,z}}{2}\mathbb{E}\|h_x^t\|^2$$
$$\quad + \frac{3\eta_x^2 L_{*,z}}{2}(L_{f,0}^2 + 2\lambda^2 L_{g,0}^2)\mathbb{E}\|v_{i,t+1} - z_i^*(x_t)\|^2$$
$$\overset{(b)}{=} \delta_v\mathbb{E}\|v_{i,t+1} - z_i^*(x_t)\|^2 + \frac{\eta_x^2 L_{g,1}^2}{\delta_{v,1}\mu_g^2}\mathbb{E}\|\widetilde{h}_x^t\|^2 + \frac{\eta_x^2 L_{*,z}}{2}\mathbb{E}\|h_x^t\|^2, \tag{26}$$

where (a) use Lemma D.3 and Lemma 1 in [44]; (b) defines $\delta_v := \delta_{v,1} + \frac{3\eta_x^2 L_{*,z}}{2}(L_{f,0}^2 + 2\lambda^2 L_{g,0}^2)$. By plugging eq. (24), eq. (25), eq. (26) into eq. (23), we have

$$\mathbb{E}\|v_{i,t+1} - z_i^*(x_{t+1})\|^2 - \mathbb{E}\|v_{i,t} - z_i^*(x_t)\|^2$$

$$\leq(-\eta_v\mu_g+\delta_v)\mathbb{E}\big\|v_{i,t}-z_i^*(x_t)\big\|^2+2\eta_v^2(1+\delta_v)\sigma_g^2+\frac{\eta_x^2L_{g,1}^2}{\delta_{v,1}\mu_g^2}\mathbb{E}\|\widetilde{h}_x^t\|^2+\eta_x^2\Big(\frac{L_{g,1}^2}{\mu_g^2}+\frac{L_{*,z}}{2}\Big)\mathbb{E}\|h_x^t\|^2.$$

Then we get

$$\frac{1}{n}\sum_{i=1}^n\Big[\mathbb{E}\big\|v_{i,t+1}-z_i^*(x_{t+1})\big\|^2-\mathbb{E}\big\|v_{i,t}-z_i^*(x_t)\big\|^2\Big]$$

$$\leq\big(-\eta_v\mu_g+\delta_v\big)\cdot\frac{1}{n}\sum_{i=1}^n\mathbb{E}\big\|v_{i,t}-z_i^*(x_t)\big\|^2+2\eta_v^2(1+\delta_v)\sigma_g^2$$

$$+\frac{\eta_x^2L_{g,1}^2}{\delta_{v,1}\mu_g^2}\mathbb{E}\|\widetilde{h}_x^t\|^2+\eta_x^2\Big(\frac{L_{g,1}^2}{\mu_g^2}+\frac{L_{*,z}}{2}\Big)\mathbb{E}\|h_x^t\|^2.$$

Thus, the first inequality in the lemma is proved. Similarly, we have

$$\mathbb{E}\|z_{i,t+1}-z_{\lambda,i}^*(x_{t+1})\|^2=\mathbb{E}\|z_{i,t+1}-z_{\lambda,i}^*(x_t)\|^2+\mathbb{E}\|z_{\lambda,i}^*(x_t)-z_{\lambda,i}^*(x_{t+1})\|^2$$
$$+2\mathbb{E}\big\langle z_{i,t+1}-z_{\lambda,i}^*(x_t),z_{\lambda,i}^*(x_t)-z_{\lambda,i}^*(x_{t+1})\big\rangle.\qquad(27)$$

We can bound the first term in eq. (27) similarly with eq. (24) as

$$\mathbb{E}\|z_{i,t+1}-z_{\lambda,i}^*(x_t)\|^2$$
$$\leq\mathbb{E}\|z_{i,t}-z_{\lambda,i}^*(x_t)\|^2+\eta_z^2\mathbb{E}\|h_{z,i}^t\|^2-2\eta_z\mathbb{E}\big\langle z_{i,t}-z_{\lambda,i}^*(x_t),h_{z,i}^t\big\rangle$$
$$=\mathbb{E}\|z_{i,t}-z_{\lambda,i}^*(x_t)\|^2+\eta_z^2\mathbb{E}\|h_{z,i}^t\|^2$$
$$-2\eta_z\mathbb{E}\big\langle z_{i,t}-z_{\lambda,i}^*(x_t),\nabla_z\mathcal{L}_i\big(x_t,y_t,z_{i,t},v_{i,t}\big)-\nabla_z\mathcal{L}_i\big(x_t,y_t,z_{\lambda,i}^*(x_t),v_{i,t}\big)\big\rangle$$
$$-2\eta_z\mathbb{E}\big\langle z_{i,t}-z_{\lambda,i}^*(x_t),\nabla_z\mathcal{L}_i\big(x_t,y_t,z_{\lambda,i}^*(x_t),v_{i,t}\big)-\nabla_z\mathcal{L}_i\big(x_t,y^*(x_t),z_{\lambda,i}^*(x_t),v_{i,t}\big)\big\rangle$$
$$=\mathbb{E}\|z_{i,t}-z_{\lambda,i}^*(x_t)\|^2+\eta_z^2\mathbb{E}\|h_{z,i}^t\|^2$$
$$-2\eta_z\mathbb{E}\big\langle z_{i,t}-z_{\lambda,i}^*(x_t),\nabla_z\mathcal{L}_i\big(x_t,y_t,z_{i,t},v_{i,t}\big)-\nabla_z\mathcal{L}_i\big(x_t,y_t,z_{\lambda,i}^*(x_t),v_{i,t}\big)\big\rangle$$
$$-2\eta_z\mathbb{E}\big\langle z_{i,t}-z_{\lambda,i}^*(x_t),\nabla_zf_i\big(x_t,y_t,z_{\lambda,i}^*(x_t)\big)-\nabla_zf_i\big(x_t,y^*(x_t),z_{\lambda,i}^*(x_t)\big)\big\rangle$$
$$\leq(1-\eta_z\lambda\mu_g)\mathbb{E}\|z_{i,t}-z_{\lambda,i}^*(x_t)\|^2+\eta_z^2\mathbb{E}\|h_{z,i}^t\|^2$$
$$+\frac{\eta_z\lambda\mu_g}{2}\mathbb{E}\|z_{i,t}-z_{\lambda,i}^*(x_t)\|^2+\frac{2\eta_z}{\lambda\mu_g}\mathbb{E}\big\|\nabla_zf_i\big(x_t,y_t,z_{\lambda,i}^*(x_t)\big)-\nabla_zf_i\big(x_t,y^*(x_t),z_{\lambda,i}^*(x_t)\big)\big\|^2$$
$$\leq\Big(1-\frac{\eta_z\lambda\mu_g}{2}\Big)\mathbb{E}\|z_{i,t}-z_{\lambda,i}^*(x_t)\|^2+\eta_z^2\mathbb{E}\|h_{z,i}^t\|^2+\frac{2\eta_zL_{f,1}^2}{\lambda\mu_g}\mathbb{E}\|y_t-y^*(x_t)\|^2$$
$$\overset{(a)}{\leq}\Big(1-\frac{\eta_z\lambda\mu_g}{2}\Big)\mathbb{E}\|z_{i,t}-z_{\lambda,i}^*(x_t)\|^2+\Big(\frac{2\eta_zL_{f,1}^2}{\lambda\mu_g}+8\eta_z^2\lambda^2L_{g,1}^2\Big)\mathbb{E}\|y_t-y^*(x_t)\|^2$$
$$+8\eta_z^2\lambda^2L_{g,1}^2\mathbb{E}\|z_{i,t}-z_{\lambda,i}^*(x_t)\|^2+8\eta_z^2\lambda^2\sigma_g^2$$
$$\overset{(b)}{\leq}\Big(1-\frac{\eta_z\lambda\mu_g}{4}\Big)\mathbb{E}\|z_{i,t}-z_{\lambda,i}^*(x_t)\|^2+\Big(\frac{2\eta_zL_{f,1}^2}{\lambda\mu_g}+8\eta_z^2\lambda^2L_{g,1}^2\Big)\mathbb{E}\|y_t-y^*(x_t)\|^2+8\eta_z^2\lambda^2\sigma_g^2,\qquad(28)$$

where (a) uses Lemma E.2 and $\lambda\geq\max\big\{\frac{L_{f,1}}{L_{g,1}},\frac{\sigma_f}{\sigma_g}\big\}$; (b) uses $\eta_z\lambda\leq\frac{\mu_g}{32L_{g,1}^2}$; we bound the second term in eq. (27) as

$$\mathbb{E}\|z_{\lambda,i}^*(x_t)-z_{\lambda,i}^*(x_{t+1})\|^2\leq\frac{144L_{g,1}^2}{\mu_g^2}\mathbb{E}\|x_t-x_{t+1}\|^2=\frac{144\eta_x^2L_{g,1}^2}{\mu_g^2}\mathbb{E}\|h_x^t\|^2;\qquad(29)$$

and we bound the last term in eq. (27) similarly with eq. (26) as

$$2\mathbb{E}\big\langle z_{i,t+1}-z_{\lambda,i}^*(x_t),z_{\lambda,i}^*(x_t)-z_{\lambda,i}^*(x_{t+1})\big\rangle$$
$$\leq\delta_z\mathbb{E}\big\|z_{i,t+1}-z_{\lambda,i}^*(x_t)\big\|^2+\frac{144\eta_x^2L_{g,1}^2}{\delta_{z,1}\mu_g^2}\mathbb{E}\big\|\widetilde{h}_x^t\big\|^2+\frac{\eta_x^2L_{*,z_\lambda}}{2}\mathbb{E}\|h_x^t\|^2,\qquad(30)$$

where $\delta_z:=\delta_{z,1}+\frac{3\eta_x^2L_{*,z_\lambda}}{2}(L_{f,0}^2+2\lambda^2L_{g,0}^2)$. By plugging eq. (28), eq. (29), eq. (30) into eq. (27), we have

$$\mathbb{E}\big\|z_{i,t+1}-z_{\lambda,i}^*(x_{t+1})\big\|^2-\mathbb{E}\big\|z_{i,t}-z_{\lambda,i}^*(x_t)\big\|^2$$

$$\leq \Big( -\frac{\eta_z \lambda \mu_g}{4} + \delta_z \Big) \mathbb{E}\big\|z_{i,t} - z^*_{\lambda,i}(x_t)\big\|^2 + (1+\delta_z)\Big( \frac{2\eta_z L^2_{f,1}}{\lambda \mu_g} + 8\eta_z^2 \lambda^2 L^2_{g,1} \Big) \mathbb{E}\|y_t - y^*(x_t)\|^2$$

$$+ \frac{144\eta_x^2 L^2_{g,1}}{\delta_{z,1}\mu_g^2} \mathbb{E}\big\|\widetilde{h}_x^t\big\|^2 + \Big( \frac{144\eta_x^2 L^2_{g,1}}{\mu_g^2} + \frac{\eta_x^2 L_{*,z_\lambda}}{2} \Big) \mathbb{E}\|h_x^t\|^2 + 8(1+\delta_z)\eta_z^2\lambda^2\sigma_g^2,$$

After telescoping, we obtain

$$\frac{1}{n}\sum_{i=1}^n \Big[ \mathbb{E}\big\|z_{i,t+1} - z^*_{\lambda,i}(x_{t+1})\big\|^2 - \mathbb{E}\big\|z_{i,t} - z^*_{\lambda,i}(x_t)\big\|^2 \Big]$$

$$\leq \Big( -\frac{\eta_z \lambda \mu_g}{4} + \delta_z \Big) \frac{1}{n}\sum_{i=1}^n \mathbb{E}\big\|z_{i,t} - z^*_{\lambda,i}(x_t)\big\|^2 + (1+\delta_z)\Big( \frac{2\eta_z L^2_{f,1}}{\lambda \mu_g} + 8\eta_z^2 \lambda^2 L^2_{g,1} \Big) \mathbb{E}\|y_t - y^*(x_t)\|^2$$

$$+ \frac{144\eta_x^2 L^2_{g,1}}{\delta_{z,1}\mu_g^2} \mathbb{E}\big\|\widetilde{h}_x^t\big\|^2 + \Big( \frac{144\eta_x^2 L^2_{g,1}}{\mu_g^2} + \frac{\eta_x^2 L_{*,z_\lambda}}{2} \Big) \mathbb{E}\|h_x^t\|^2 + 8(1+\delta_z)\eta_z^2\lambda^2\sigma_g^2.$$

Then the second inequality in the lemma is proved. Last, for $y_t$ and $y^*(x_t)$, we have

$$\mathbb{E}\|y_{t+1} - y^*(x_{t+1})\|^2 = \mathbb{E}\|y_{t+1} - y^*(x_t)\|^2 + \mathbb{E}\|y^*(x_t) - y^*(x_{t+1})\|^2$$
$$+ 2\mathbb{E}\big\langle y_{t+1} - y^*(x_t), y^*(x_t) - y^*(x_{t+1}) \big\rangle. \tag{31}$$

We can bound the first term in eq. (31) as

$$\mathbb{E}\|y_{t+1} - y^*(x_t)\|^2$$
$$= \mathbb{E}\|y_t + h_y^t - y^*(x_t)\|^2$$
$$= \mathbb{E}\|y_t - y^*(x_t)\|^2 + \eta_y^2 \mathbb{E}\|h_y^t\|^2 + 2\eta_y \mathbb{E}\big\langle y_t - y^*(x_t), h_y^t \big\rangle$$
$$\overset{(a)}{=} \mathbb{E}\|y_t - y^*(x_t)\|^2 + \eta_y^2 \mathbb{E}\|h_y^t\|^2$$
$$+ 2\eta_y \mathbb{E}\Big\langle y_t - y^*(x_t), \frac{1}{n}\sum_{i=1}^n \nabla_y f_i\big(x_t, y_t, v_{i,t}\big) - \nabla_y f_i\big(x_t, y_t, z_i^*(x_t)\big) \Big\rangle$$
$$+ 2\eta_y \mathbb{E}\Big\langle y_t - y^*(x_t), \frac{1}{n}\sum_{i=1}^n \nabla_y f_i\big(x_t, y_t, z_i^*(x_t)\big) - \nabla_y f_i\big(x_t, y^*(x_t), z_i^*(x_t)\big) \Big\rangle$$
$$\overset{(b)}{\leq} \mathbb{E}\|y_t - y^*(x_t)\|^2 + \eta_y^2 \mathbb{E}\|h_y^t\|^2$$
$$+ \eta_y \Big( \mu_f \mathbb{E}\|y_t - y^*(x_t)\|^2 + \frac{1}{\mu_f} \cdot \mathbb{E}\Big\| \frac{1}{n}\sum_{i=1}^n \nabla_y f_i\big(x_t, y_t, v_{i,t}\big) - \nabla_y f_i\big(x_t, y_t, z_i^*(x_t)\big) \Big\|^2 \Big)$$
$$- 2\eta_y \mu_f \mathbb{E}\|y_t - y^*(x_t)\|^2$$
$$\overset{(c)}{\leq} (1 - \eta_y \mu_f) \mathbb{E}\|y_t - y^*(x_t)\|^2 + \eta_y^2 \mathbb{E}\|h_y^t\|^2 + \frac{\eta_y L^2_{f,1}}{\mu_f} \cdot \frac{1}{n}\sum_{i=1}^n \mathbb{E}\|v_{i,t} - z_i^*(x_t)\|^2$$
$$\overset{(d)}{\leq} \Big( 1 - \frac{\eta_y \mu_f}{2} + \eta_y^2 \Big(1 + \frac{\beta^2_{th}}{|I_t|}\Big) L^2_{f,1} \Big) \mathbb{E}\|y_t - y^*(x_t)\|^2 + \eta_y^2 \Big( \frac{\sigma_f^2}{|I_t|} + \frac{(n - |I_t|)\sigma^2_{th}}{(n-1)|I_t|} \Big)$$
$$+ \Big( \frac{\eta_y L^2_{f,1}}{\mu_f} + \eta_y^2 \Big(1 + \frac{\beta^2_{th}}{|I_t|}\Big) L^2_{f,1} \Big) \cdot \frac{1}{n}\sum_{i=1}^n \mathbb{E}\|v_{i,t} - z_i^*(x_t)\|^2, \tag{32}$$

where (a) uses the definition of $y^*(x_t)$ and eq. (5); (b) uses strong concavity of $f_i$ in $y$; (c) follows from definition of $y^*(x_t)$ and Assumption 4.4; (d) uses Lemma E.2. We can bound the second term in eq. (31) as

$$\mathbb{E}\|y^*(x_t) - y^*(x_{t+1})\|^2 \overset{(a)}{\leq} \Big(1 + \frac{L_{g,1}}{\mu_g}\Big)^2 \frac{L^2_{f,1}}{\mu_f^2} \mathbb{E}\|x_t - x_{t+1}\|^2 = \eta_x^2 \Big(1 + \frac{L_{g,1}}{\mu_g}\Big)^2 \frac{L^2_{f,1}}{\mu_f^2} \mathbb{E}\|h_x^t\|^2, \tag{33}$$

where (a) follows from Lemma D.2. Also, we can get the bound of the last term as

$$2\mathbb{E}\big\langle y_{t+1} - y^*(x_t), y^*(x_t) - y^*(x_{t+1}) \big\rangle$$

$$
\begin{aligned}
= & - 2\mathbb{E}\langle y_{t+1} - y^*(x_t), \nabla y^*(x_t)(x_{t+1} - x_t)\rangle \\
& - 2\mathbb{E}\langle y_{t+1} - y^*(x_t), y^*(x_{t+1}) - y^*(x_t) - \nabla y^*(x_t)(x_{t+1} - x_t)\rangle \\
= & 2\mathbb{E}\big\| y_{t+1} - y^*(x_t)\big\| \cdot \big\| \eta_x \nabla y^*(x_t)\widetilde{h}_x^t\big\| \\
& + 2\mathbb{E}\big\| y_{t+1} - y^*(x_t)\big\| \cdot \big\| y^*(x_{t+1}) - y^*(x_t) - \nabla y^*(x_t)(x_{t+1} - x_t)\big\| \\
\overset{(a)}{=} & \delta_{y,1}\mathbb{E}\big\| y_{t+1} - y^*(x_t)\big\|^2 + \frac{\eta_x^2}{\delta_{y,1}}\Big(1 + \frac{L_{g,1}}{\mu_g}\Big)^2 \frac{L_{f,1}^2}{\mu_f^2}\mathbb{E}\big\|\widetilde{h}_x^t\big\|^2 \\
& + \mathbb{E}\big\| y_{t+1} - y^*(x_t)\big\| \cdot L_{*,y}\big\| x_{t+1} - x_t\big\|^2 \\
\overset{(b)}{\leq} & \delta_{y,1}\mathbb{E}\big\| y_{t+1} - y^*(x_t)\big\|^2 + \frac{\eta_x^2}{\delta_{y,1}}\Big(1 + \frac{L_{g,1}}{\mu_g}\Big)^2 \frac{L_{f,1}^2}{\mu_f^2}\mathbb{E}\big\|\widetilde{h}_x^t\big\|^2 \\
& + \frac{L_{*,y}}{2}\mathbb{E}\big\| y_{t+1} - y^*(x_t)\big\|^2 \cdot \big\| x_{t+1} - x_t\big\|^2 + \frac{L_{*,y}}{2}\mathbb{E}\big\| x_{t+1} - x_t\big\|^2 \\
\overset{(c)}{\leq} & \Big(\delta_{y,1} + \frac{3\eta_x^2 L_{*,y}}{2}(L_{f,0}^2 + 2\lambda^2 L_{g,0}^2)\Big)\mathbb{E}\big\| y_{t+1} - y^*(x_t)\big\|^2 \\
& + \frac{\eta_x^2}{\delta_{y,1}}\Big(1 + \frac{L_{g,1}}{\mu_g}\Big)^2 \frac{L_{f,1}^2}{\mu_f^2}\mathbb{E}\big\|\widetilde{h}_x^t\big\|^2 + \frac{\eta_x^2 L_{*,y}}{2}\mathbb{E}\big\| h_x^t\big\|^2 \\
\overset{(d)}{\leq} & \delta_y \mathbb{E}\big\| y_{t+1} - y^*(x_t)\big\|^2 + \frac{\eta_x^2}{\delta_{y,1}}\Big(1 + \frac{L_{g,1}}{\mu_g}\Big)^2 \frac{L_{f,1}^2}{\mu_f^2}\mathbb{E}\big\|\widetilde{h}_x^t\big\|^2 + \frac{\eta_x^2 L_{*,y}}{2}\mathbb{E}\big\| h_x^t\big\|^2, \quad (34)
\end{aligned}
$$

where (a) uses Lemma D.2 and Lemma D.3; (b) use Lemma D.3 and Lemma 1 in [44]; (c) follows from Assumption 4.4; (d) defines $\delta_y = \delta_{y,1} + \frac{3\eta_x^2 L_{*,y}}{2}(L_{f,0}^2 + 2\lambda^2 L_{g,0}^2)$. By plugging eq. (32), eq. (33), eq. (34) into eq. (34), we get

$$
\begin{aligned}
& \mathbb{E}\| y_{t+1} - y^*(x_{t+1})\|^2 - \mathbb{E}\| y_t - y^*(x_t)\|^2 \\
& \leq \Big[ -\eta_y \mu_f + \eta_y^2(1 + \delta_y)\Big(1 + \frac{\beta_{th}^2}{|I_t|}\Big)L_{f,1}^2 + \delta_y\Big]\mathbb{E}\| y_t - y^*(x_t)\|^2 \\
& \quad + \eta_y^2(1 + \delta_y)\Big(\frac{\sigma_f^2}{|I_t|} + \frac{(n - |I_t|)\sigma_{th}^2}{(n-1)|I_t|}\Big) + \frac{\eta_x^2}{\delta_{y,1}}\Big(1 + \frac{L_{g,1}}{\mu_g}\Big)^2 \frac{L_{f,1}^2}{\mu_f^2}\mathbb{E}\big\|\widetilde{h}_x^t\big\|^2 \\
& \quad + \Big(\frac{\eta_y L_{f,1}^2}{\mu_f} + \eta_y^2\Big(1 + \frac{\beta_{th}^2}{|I_t|}\Big)L_{f,1}^2\Big)(1 + \delta_y) \cdot \frac{1}{n}\sum_{i=1}^n \mathbb{E}\| v_{i,t} - z_i^*(x_t)\|^2 \\
& \quad + \eta_x^2\Big(\frac{L_{*,y}}{2} + \Big(1 + \frac{L_{g,1}}{\mu_g}\Big)^2 \frac{L_{f,1}^2}{\mu_f^2}\Big)\mathbb{E}\| h_x^t\|^2.
\end{aligned}
$$

Then the last inequality is proved. Thus, the proof is complete. $\qquad\square$

## E.4 Descent in the Lyapunov Function and Proof of Theorem 4.10

We define the Lyapunov function as

$$
\begin{aligned}
\Psi_t := & \mathcal{L}^*(x_t) + K_y \mathbb{E}\| y_t - y^*(x_t)\|^2 + K_z \frac{1}{n}\sum_{i=1}^n \mathbb{E}\| z_{i,t} - z_{\lambda,i}^*(x_t)\|^2 \\
& + K_v \frac{1}{n}\sum_{i=1}^n \mathbb{E}\| v_{i,t} - z_i^*(x_t)\|^2, \quad (35)
\end{aligned}
$$

where the coefficients are given by

$$
K_y = \frac{\eta_x}{\eta_y} \cdot \frac{2}{\mu_f}\Big(\frac{3L_{f,1}^2}{2} + \frac{216 L_{g,1}^2 L_{f,1}^2}{\mu_g^2} + \frac{864 L_{g,1}^2}{\mu_g}\Big), \quad K_z = \frac{\eta_x \lambda^2}{\eta_z \lambda} \cdot \frac{54 L_{g,1}^2}{\mu_g}, \quad K_v = \frac{\eta_x \lambda^2}{\eta_v} \cdot \frac{54 L_{g,1}^2}{\mu_g};
$$

$$
\delta_{y,1} = \frac{\eta_y \mu_f}{8}, \quad \delta_{z,1} = \frac{\eta_z \lambda \mu_g}{8}, \quad \delta_{v,1} = \frac{\eta_v \mu_g}{4}. \quad (36)
$$

For convenience, we define the following constants:

$$C_1 := \max\left\{ \frac{L_{*,1}}{2}, \frac{54L_{g,1}^2}{\mu_g}\left(\frac{L_{g,1}^2}{\mu_g^2} + \frac{L_{*,z}}{2}\right), \frac{54L_{g,1}^2}{\mu_g}\left(\frac{144L_{g,1}^2}{\mu_g^2} + \frac{L_{*,z_\lambda}}{2}\right), \right.$$
$$\left. \frac{2}{\mu_f}\left(\frac{3L_{f,1}^2}{2} + \frac{216L_{g,1}^2 L_{f,1}^2}{\mu_g^2} + \frac{864L_{g,1}^2}{\mu_g}\right)\left(\frac{L_{*,y}}{2} + \left(1 + \frac{L_{g,1}}{\mu_g}\right)^2 \frac{L_{f,1}^2}{\mu_f}\right)\right\},$$

$$C_2 := \max\left\{ \frac{62208L_{g,1}^4}{\mu_g^4}, \frac{16}{\mu_f^2}\left(\frac{3L_{f,1}^2}{2} + \frac{216L_{g,1}^2 L_{f,1}^2}{\mu_g^2} + \frac{864L_{g,1}^2}{\mu_g}\right)\left(1 + \frac{L_{g,1}}{\mu_g}\right)^2 \frac{L_{f,1}^2}{\mu_f^2}\right\}$$

$$C_3 := \max\left\{ \frac{864L_{g,1}^2 \sigma_g^2}{\mu_g}, \frac{4}{\mu_f}\left(\frac{3L_{f,1}^2}{2} + \frac{216L_{g,1}^2 L_{f,1}^2}{\mu_g^2} + \frac{864L_{g,1}^2}{\mu_g}\right)(\sigma_f^2 + \sigma_{th}^2)\right\}. \tag{37}$$

We also constrain the conditions as below:

$$\lambda \geq \left\{ \frac{2L_{f,1}}{\mu_g}, \left(1 + \frac{L_{g,1}}{\mu_g}\right)\frac{L_{f,1}^2}{3\mu_f L_{g,1}}, \left(1 + \frac{L_{g,1}}{\mu_g}\right)\frac{L_{f,1}L_{f,2}}{3\mu_f L_{g,1}}, \frac{L_{f,1}L_{*,y}}{6L_{g,1}}\left(1 + \left(1 + \frac{L_{g,1}}{\mu_g}\right)\frac{L_{f,1}}{\mu_f} + \frac{12L_{g,1}}{\mu_g}\right)^{-1}, \right.$$
$$\left. \left(\left(1 + \frac{L_{g,1}}{\mu_g}\right)\frac{L_{f,1}}{\mu_f} + 1\right)\frac{L_{f,1}}{L_{g,1}}, \frac{L_{f,2}}{L_{g,2}}, \frac{L_{f,0}}{L_{g,0}}, \frac{\sigma_f}{\sigma_g}, \frac{16L_{f,1}^2}{27\mu_f^2 L_{g,1}^2}\left(\frac{3L_{f,1}^2}{2} + \frac{216L_{g,1}^2 L_{f,1}^2}{\mu_g^2} + \frac{864L_{g,1}^2}{\mu_g}\right)\right\},$$

$$\eta_x \leq \frac{1}{16C_1}, \quad \eta_y \leq \min\left\{ \frac{\mu_f}{8(1+\beta_{th}^2)L_{f,1}^2}, \frac{1}{(1+\beta_{th}^2)\mu_f}\right\}, \quad \eta_z\lambda \leq \min\left\{ \frac{\mu_g}{64L_{g,1}^2}, \frac{4}{\mu_g}\right\},$$

$$\eta_v \leq \min\left\{ \frac{\mu_g}{8L_{g,1}^2}, \frac{2}{\mu_g}\right\}, \quad \eta_z\lambda^3 \leq \frac{\mu_g}{L_{g,1}^2}, \quad \frac{\eta_x^2}{\eta_y^2} \leq \frac{1}{12C_2}, \quad \frac{\eta_x^2}{\eta_z^2} \leq \frac{1}{12C_2}, \quad \frac{\eta_x^2\lambda^2}{\eta_v^2} \leq \frac{1}{12C_2},$$

$$\frac{\eta_x^2\lambda^2}{\eta_y} \leq \min\left\{ \frac{1}{16C_1}, \frac{\mu_f}{36L_{*,y}L_{g,0}^2}\right\}, \quad \frac{\eta_x^2\lambda}{\eta_z} \leq \min\left\{ \frac{1}{16C_1}, \frac{\mu_f}{36L_{*,z_\lambda}L_{g,0}^2}\right\},$$

$$\frac{\eta_x^2}{\eta_v} \leq \min\left\{ \frac{1}{16C_1}, \frac{\mu_f}{36L_{*,z}L_{g,0}^2}\right\}. \tag{38}$$

Plugging Lemma E.1, Lemma E.3 into eq. (35) and using eq. (36), we have **the descent in the Lyapunov function** as

$$\Psi_{t+1} - \Psi_t \leq -\frac{\eta_x}{2}\mathbb{E}\|\mathcal{H}^*(x_t)\|^2 + \frac{\eta_x^2\lambda^2}{|I_t|}\left(1 + \frac{\eta_x}{\eta_y} + \frac{\eta_x\lambda^2}{(\eta_z\lambda)} + \frac{\eta_x\lambda^2}{\eta_v}\right)C_1 \cdot 9(L_{g,0}^2 + \sigma_g^2)$$
$$+ \left(\eta_x\eta_y + \eta_x(\eta_z\lambda)\lambda^2 + \eta_x\eta_v\lambda^2\right)C_3. \tag{39}$$

*Proof.* To simplify the problem, we assume $|I_t| = P$ for $t = 0, ..., T-1$. By taking summation of eq. (39) from $t = 0$ to $T-1$, we get

$$\frac{1}{T}\sum_{t=0}^{T-1}\frac{\eta_x}{2}\mathbb{E}\|\mathcal{H}^*(x_t)\|^2 \leq \frac{1}{T}(\Psi_0 - \Psi_T) + \frac{\eta_x^2\lambda^2}{P}\left(1 + \frac{\eta_x}{\eta_y} + \frac{\eta_x\lambda^2}{(\eta_z\lambda)} + \frac{\eta_x\lambda^2}{\eta_v}\right)C_2$$
$$+ \left(\eta_x\eta_y + \eta_x(\eta_z\lambda)\lambda^2 + \eta_x\eta_v\lambda^2\right)C_3 \tag{40}$$

By using Lemma D.5 and eq. (40), we have

$$\frac{1}{T}\sum_{t=0}^{T-1}\mathbb{E}\|\nabla\Phi(x_t)\|^2 \leq \frac{2}{T}\sum_{t=0}^{T-1}\mathbb{E}\|\nabla\Phi(x_t) - \mathcal{H}^*(x_t)\|^2 + \frac{2}{T}\sum_{t=0}^{T-1}\mathbb{E}\|\mathcal{H}^*(x_t)\|^2$$
$$\overset{(a)}{\leq} \frac{2C_{gap}}{\lambda^2} + \frac{4(\Psi_0 - \Psi_T)}{T\eta_x} + \frac{4\eta_x\lambda^2}{P}\left(1 + \frac{\eta_x}{\eta_y} + \frac{\eta_x\lambda^2}{(\eta_z\lambda)} + \frac{\eta_x\lambda^2}{\eta_v}\right)C_2$$
$$+ 4\left(\eta_y + (\eta_z\lambda)\lambda^2 + \eta_v\lambda^2\right)C_3$$
$$\overset{(b)}{\leq} \mathcal{O}(T^{-\frac{2}{7}}), \tag{41}$$

where (a) uses eq. (40); (b) takes $\eta_x = \mathcal{O}(T^{-\frac{5}{7}})$, $\eta_y = \mathcal{O}(T^{-\frac{2}{7}})$, $\eta_z = \mathcal{O}(T^{-\frac{5}{7}})$, $\eta_v = \mathcal{O}(T^{-\frac{4}{7}})$, $\lambda = \mathcal{O}(T^{\frac{1}{7}})$, which satisfies eq. (38). Thus, Theorem 4.10 is proved. $\square$

## E.5 Proof of Corollary 4.11

*Proof.* From eq. (41), to achieve $\epsilon$-accurate stationary point of the objective function $\Phi(x)$ in definition 4.2, we let $\frac{1}{T}\sum_{t=0}^{T-1}\mathbb{E}\|\nabla\Phi(x_t)\|^2 = \mathcal{O}(T^{-\frac{2}{7}}) \leq \epsilon$. As a result, we can see that the epochs number we need is $T = \mathcal{O}(\epsilon^{-\frac{7}{2}})$. The total sample complexity is $PT = \mathcal{O}(P\epsilon^{-\frac{7}{2}})$. Then, we finish the proof. $\square$

# F  Proofs of Theorem 4.12 and Corollary 4.13

## F.1  Descent in Objective Function

**Lemma F.1.** *Under Assumptions 4.3, 4.4, 4.5 and Lemma D.6, for $L_{*,1}$-smooth $\mathcal{L}^*(x)$, the consecutive iterates of Algorithm 2 satisfy:*

$$\mathbb{E}\big[\mathcal{L}^*(x_{t+1}) - \mathcal{L}^*(x_t)\big]$$

$$\leq -\frac{\eta_x}{2}\mathbb{E}\|\mathcal{H}^*(x_t)\|^2 - \frac{\eta_x}{2}\mathbb{E}\|\widetilde{h}_x^t\|^2 + \frac{\eta_x^2 L_{*,1}}{2}\mathbb{E}\left\|\frac{1}{|I_t|}\sum_{i\in I_t}\nabla_x\mathcal{L}_i(x_t,y_t,z_{i,t}^K,v_{i,t}^K)\right\|^2$$

$$+ \frac{3\eta_x L_{f,1}^2}{2}\mathbb{E}\|y_t - y^*(x_t)\|^2 + \frac{3\eta_x L_{\lambda,1}^2}{2}\mathbb{E}\left[\frac{1}{n}\sum_{i=1}^n\|z_{i,t}^K - z_{\lambda,i}^*(x_t)\|^2 + \frac{1}{n}\sum_{i=1}^n\|v_{i,t}^K - z_i^*(x_t)\|^2\right]$$

*for all $t \in \{0, 1, ..., T-1\}$, where we assume $\lambda \geq \frac{2L_{f,1}}{\mu_g}$.*

*Proof.* Recall the definitions of $\mathcal{L}^*(x)$ and $\mathcal{H}^*(x)$ in eq. (6). By using the smoothness of $\mathcal{L}^*(x_t)$ in Lemma D.6, we have that

$$\mathbb{E}\big[\mathcal{L}^*(x_{t+1})\big] \leq \mathbb{E}\big[\mathcal{L}^*(x_t)\big] + \mathbb{E}\langle\mathcal{H}^*(x_t), x_{t+1} - x_t\rangle + \frac{L_{*,1}}{2}\mathbb{E}\|x_{t+1} - x_t\|^2$$

$$= \mathbb{E}\big[\mathcal{L}^*(x_t)\big] - \eta_x\mathbb{E}\langle\mathcal{H}^*(x_t), \widetilde{h}_x^t\rangle + \frac{\eta_x^2 L_{*,1}}{2}\mathbb{E}\left\|\frac{1}{|I_t|}\sum_{i\in I_t}\nabla_x\mathcal{L}_i(x_t,y_t,z_{i,t}^K,v_{i,t}^K)\right\|^2$$

$$= \mathbb{E}\big[\mathcal{L}^*(x_t)\big] - \frac{\eta_x}{2}\mathbb{E}\|\mathcal{H}^*(x_t)\|^2 - \frac{\eta_x}{2}\mathbb{E}\|\widetilde{h}_x^t\|^2 + \frac{\eta_x}{2}\mathbb{E}\|\mathcal{H}^*(x_t) - \widetilde{h}_x^t\|^2$$

$$+ \frac{\eta_x^2 L_{*,1}}{2}\mathbb{E}\left\|\frac{1}{|I_t|}\sum_{i\in I_t}\nabla_x\mathcal{L}_i(x_t,y_t,z_{i,t}^K,v_{i,t}^K)\right\|^2$$

$$\overset{(a)}{\leq} \mathbb{E}\big[\mathcal{L}^*(x_t)\big] - \frac{\eta_x}{2}\mathbb{E}\|\mathcal{H}^*(x_t)\|^2 - \frac{\eta_x}{2}\mathbb{E}\|\widetilde{h}_x^t\|^2$$

$$+ \frac{\eta_x^2 L_{*,1}}{2}\mathbb{E}\left\|\frac{1}{|I_t|}\sum_{i\in I_t}\nabla_x\mathcal{L}_i(x_t,y_t,z_{i,t}^K,v_{i,t}^K)\right\|^2 + \frac{3\eta_x L_{f,1}^2}{2}\mathbb{E}\|y_t - y^*(x_t)\|^2$$

$$+ \frac{3\eta_x L_{\lambda,1}^2}{2}\mathbb{E}\left[\frac{1}{n}\sum_{i=1}^n\|z_{i,t}^K - z_{\lambda,i}^*(x_t)\|^2 + \frac{1}{n}\sum_{i=1}^n\|v_{i,t}^K - z_i^*(x_t)\|^2\right],$$

where (a) uses

$$\mathbb{E}\|\mathcal{H}^*(x_t) - \widetilde{h}_x^t\|^2$$

$$= \mathbb{E}\left\|\frac{1}{n}\sum_{i=1}^n\nabla_x\mathcal{L}_i\big(x_t,y^*(x_t),z_{\lambda,i}^*(x_t),z_i^*(x_t)\big) - \nabla_x\mathcal{L}_i\big(x_t,y_t,z_{i,t}^K,v_{i,t}^K\big)\right\|^2$$

$$\overset{(a.1)}{\leq} \frac{1}{n}\sum_{i=1}^n\mathbb{E}\|\nabla_x\mathcal{L}_i\big(x_t,y^*(x_t),z_{\lambda,i}^*(x_t),z_i^*(x_t)\big) - \nabla_x\mathcal{L}_i\big(x_t,y_t,z_{i,t}^K,v_{i,t}^K\big)\|^2$$

$$\overset{(a.2)}{\leq} 3L_{f,1}^2\mathbb{E}\|y_t - y^*(x_t)\|^2 + 3L_{\lambda,1}^2\mathbb{E}\left[\frac{1}{n}\sum_{i=1}^n\|z_{i,t}^K - z_{\lambda,i}^*(x_t)\|^2 + \frac{1}{n}\sum_{i=1}^n\|v_{i,t}^K - z_i^*(x_t)\|^2\right],$$

and (a.1) uses Jensen inequality; (a.2) follows from Lemma D.1. Then, the proof is complete. $\square$

## F.2 Bounds of Estimators

**Lemma F.2.** *Under Assumptions 4.3, 4.4, 4.5, we have the bounds of the estimators of $y_t$ and $x_t$ as*

$$\mathbb{E}\left\|\frac{1}{|I_t|}\sum_{i\in I_t}\nabla_y f_i(x_t, y_t, v_{i,t}^K)\right\|^2 \leq \left(1+\frac{\beta_{th}^2}{|I_t|}\right)L_{f,1}^2\left(\mathbb{E}\|y_t - y^*(x_t)\|^2 + \frac{1}{n}\sum_{i=1}^n \mathbb{E}\|v_{i,t}^K - z_i^*(x_t)\|^2\right)$$
$$+\frac{(n-|I_t|)\sigma_{th}^2}{(n-1)|I_t|},$$

$$\mathbb{E}\left\|\frac{1}{|I_t|}\sum_{i\in I_t}\nabla_x \mathcal{L}_i(x_t, y_t, z_{i,t}^K, v_{i,t}^K)\right\|^2 \leq \mathbb{E}\|\widetilde{h}_x^t\|^2 + \frac{3(n-|I_t|)}{(n-1)|I_t|}(L_{f,0}^2 + 2\lambda^2 L_{g,0}^2)$$

*for any $i \in \{1, ..., n\}$ and $t \in \{0, 1, ..., T-1\}$.*

*Proof.* For the estimator of $y$, we have

$$\mathbb{E}\left\|\frac{1}{|I_t|}\sum_{i\in I_t}\nabla_y f_i(x_t, y_t, v_{i,t}^K)\right\|^2$$

$$\overset{(a)}{=} \frac{n(|I_t|-1)}{|I_t|(n-1)}\mathbb{E}\left\|\frac{1}{n}\sum_{i=1}^n\nabla_y f_i(x_t, y_t, v_{i,t}^K)\right\|^2 + \frac{n-|I_t|}{(n-1)|I_t|}\cdot\frac{1}{n}\sum_{i=1}^n\mathbb{E}\left\|\nabla_y f_i(x_t, y_t, v_{i,t}^K)\right\|^2$$

$$\overset{(b)}{\leq} \frac{(n-|I_t|)\sigma_{th}^2}{(n-1)|I_t|} + \frac{n(|I_t|-1)+\beta_{th}^2(n-|I_t|)}{|I_t|(n-1)}\mathbb{E}\left\|\frac{1}{n}\sum_{i=1}^n\nabla_y f_i(x_t, y_t, v_{i,t}^K)\right\|^2$$

$$\overset{(c)}{\leq} \frac{(n-|I_t|)\sigma_{th}^2}{(n-1)|I_t|}$$
$$+ \frac{n(|I_t|-1)+\beta_{th}^2(n-|I_t|)}{|I_t|(n-1)}\mathbb{E}\left\|\frac{1}{n}\sum_{i=1}^n\nabla_y f_i(x_t, y_t, v_{i,t}^K) - \nabla_y f_i\big(x_t, y^*(x_t), z_i^*(x_t)\big)\right\|^2$$

$$\overset{(d)}{\leq} \frac{(n-|I_t|)\sigma_{th}^2}{(n-1)|I_t|} + \frac{n(|I_t|-1)+\beta_{th}^2(n-|I_t|)}{|I_t|(n-1)}L_{f,1}^2\left(\mathbb{E}\|y_t - y^*(x_t)\|^2 + \frac{1}{n}\sum_{i=1}^n\mathbb{E}\|v_{i,t}^K - z_i^*(x_t)\|^2\right)$$

$$\overset{(e)}{\leq} \frac{(n-|I_t|)\sigma_{th}^2}{(n-1)|I_t|} + \left(1+\frac{\beta_{th}^2}{|I_t|}\right)L_{f,1}^2\left(\mathbb{E}\|y_t - y^*(x_t)\|^2 + \frac{1}{n}\sum_{i=1}^n\mathbb{E}\|v_{i,t}^K - z_i^*(x_t)\|^2\right),$$

where (a) follows from the Lemma A.1 in [32]; (b) uses Assumption 4.5, 4.6; (c) follows from definition $y^*(x) = \arg\max_y \frac{1}{n}\sum_i^n f_i\big(x, y, z_i^*(x)\big)$; (d) uses Assumption 4.4 and (e) uses $\beta_{th} \geq 1$. Next, for the estimator of $y$, we have

$$\mathbb{E}\left\|\frac{1}{|I_t|}\sum_{i\in I_t}\nabla_x \mathcal{L}_i(x_t, y_t, z_{i,t}^K, v_{i,t}^K)\right\|^2 \overset{(a)}{=} \frac{n(|I_t|-1)}{|I_t|(n-1)}\mathbb{E}\left\|\frac{1}{n}\sum_{i=1}^n\nabla_x \mathcal{L}_i(x_t, y_t, z_{i,t}^K, v_{i,t}^K)\right\|^2$$

$$+ \frac{n-|I_t|}{(n-1)|I_t|}\cdot\frac{1}{n}\sum_{i=1}^n\mathbb{E}\left\|\nabla_x \mathcal{L}_i(x_t, y_t, z_{i,t}^K, v_{i,t}^K)\right\|^2$$

$$\leq \mathbb{E}\|\widetilde{h}_x^t\|^2 + \frac{3(n-|I_t|)}{(n-1)|I_t|}(L_{f,0}^2 + 2\lambda^2 L_{g,0}^2). \tag{42}$$

Then, the proof is complete. □

## F.3 Bounds of Sub-loop Errors

**Lemma F.3.** *Under Assumptions 4.3, 4.4, for $\forall \delta \in \mathbb{R}^+$, we assume that $\|z_i^*(x_t)\| \leq B$ for some $B < \infty$. Then we have*

$$\max\left\{\mathbb{E}\|v_{i,t}^K - z_i^*(x_t)\|^2, \mathbb{E}\|z_{i,t}^K - z_{\lambda,i}^*(x_t)\|^2\right\} \leq \epsilon_{sub}$$

*when $K \geq \max\left\{\frac{1}{\eta_t^v \mu_g}\log\frac{2\left(\|v_{i,t}^0\|^2+B^2\right)}{\epsilon_{sub}}, \frac{1}{\eta_t^z L_{f,1}}\log\frac{3\left(\|z_{i,t}^0\|^2+\frac{L_{f,0}^2}{4L_{f,1}^2}+B^2\right)}{\epsilon_{sub}}\right\}$, where $\eta_t^v \in (0, \frac{1}{2L_{g,1}})$, $\eta_t^z \in (0, \frac{1}{4\lambda L_{g,1}})$, $\lambda \geq \frac{L_{f,1}}{\mu_g}$.*

*Proof.* From Lemma D.1, we have that $\mathcal{L}_i$ is $\frac{\lambda \mu_g}{2}$-strongly convex in $z$ and we have that $\mathcal{L}_i$ is $2\lambda L_{g,1}$-Lipschitz continue in $z$ when $\lambda \geq \frac{L_{f,1}}{\mu_g}$; also, from Assumption 4.3, 4.4, we have that $g_i$ is $\mu_g$-strongly convex in $v$ and $L_{g,1}$-smooth in $v$. According to Theorem 3.6 in [11], by taking $0 < \eta_t^v < \frac{1}{2L_{g,1}}$ and $0 < \eta_t^z < \frac{1}{4\lambda L_{g,1}}$, we have

$$\mathbb{E}\|v_{i,t}^K - z_i^*(x_t)\|^2 \leq (1 - \eta_t^v \mu_g)^K \mathbb{E}\|v_{i,t}^0 - z_i^*(x_t)\|^2,$$

$$\mathbb{E}\|z_{i,t}^K - z_{\lambda,i}^*(x_t)\|^2 \leq (1 - \frac{\eta_t^z \lambda \mu_g}{2})^K \mathbb{E}\|z_{i,t}^0 - z_{\lambda,i}^*(x_t)\|^2.$$

To make sure $\mathbb{E}\|v_{i,t}^K - z_i^*(x_t)\|^2 \leq \epsilon_{sub}$ and $\mathbb{E}\|z_{i,t}^K - z_{\lambda,i}^*(x_t)\|^2 \leq \epsilon_{sub}$ for some $\epsilon_{sub} \geq 0$, we let

$$(1 - \eta_t^v \mu_g)^K \mathbb{E}\|v_{i,t}^0 - z_i^*(x_t)\|^2 \leq 2(1 - \eta_t^v \mu_g)^K \left(\|v_{i,t}^0\|^2 + B^2\right) \leq \epsilon_{sub},$$

and

$$(1 - \frac{\eta_t^z \lambda \mu_g}{2})^K \mathbb{E}\|z_{i,t}^0 - z_{\lambda,i}^*(x_t)\|^2$$

$$\leq 3(1 - \frac{\eta_t^z \lambda \mu_g}{2})^K \left(\|z_{i,t}^0\|^2 + \mathbb{E}\|z_i^*(x_t) - z_{\lambda,i}^*(x_t)\|^2 + \mathbb{E}\|z_i^*(x_t)\|^2\right)$$

$$\overset{(a)}{\leq} 3(1 - \frac{\eta_t^z \lambda \mu_g}{2})^K \left(\|z_{i,t}^0\|^2 + \frac{L_{f,0}^2}{\mu_g^2 \lambda^2} + B^2\right)$$

$$\overset{(b)}{\leq} 3(1 - \frac{\eta_t^z \lambda \mu_g}{2})^K \left(\|z_{i,t}^0\|^2 + \frac{L_{f,0}^2}{4L_{f,1}^2} + B^2\right)$$

$$\leq \epsilon_{sub},$$

where (a) uses Lemma D.4; (b) take $\lambda \geq \frac{2L_{f,1}}{\mu_g}$. Both can be achieved by taking

$$K \geq \max\left\{\frac{1}{\eta_t^v \mu_g} \log \frac{2(\|v_{i,t}^0\|^2 + B^2)}{\epsilon_{sub}}, \frac{1}{\eta_t^z L_{f,1}} \log \frac{3(\|z_{i,t}^0\|^2 + \frac{L_{f,0}^2}{4L_{f,1}^2} + B^2)}{\epsilon_{sub}}\right\}.$$

Then the proof is complete. $\qquad \square$

**Lemma F.4.** *Under the Assumptions 4.3, 4.4, 4.5, the iterates of $y_t$ updates according to Algorithm 2 satisfy*

$$\mathbb{E}\|y_{t+1} - y^*(x_{t+1})\|^2 - \mathbb{E}\|y_t - y^*(x_t)\|^2$$

$$\leq \left[-\eta_y \mu_f + \eta_y^2(1 + \delta_y)\left(1 + \frac{\beta_{th}^2}{|I_t|}\right)L_{f,1}^2 + \delta_y\right]\mathbb{E}\|y_t - y^*(x_t)\|^2 + \eta_y^2(1 + \delta_y)\frac{(n - |I_t|)\sigma_{th}^2}{(n-1)|I_t|}$$

$$+ \left(\frac{\eta_y L_{f,1}^2}{\mu_f} + \eta_y^2\left(1 + \frac{\beta_{th}^2}{|I_t|}\right)L_{f,1}^2\right)(1 + \delta_y) \cdot \frac{1}{n}\sum_{i=1}^n \mathbb{E}\|v_{i,t} - z_i^*(x_t)\|^2$$

$$+ \frac{\eta_x^2}{\delta_{y,1}}\left(1 + \frac{L_{g,1}}{\mu_g}\right)^2 \frac{L_{f,1}^2}{\mu_f^2}\mathbb{E}\|\widetilde{h}_x^t\|^2$$

$$+ \eta_x^2\left(\frac{L_{*,y}}{2} + \left(1 + \frac{L_{g,1}}{\mu_g}\right)^2 \frac{L_{f,1}^2}{\mu_f^2}\right)\mathbb{E}\left\|\frac{1}{|I_t|}\sum_{i \in I_t}\nabla_x \mathcal{L}_i(x_t, y_t, z_{i,t}^K, v_{i,t}^K)\right\|^2$$

*for any $t \in \{0, ..., T - 1\}$, where we assume $\lambda \geq \{2L_{f,1}/\mu_g, (1 + \frac{L_{g,1}}{\mu_g})\frac{L_{f,1}^2}{3\mu_f L_{g,1}}, (1 + \frac{L_{g,1}}{\mu_g})\frac{L_{f,1}L_{f,2}}{3\mu_f L_{g,1}}, \frac{L_{f,1}L_{*,y}}{6L_{g,1}}\left(1 + (1 + \frac{L_{g,1}}{\mu_g})\frac{L_{f,1}}{\mu_f} + \frac{12L_{g,1}}{\mu_g}\right)^{-1}, \left((1 + \frac{L_{g,1}}{\mu_g})\frac{L_{f,1}}{\mu_f} + 1\right)\frac{L_{f,1}}{L_{g,1}}\}$.*

*Proof.* For $y$ and $y^*(x)$, we have

$$\mathbb{E}\|y_{t+1} - y^*(x_{t+1})\|^2 = \mathbb{E}\|y_{t+1} - y^*(x_t)\|^2 + \mathbb{E}\|y^*(x_t) - y^*(x_{t+1})\|^2$$
$$+ 2\mathbb{E}\langle y_{t+1} - y^*(x_t), y^*(x_t) - y^*(x_{t+1})\rangle. \tag{43}$$

We can bound the first term in eq. (43) as

$$\mathbb{E}\|y_{t+1} - y^*(x_t)\|^2$$

$$= \mathbb{E}\left\| y_t + \frac{1}{|I_t|} \sum_{i \in I_t} \nabla_y f_i(x_t, y_t, v_{i,t}^K) - y^*(x_t) \right\|^2$$

$$= \mathbb{E}\|y_t - y^*(x_t)\|^2 + \eta_y^2 \mathbb{E}\left\| \frac{1}{|I_t|} \sum_{i \in I_t} \nabla_y f_i(x_t, y_t, v_{i,t}^K) \right\|^2$$

$$+ 2\eta_y \mathbb{E}\left\langle y_t - y^*(x_t), \frac{1}{|I_t|} \sum_{i \in I_t} \nabla_y f_i(x_t, y_t, v_{i,t}^K) \right\rangle$$

$$\overset{(a)}{=} \mathbb{E}\|y_t - y^*(x_t)\|^2 + \eta_y^2 \mathbb{E}\left\| \frac{1}{|I_t|} \sum_{i \in I_t} \nabla_y f_i(x_t, y_t, v_{i,t}^K) \right\|^2$$

$$+ 2\eta_y \mathbb{E}\left\langle y_t - y^*(x_t), \frac{1}{n} \sum_{i=1}^{n} \nabla_y f_i(x_t, y_t, v_{i,t}^K) - \nabla_y f_i(x_t, y_t, z_i^*(x_t)) \right\rangle$$

$$+ 2\eta_y \mathbb{E}\left\langle y_t - y^*(x_t), \frac{1}{n} \sum_{i=1}^{n} \nabla_y f_i(x_t, y_t, z_i^*(x_t)) - \nabla_y f_i(x_t, y^*(x_t), z_i^*(x_t)) \right\rangle$$

$$\overset{(b)}{\leq} \mathbb{E}\|y_t - y^*(x_t)\|^2 + \eta_y^2 \mathbb{E}\left\| \frac{1}{|I_t|} \sum_{i \in I_t} \nabla_y f_i(x_t, y_t, v_{i,t}^K) \right\|^2$$

$$+ \eta_y \left( \mu_f \mathbb{E}\|y_t - y^*(x_t)\|^2 + \frac{1}{\mu_f} \mathbb{E}\left\| \frac{1}{n} \sum_{i=1}^{n} \nabla_y f_i(x_t, y_t, v_{i,t}^K) - \nabla_y f_i(x_t, y_t, z_i^*(x_t)) \right\|^2 \right)$$

$$- 2\eta_y \mu_f \mathbb{E}\|y_t - y^*(x_t)\|^2$$

$$\overset{(c)}{\leq} (1 - \eta_y \mu_f) \mathbb{E}\|y_t - y^*(x_t)\|^2 + \eta_y^2 \mathbb{E}\left\| \frac{1}{|I_t|} \sum_{i \in I_t} \nabla_y f_i(x_t, y_t, v_{i,t}^K) \right\|^2$$

$$+ \frac{\eta_y L_{f,1}^2}{\mu_f} \cdot \frac{1}{n} \sum_{i=1}^{n} \mathbb{E}\|v_{i,t}^K - z_i^*(x_t)\|^2$$

$$\overset{(d)}{\leq} \left( 1 - \eta_y \mu_f + \eta_y^2 \left(1 + \frac{\beta_{th}^2}{|I_t|}\right) L_{f,1}^2 \right) \mathbb{E}\|y_t - y^*(x_t)\|^2 + \eta_y^2 \frac{(n - |I_t|)\sigma_{th}^2}{(n-1)|I_t|}$$

$$+ \left( \frac{\eta_y L_{f,1}^2}{\mu_f} + \eta_y^2 \left(1 + \frac{\beta_{th}^2}{|I_t|}\right) L_{f,1}^2 \right) \cdot \frac{1}{n} \sum_{i=1}^{n} \mathbb{E}\|v_{i,t}^K - z_i^*(x_t)\|^2 \tag{44}$$

where (a) uses the definition of $y^*(x_t)$ and eq. (5); (b) uses strong concavity of $f_i$ in $y$; (c) follos from definition of $y^*(x_t)$ and Assumption 4.4; (d) uses Lemma F.2. We can bound the second term in eq. (43) as

$$\mathbb{E}\|y^*(x_t) - y^*(x_{t+1})\|^2 \overset{(a)}{\leq} \left(1 + \frac{L_{g,1}}{\mu_g}\right)^2 \frac{L_{f,1}^2}{\mu_f^2} \mathbb{E}\|x_t - x_{t+1}\|^2$$

$$= \eta_x^2 \left(1 + \frac{L_{g,1}}{\mu_g}\right)^2 \frac{L_{f,1}^2}{\mu_f^2} \mathbb{E}\left\| \frac{1}{|I_t|} \sum_{i \in I_t} \nabla_y f_i(x_t, y_t, v_{i,t}^K) \right\|^2, \tag{45}$$

where (a) follows from Lemma D.2. Also, we can get the bound of the last term as

$$2\mathbb{E}\langle y_{t+1} - y^*(x_t), y^*(x_t) - y^*(x_{t+1}) \rangle$$

$$= -2\mathbb{E}\langle y_{t+1} - y^*(x_t), \nabla y^*(x_t)(x_{t+1} - x_t) \rangle$$

$$- 2\mathbb{E}\langle y_{t+1} - y^*(x_t), y^*(x_{t+1}) - y^*(x_t) - \nabla y^*(x_t)(x_{t+1} - x_t) \rangle$$

$$= 2\mathbb{E}\|y_{t+1} - y^*(x_t)\| \cdot \|\eta_x \nabla y^*(x_t)\tilde{h}_x^t\|$$

$$+ 2\mathbb{E}\|y_{t+1} - y^*(x_t)\| \cdot \|y^*(x_{t+1}) - y^*(x_t) - \nabla y^*(x_t)(x_{t+1} - x_t)\|$$

$$
\overset{(a)}{=} \delta_{y,1}\mathbb{E}\big\|y_{t+1}-y^*(x_t)\big\|^2 + \frac{\eta_x^2}{\delta_{y,1}}\left(1+\frac{L_{g,1}}{\mu_g}\right)^2\frac{L_{f,1}^2}{\mu_f^2}\mathbb{E}\big\|\widetilde{h}_x^t\big\|^2
$$
$$
+ \mathbb{E}\big\|y_{t+1}-y^*(x_t)\big\| \cdot L_{*,y}\big\|x_{t+1}-x_t\big\|^2
$$
$$
\leq \delta_{y,1}\mathbb{E}\big\|y_{t+1}-y^*(x_t)\big\|^2 + \frac{\eta_x^2}{\delta_{y,1}}\left(1+\frac{L_{g,1}}{\mu_g}\right)^2\frac{L_{f,1}^2}{\mu_f^2}\mathbb{E}\big\|\widetilde{h}_x^t\big\|^2
$$
$$
+ \frac{L_{*,y}}{2}\mathbb{E}\big\|y_{t+1}-y^*(x_t)\big\|^2 \cdot \big\|x_{t+1}-x_t\big\|^2 + \frac{L_{*,y}}{2}\mathbb{E}\big\|x_{t+1}-x_t\big\|^2
$$
$$
\overset{(b)}{\leq} \left(\delta_{y,1}+\frac{3\eta_x^2 L_{*,y}}{2}(L_{f,0}^2+2\lambda^2 L_{g,0}^2)\right)\mathbb{E}\big\|y_{t+1}-y^*(x_t)\big\|^2
$$
$$
+ \frac{\eta_x^2}{\delta_{y,1}}\left(1+\frac{L_{g,1}}{\mu_g}\right)^2\frac{L_{f,1}^2}{\mu_f^2}\mathbb{E}\big\|\widetilde{h}_x^t\big\|^2 + \frac{\eta_x^2 L_{*,y}}{2}\mathbb{E}\left\|\frac{1}{|I_t|}\sum_{i\in I_t}\nabla_x\mathcal{L}_i(x_t,y_t,z_{i,t}^K,v_{i,t}^K)\right\|^2
$$
$$
\overset{(c)}{\leq} \delta_y\mathbb{E}\big\|y_{t+1}-y^*(x_t)\big\|^2 + \frac{\eta_x^2}{\delta_{y,1}}\left(1+\frac{L_{g,1}}{\mu_g}\right)^2\frac{L_{f,1}^2}{\mu_f^2}\mathbb{E}\big\|\widetilde{h}_x^t\big\|^2
$$
$$
+ \frac{\eta_x^2 L_{*,y}}{2}\mathbb{E}\left\|\frac{1}{|I_t|}\sum_{i\in I_t}\nabla_x\mathcal{L}_i(x_t,y_t,z_{i,t}^K,v_{i,t}^K)\right\|^2, \tag{46}
$$

where (a) uses Lemma D.2, D.3 and Lemma 1 in [44]; (b) follows from Assumption 4.4; (c) defines $\delta_y = \delta_{y,1} + \frac{3\eta_x^2 L_{*,y}}{2}(L_{f,0}^2+2\lambda^2 L_{g,0}^2)$. By plugging eq. (44), eq. (45), eq. (46) into eq. (43), we get

$$
\mathbb{E}\|y_{t+1}-y^*(x_{t+1})\|^2 - \mathbb{E}\|y_t-y^*(x_t)\|^2
$$
$$
\leq \left[-\eta_y\mu_f + \eta_y^2(1+\delta_y)\left(1+\frac{\beta_{th}^2}{|I_t|}\right)L_{f,1}^2 + \delta_y\right]\mathbb{E}\|y_t-y^*(x_t)\|^2 + \eta_y^2(1+\delta_y)\frac{(n-|I_t|)\sigma_{th}^2}{(n-1)|I_t|}
$$
$$
+ \left(\frac{\eta_y L_{f,1}^2}{\mu_f} + \eta_y^2\left(1+\frac{\beta_{th}^2}{|I_t|}\right)L_{f,1}^2\right)(1+\delta_y)\cdot\frac{1}{n}\sum_{i=1}^n\mathbb{E}\|v_{i,t}-z_i^*(x_t)\|^2
$$
$$
+ \frac{\eta_x^2}{\delta_{y,1}}\left(1+\frac{L_{g,1}}{\mu_g}\right)^2\frac{L_{f,1}^2}{\mu_f^2}\mathbb{E}\big\|\widetilde{h}_x^t\big\|^2
$$
$$
+ \eta_x^2\left(\frac{L_{*,y}}{2}+\left(1+\frac{L_{g,1}}{\mu_g}\right)^2\frac{L_{f,1}^2}{\mu_f^2}\right)\mathbb{E}\left\|\frac{1}{|I_t|}\sum_{i\in I_t}\nabla_x\mathcal{L}_i(x_t,y_t,z_{i,t}^K,v_{i,t}^K)\right\|^2.
$$

Then the proof is complete. $\qquad\qquad\square$

## F.4 Descent in the Lyapunov Function and Proof of Theorem 4.12

We define the Lyapunov function as

$$
\Psi(x) := \mathcal{L}^*(x) + K_y\mathbb{E}\|y_t-y^*(x_t)\|^2, \tag{47}
$$

where the coefficient is given by $K_y = \frac{3\eta_x L_{f,1}^2}{\eta_y\mu_f}$. We also constrain the conditions as below:

$$
\delta_{y,1} = \frac{\eta_y\mu_f}{4}, \quad \eta_x \leq \frac{1}{3L_{*,1}}, \quad \eta_y \leq \min\left\{\frac{1}{(1+\beta_{th}^2)\mu_f}, \frac{\mu_f}{8(1+\beta_{th}^2)L_{g,1}^2}\right\},
$$
$$
\frac{\eta_x^2}{\eta_y} \leq \min\left\{\frac{\mu_g}{12L_{*,y}(L_{f,0}^2+2\lambda^2 L_{g,0}^2)}, \frac{\mu_f}{18L_{f,1}^2}\left(\frac{L_{*,y}}{2}+\left(1+\frac{6L_{g,1}}{\mu_g}\right)^2\frac{L_{f,1}^2}{\mu_f^2}\right)^{-1}\right\},
$$
$$
\frac{\eta_x}{\eta_y} \leq \frac{\mu_f^2}{6\sqrt{2}L_{f,1}^2}\left(1+\frac{6L_{g,1}}{\mu_g}\right)^{-1},
$$
$$
\lambda \geq \max\left\{2L_{f,1}/\mu_g, (1+\frac{L_{g,1}}{\mu_g})\frac{L_{f,1}^2}{3\mu_f L_{g,1}}, (1+\frac{L_{g,1}}{\mu_g})\frac{L_{f,1}L_{f,2}}{3\mu_f L_{g,1}},\right.
$$

$$\frac{L_{f,1}L_{*,y}}{6L_{g,1}}\Big(1+(1+\frac{L_{g,1}}{\mu_g})\frac{L_{f,1}}{\mu_f}+\frac{12L_{g,1}}{\mu_g}\Big)^{-1},\Big((1+\frac{L_{g,1}}{\mu_g})\frac{L_{f,1}}{\mu_f}+1\Big)\frac{L_{f,1}}{L_{g,1}}\Big\}. \quad (48)$$

Plugging Lemma F.1, Lemma E.3 into eq. (47), we have **the descent in the Lyapunov function** as

$$\Psi_{t+1}-\Psi_t$$

$$\leq -\frac{\eta_x}{2}\mathbb{E}\|\mathcal{H}^*(x_t)\|^2-\eta_x\Big(\frac{1}{2}-\frac{K_y\eta_x}{\eta_y}\cdot\frac{4}{\mu_f}\Big(1+\frac{L_{g,1}}{\mu_g}\Big)^2\frac{L_{f,1}^2}{\mu_f^2}\Big)\mathbb{E}\|\widetilde{h}_x^t\|^2$$

$$+\eta_x^2\Big[\frac{L_{*,1}}{2}+K_y\Big(\frac{L_{*,y}}{2}+\Big(1+\frac{L_{g,1}}{\mu_g}\Big)^2\frac{L_{f,1}^2}{\mu_f^2}\Big)\Big]\mathbb{E}\Big\|\frac{1}{|I_t|}\sum_{i\in I_t}\nabla_x\mathcal{L}_i(x_t,y_t,z_{i,t}^K,v_{i,t}^K)\Big\|^2$$

$$+2K_y\eta_y^2\frac{(n-|I_t|)\sigma_{th}^2}{(n-1)|I_t|}+\frac{3\eta_x L_{\lambda,1}^2}{2}\mathbb{E}\Big[\frac{1}{n}\sum_{i=1}^n\|z_{i,t}^K-z_{\lambda,i}^*(x_t)\|+\frac{1}{n}\sum_{i=1}^n\|v_{i,t}^K-z_i^*(x_t)\|\Big]$$

$$+2K_y\Big(\frac{\eta_y L_{f,1}^2}{\mu_f}+\eta_y^2\Big(1+\frac{\beta_{th}^2}{|I_t|}\Big)L_{f,1}^2\Big)\cdot\frac{1}{n}\sum_{i=1}^n\mathbb{E}\|v_{i,t}^K-z_i^*(x_t)\|^2$$

$$\overset{(a)}{\leq}-\frac{\eta_x}{2}\mathbb{E}\|\mathcal{H}^*(x_t)\|^2-\eta_x\Big(\frac{1}{2}-\frac{K_y\eta_x}{\eta_y}\cdot\frac{4}{\mu_f}\Big(1+\frac{L_{g,1}}{\mu_g}\Big)^2\frac{L_{f,1}^2}{\mu_f^2}\Big)\mathbb{E}\|\widetilde{h}_x^t\|^2$$

$$+\eta_x^2\Big[\frac{L_{*,1}}{2}+K_y\Big(\frac{L_{*,y}}{2}+\Big(1+\frac{L_{g,1}}{\mu_g}\Big)^2\frac{L_{f,1}^2}{\mu_f^2}\Big)\Big]\mathbb{E}\|\widetilde{h}_x^t\|^2$$

$$+\eta_x^2\Big[\frac{L_{*,1}}{2}+K_y\Big(\frac{L_{*,y}}{2}+\Big(1+\frac{L_{g,1}}{\mu_g}\Big)^2\frac{L_{f,1}^2}{\mu_f^2}\Big)\Big]\frac{3(n-|I_t|)}{(n-1)|I_t|}(L_{f,0}^2+2\lambda^2 L_{g,0}^2)$$

$$+2K_y\eta_y^2\frac{(n-|I_t|)\sigma_{th}^2}{(n-1)|I_t|}+\frac{3\eta_x L_{\lambda,1}^2}{2}\mathbb{E}\Big[\frac{1}{n}\sum_{i=1}^n\|z_{i,t}^K-z_{\lambda,i}^*(x_t)\|+\frac{1}{n}\sum_{i=1}^n\|v_{i,t}^K-z_i^*(x_t)\|\Big]$$

$$+\frac{4K_y\eta_y L_{f,1}^2}{\mu_f}\cdot\frac{1}{n}\sum_{i=1}^n\mathbb{E}\|v_{i,t}^K-z_i^*(x_t)\|^2$$

$$\overset{(b)}{\leq}-\frac{\eta_x}{2}\mathbb{E}\|\mathcal{H}^*(x_t)\|^2+\eta_x^2\Big[\frac{L_{*,1}}{2}+K_y\Big(\frac{L_{*,y}}{2}+\Big(1+\frac{L_{g,1}}{\mu_g}\Big)^2\frac{L_{f,1}^2}{\mu_f^2}\Big)\Big]\frac{3(n-|I_t|)}{(n-1)|I_t|}(L_{f,0}^2+2\lambda^2 L_{g,0}^2)$$

$$+2K_y\eta_y^2\frac{(n-|I_t|)\sigma_{th}^2}{(n-1)|I_t|}+\eta_x\Big(3L_{\lambda,1}^2+\frac{12L_{f,1}^4}{\mu_f^2}\Big)\epsilon_{sub}$$

$$\overset{(c)}{\leq}-\frac{\eta_x}{2}\mathbb{E}\|\mathcal{H}^*(x_t)\|^2+\eta_x^2\Big(1+\frac{\eta_x}{\eta_y}\Big)C_4\frac{(n-|I_t|)\lambda^2}{(n-1)|I_t|}+\eta_x\eta_y\frac{n-|I_t|}{(n-1)|I_t|}\frac{6L_{f,1}^2\sigma_{th}^2}{\mu_f}$$

$$+\eta_x\Big(3L_{\lambda,1}^2+\frac{12L_{f,1}^4}{\mu_f^2}\Big)\epsilon_{sub}, \quad (49)$$

where (a) uses Lemma F.2; (b) uses eq. (48) and takes

$$K\geq\max\Big\{\frac{1}{\eta_t^v\mu_g}\log\frac{2\mathbb{E}\|v_{i,t}^0-z_i^*(x_t)\|^2}{\epsilon_{sub}},\frac{2}{\eta_t^z\lambda\mu_g}\log\frac{2\mathbb{E}\|z_{i,t}^0-z_{\lambda,i}^*(x_t)\|^2}{\epsilon_{sub}}\Big\};$$

(c) defines $C_4:=\frac{3(L_{f,0}^2+2\lambda^2 L_{g,0}^2)}{\lambda^2}\cdot\max\Big\{\frac{L_{*,1}}{2},\frac{3L_{f,1}^2}{\mu_f}\Big(\frac{L_{*,y}}{2}+\Big(1+\frac{L_{g,1}}{\mu_g}\Big)^2\frac{L_{f,1}^2}{\mu_f^2}\Big)\Big\}$ and plugs in $K_y$.

### F.5 Proof of Theorem 4.12

*Proof.* For partial block participation, we take the summation of eq. (49) from $t=0$ to $T-1$. Then we have

$$\frac{1}{T}\sum_{t=0}^{T-1}\frac{\eta_x}{2}\mathbb{E}\|\mathcal{H}^*(x_t)\|^2\leq\Psi(x_0)-\Psi(x_T)+\eta_x^2\Big(1+\frac{\eta_x}{\eta_y}\Big)C_4\frac{(n-P)\lambda^2}{(n-1)P}$$

$$+\eta_x\eta_y\frac{n-P}{(n-1)P}\frac{6L_{f,1}^2\sigma_{th}^2}{\mu_f}+\eta_x\Big(3L_{\lambda,1}^2+\frac{12L_{f,1}^4}{\mu_f^2}\Big)\epsilon_{sub}. \quad (50)$$

For partial block participation, By using Lemma D.5, we have

$$\frac{1}{T}\sum_{t=0}^{T-1}\mathbb{E}\|\nabla\Phi(x_t)\|^2\leq\frac{2}{T}\sum_{t=0}^{T-1}\mathbb{E}\|\nabla\Phi(x_t)-\mathcal{H}^*(x_t)\|^2+\frac{2}{T}\sum_{t=0}^{T-1}\mathbb{E}\|\mathcal{H}^*(x_t)\|^2$$

$$\overset{(a)}{\leq} \frac{2C_{gap}}{\lambda^2} + \frac{2\big(\Psi(x_0) - \Psi(x_T)\big)}{T\eta_x} + 4\eta_x\Big(1 + \frac{\eta_x}{\eta_y}\Big)C_4 \frac{(n-P)\lambda^2}{(n-1)P}$$

$$+ \eta_y \frac{n-P}{(n-1)P} \frac{24L_{f,1}^2\sigma_{th}^2}{\mu_f} + 4\Big(3L_{\lambda,1}^2 + \frac{12L_{f,1}^4}{\mu_f^2}\Big)\epsilon_{sub}$$

$$\overset{(b)}{\leq} \frac{2C_{gap}}{\lambda^2} + \frac{2\big(\Psi(x_0) - \Psi(x_T)\big)}{T\eta_x} + \frac{4\eta_x\lambda^2}{P}\Big(1 + \frac{\eta_x}{\eta_y}\Big)C_4 + \frac{\eta_y}{P}\frac{24L_{f,1}^2\sigma_{th}^2}{\mu_f}$$

$$+ 4\Big(3L_{\lambda,1}^2 + \frac{12L_{f,1}^4}{\mu_f^2}\Big)\epsilon_{sub}$$

$$\leq \frac{2C_{gap}}{\lambda^2} + \frac{2\big(\Psi(x_0) - \Psi(x_T)\big)}{T\eta_x} + \frac{4\eta_x\lambda^2}{P}\Big(1 + \frac{\eta_x}{\eta_y}\Big)C_4 + \frac{\eta_y}{P}\frac{24L_{f,1}^2\sigma_{th}^2}{\mu_f}$$

$$+ 4\Big(9\lambda^2 L_{g,1}^2 + \frac{12L_{f,1}^4}{\mu_f^2}\Big)\epsilon_{sub}$$

$$\overset{(c)}{\leq} \mathcal{O}(P^{-\frac{1}{5}}T^{-\frac{1}{3}}). \tag{51}$$

where (a) follows from eq. (50); (b) follows from $1 \leq P < n$; (c) takes $\eta_x = \mathcal{O}(P^{\frac{1}{5}}T^{-\frac{2}{3}})$, $\eta_y = \mathcal{O}(P^{-\frac{1}{5}}T^{-\frac{1}{2}})$, $\lambda = \mathcal{O}(P^{\frac{1}{10}}T^{\frac{1}{6}})$, $\epsilon_{sub} = \mathcal{O}(P^{-\frac{2}{5}}T^{-\frac{2}{3}})$.

Next, for full block participation $(n = P)$, we have **the descent in the Lyapunov function for full block participation** as

$$\Psi(x_{t+1}) - \Psi(x_t) \leq -\frac{\eta_x}{2}\mathbb{E}\|\mathcal{H}^*(x_t)\|^2 + \eta_x\Big(3L_{\lambda,1}^2 + \frac{12L_{f,1}^4}{\mu_f^2}\Big)\epsilon_{sub}. \tag{52}$$

Taking summation of eq. (52) from $t = 0$ to $T - 1$, we have

$$\frac{1}{T}\sum_{t=0}^{T-1}\frac{\eta_x}{2}\mathbb{E}\|\mathcal{H}^*(x_t)\|^2 \leq \Psi(x_0) - \Psi(x_T) + \eta_x\Big(3L_{\lambda,1}^2 + \frac{12L_{f,1}^4}{\mu_f^2}\Big)\epsilon_{sub}. \tag{53}$$

By using Lemma D.5 and eq. (53), we have

$$\frac{1}{T}\sum_{t=0}^{T-1}\mathbb{E}\|\nabla\Phi(x_t)\|^2 \leq \frac{2}{T}\sum_{t=0}^{T-1}\mathbb{E}\|\nabla\Phi(x_t) - \mathcal{H}^*(x_t)\|^2 + \frac{2}{T}\sum_{t=0}^{T-1}\mathbb{E}\|\mathcal{H}^*(x_t)\|^2$$

$$\leq \frac{2C_{gap}}{\lambda^2} + \frac{4\big(\Psi(x_0) - \Psi(x_T)\big)}{T\eta_x} + 12\Big(L_{\lambda,1}^2 + \frac{4L_{f,1}^4}{\mu_f^2}\Big)\epsilon_{sub}$$

$$\leq \frac{2C_{gap}}{\lambda^2} + \frac{4\big(\Psi(x_0) - \Psi(x_T)\big)}{T\eta_x} + 12\Big(9\lambda^2 L_{g,1}^2 + \frac{4L_{f,1}^4}{\mu_f^2}\Big)\epsilon_{sub}$$

$$\leq \mathcal{O}(T^{-1}). \tag{54}$$

By taking $\eta_x = \mathcal{O}(1)$, $\eta_y = \mathcal{O}(1)$, $\lambda = \mathcal{O}(T^{\frac{1}{2}})$, $\epsilon_{sub} = \mathcal{O}(T^{-2})$. Then Theorem 4.12 is proved. $\square$

### F.6 Proof of Corollary 4.13

*Proof.* For tasks participate in updates partially, by eq. (51), we can find the $\epsilon$-stationary point in definition 4.2 once we take $T = \mathcal{O}(P^{-\frac{3}{5}}\epsilon^{-3})$. Note that we set the error of sub-loop as $\epsilon_{sub} = \mathcal{O}(P^{-\frac{2}{5}}T^{-\frac{2}{3}}) = \mathcal{O}(\epsilon^2)$. According to Lemma F.3, once we take $\eta_v = \mathcal{O}(1)$ and $\eta_z = \mathcal{O}(P^{-\frac{1}{10}}T^{-\frac{1}{6}})$, we have the iteration number of sub-loop as $K = \mathcal{O}\big(\log(\frac{1}{\epsilon})\big)$. Thus, we have the total sample complexity $PKT = \mathcal{O}(P^{\frac{2}{5}}\epsilon^{-3}\log(\frac{1}{\epsilon})) = \widetilde{\mathcal{O}}(P^{\frac{2}{5}}\epsilon^{-3})$.

Similarly, by eq. (54), we can find the $\epsilon$-stationary point in definition 4.2 once we take $T = \mathcal{O}(\epsilon^{-1})$. Note that we set the error of sub-loop as $\epsilon_{sub} = \mathcal{O}(T^{-2}) = \mathcal{O}(\epsilon^2)$. Once we take $\eta_v = \mathcal{O}(1)$ and $\eta_z = \mathcal{O}(T^{-\frac{1}{2}})$, we have the iteration number of sub-loop as $K = \mathcal{O}\big(\log(\frac{1}{\epsilon})\big)$. Thus, we have the total sample complexity $nKT = \mathcal{O}(n\epsilon^{-1}\log(\frac{1}{\epsilon})) = \widetilde{\mathcal{O}}(n\epsilon^{-1})$. $\square$

