# OpenReview forum: "First-Order Minimax Bilevel Optimization"
_NeurIPS.cc/2024/Conference — NeurIPS 2024 poster_

### Official Review · Reviewer_mptp · 2024-06-23

**Soundness:** 2
**Presentation:** 3
**Contribution:** 3
**Rating:** 6
**Confidence:** 4

**Summary:**

This paper proposes two novel algorithms, FOSL and MemCS, for multi-block minimax bilevel optimization problems, avoiding the high complexity of computing the second-order gradient. Their theoretical analysis is quite solid and their experiments show proposed algorithms have superior performance and robustness in applications.

Overall, the proposed algorithms are novel and well-justified with practical applications. However, further elaboration is needed regarding their practicality and efficiency.

**Strengths:**

The paper is well-written and well-structured, with strong motivation and clear logic.

The authors provide a reformulated version of minimax bilevel optimization problems and demonstrate the gap between the reformulated and original problems. The convergence analysis is also solid and thorough.

They also validate the experimental results of FOSL and MemCS in Deep AUC Maximization and meta-learning cases respectively.

**Weaknesses:**

Since the paper primarily claims the efficiency of the proposed first-order algorithms, the authors should discuss the scale of problems these algorithms are suitable for. For example, second-order methods may fail when dealing with a large number of variables. In contrast, first-order algorithms generally handle larger-scale problems better, but the paper lacks experimental comparisons in this regard.

Besides, the gradient calculation procedure is not mentioned in the paper. I believe it is quite important for incorporating optimization problems in neural networks.

There is also a typo in the title of algorithm 2: Cold-star

**Questions:**

- Since the theoretical proof is based on various assumptions, are there any restrictions on FOSL and MemCS? For example, do they only handle convex optimization problems? Including the previously mentioned issue of the optimization problem scale, I think the applicable scope of the two algorithms should be discussed in detail.

- Could the authors include a comparison of the runtime differences between FOSL, MemCS, and second-order methods? This would better illustrate the effectiveness of the proposed methods.

- In the experimental results, FOSL and MemCS perform better. Is this because these two algorithms have a better gradient estimator? The authors should provide an explanation for this.

- Additionally, I would like to know how the gradient of the optimization problem is calculated. Is it from implicit differentiation, unrolling methods, or are there any direct analytical solutions of the gradient?

If my concerns are well addressed, I would be happy to raise my score.

---

> ### Author Rebuttal · Authors · 2024-08-06
>
> We thank the reviewer mptp for the time and valuable feedback!
>
> **W1: Since the paper primarily claims the efficiency of the proposed first-order algorithms, the authors should discuss the scale of problems these algorithms are suitable for.**
>
> A: From the perspective of scale, compared to deep AUC maximization, the application of meta-learning is comparably a larger-scale problem due to the vast number of tasks and number of variables. The need to calculate the second-order derivative introduces high computational cost to second-order methods (e.g. MAML), making it hard to scale in this problem. However, as a first-order method, our algorithm can scale well in the meta-learning setting. In Table 1 we show the memory consumption of our MemCS compared with MAML with a different number of lower-level update steps. As a second-order method, MAML's memory cost increases with lower-level update steps, whereas our MemCS algorithm maintains stable memory consumption. Additionally, Table 2 (referenced in the answer to Question 2) shows that the average iteration time of MemCS is only one-third that of MAML. This demonstrates that our algorithm is more efficient in terms of both memory and computational costs. We will conduct further studies on larger-scale problems in our future research.
>
> Table 1. Memory cost in robust meta-learning application.
> | Lower-level update step number |   MAML   |  MemCS   |
> |:------------------------------:|:--------:|:--------:|
> |             t = 10             | 8560 MB  | 7762 MB  |
> |             t = 15             | 12006 MB | 7739  MB |
> |             t = 20             | 15478 MB | 7632 MB  |
> |             t = 25             | 18922MB  | 7817 MB  |
> |             t = 30             | 22368 MB | 7444 MB  |
>
> Thanks for pointing out the typo and please see the response to the gradient calculation in **Answer to Q4**.
>
> **Answer to Q1:** This is a good point. Our methods can handle non-convex optimization problems, because the overall objective function $\Phi(x):=F(x,y^*(x),z^*(x))$ is generally non-convex with respect to $x$. The concavity/convexity assumptions are made only for the maximization problem in $y$ and the minimization problem in $z$. These two optimization problems are much easier to solve than optimizing $\Phi(x)$ and usually satisfy the convexity property. This can be more clearly seen in our applications of meta-learning and deep AUC maximization. In our application of the robust meta-learning setting, the lower-level problem is optimizing a linear layer with cross-entropy loss, which satisfies the strong convexity assumption. The maximization in the upper-level minimax problem is a combination of a negative hinge function and a linear function, making it a concave function. In our deep AUC maximization application, the lower-level function uses square loss, which also satisfies the strong convexity assumption. The maximization in the upper-level minimax problem is a negative quadratic function, which is concave.
>
> It is also worth mentioning that our assumptions, such as lower-level strong convexity, Lipschitz continuity, and bounded variance, are primarily made for theoretical analysis. These assumptions have also been adopted in existing theoretical works on bilevel optimization (e.g., [1][2][3]) and minimax bilevel optimization (e.g., [4]). For practical use, our algorithms can also be applied to broader applications where these assumptions may not hold; however, in such cases, the convergence guarantee may not be developed.
>
> We will add the above discussions and the applicable score of our algorithms in the revision.
>
> [1] Approximation methods for bilevel programming.
>
> [2] Bilevel optimization: Convergence analysis and enhanced design.
>
> [3] A framework for bilevel optimization that enables stochastic and global variance reduction algorithms.
>
> [4] Multi-block min-max bilevel optimization with applications in multi-task deep AUC maximization.
>
> **Answer to Q2:** Thank you for the suggestion. We have conducted a further evaluation of the average iteration time for our proposed algorithms and the second-order method in a robust meta-learning setting, as detailed in Table 2. The results demonstrate that our algorithms run significantly faster than the second-order method.
>
> Table 2. Average iteration time of different algorithms on the robust meta-learning setting.
> |         Method         | Second-order method | FOSL  | MemCS |
> |:----------------------:|:-------------------:|:-----:|:-----:|
> | Average iteration time |        9.41s        | 1.42s | 3.15s |
>
> **Answer to Q3:** The reasons for the improved performance of FOSL and MemCS differ between the two applications. For deep AUC maximization, the answer is yes. In the implementation of the baseline method (mAUC-CT), second-order information is discarded, whereas we leverage this information through the calculation of the hyper-gradient. For robust meta-learning, our method is specifically designed to address the minimax bilevel optimization problem. This design allows for easy integration with rank-based methods, enhancing the robustness of training and consequently leading to better performance.
>
> **Answer to Q4:** This is a good question. In this work, we use $\nabla \mathcal{L}^*(x) = \nabla_x F(x,y^*(x),z_{\lambda}^*(x)) + \lambda(\nabla_x G(x,z_{\lambda}^*(x)) - \nabla_x G(x,z^*(x)))$ as the gradient to update $x$, which is a first-order approximate of the vanilla hypergradient $\nabla \Phi(x)$. Proposition 4.7 shows that the gap between $\nabla \mathcal{L}^*(x)$ and $\nabla \Phi(x)$ is proportional to $1/\lambda$, and hence can be made sufficiently small by choosing the regularization parameter $\lambda$ large enough. Since $y^*(x),z_{\lambda}^*(x),z^*(x)$ can not be obtained directly, we use $\nabla_x \mathcal{L}(x_{t},y_{t},z_{t},v_{t}) = \nabla_x F(x_t,y_t,z_t) + \lambda(\nabla_x G(x_t,z_t) - \nabla_x G(x_t,z_t))$ to approximate $\nabla \mathcal{L}^*(x)$ at the $(t+1)_{th}$ iteration.

---

> > ### Comment · Reviewer_mptp · 2024-08-10
> > **Response**
> >
> > Thanks for your detailed explanation. I will raise my score to 6.

---

> > > ### Author Response · Authors · 2024-08-11
> > >
> > > Dear Reviewer mptp,
> > >
> > > Thanks so much for your updates and for raising your score. We are happy that our responses clarify your questions. We will take your suggestions into our revision.
> > >
> > > Best, Authors

---

### Official Review · Reviewer_pAAu · 2024-07-09

**Soundness:** 2
**Presentation:** 3
**Contribution:** 2
**Rating:** 5
**Confidence:** 2

**Summary:**

This work proposes two novel fully first-order algorithms, named FOSL and MemCS, for multi-block minimax bilevel optimization problems. Specifically, the authors reformulate the lower-level problem as a value-function-based constraint and transform the minimax bilevel optimization into a surrogate minimax problem. FOSL and MemCS are proposed to solve the surrogate minimax problem by alternately updating the parameters through SGD. Theoretical analysis of the convergence is conducted and extensive experiments on deep AUC maximization and rank-based robust meta-learning show the effectiveness of the proposed method.

**Strengths:**

The proposed method is sound and efficient.

**Weaknesses:**

1. I have concerns about the soundness of reformulating the minimax bilevel optimization as a minimax problem in Eq. 2. I hope the authors can conduct more analysis or provide some references that utilize the same technique.

2. I have a question about Proposition 4.9. Will it guarantee that $E|| y_{t+1}-y^*(x_{t+1}) || - E|| y_{t}-y^*(x_{t}) || \le 0$?

**Questions:**

Please see the weakness.

**Limitations:**

No potential negative societal impact.

---

> ### Author Rebuttal · Authors · 2024-08-06
>
> We thank the reviewer pAAu for the time and valuable feedback!
>
> **W1: I have concerns about the soundness of reformulating the minimax bilevel optimization as a minimax problem in Eq.(2). I hope the authors can conduct more analysis or provide some references that utilize the same technique.**
>
> A: This is a good question. The lower level problem in Eq.(1) aims to find an optimal solution $z_i^*$ of $g_i(x,z_i)$. In Eq. (2), the lower-level problem is converted into a constraint $g_i(x,z_i) - g_i(x,z_i^*) \leq 0$. Since $g_i(x,z_i)$ has a unique minimizer $z_i^*$, this constraint is satisfied if and only if $z_i=z_i^*$. As a result, the objective function in Eq. (2) becomes $\min_x\max_y \frac{1}{n}\sum_{i=1}^nf_i(x,y,z_i^*)$, which is the same as that in Eq. (1). Similar techniques also appear in [1][2][3].
>
> [1] On solving simple bilevel programs with a nonconvex lower level program.
>
> [2] A fully first-order method for stochastic bilevel optimization.
>
> [3] A value-function-based interior-point method for non-convex bi-level optimization.
>
> **W2: I have a question about Proposition 4.9. Will it guarantee that** $E\\|y_{t+1}-y^*(x_{t+1})\\| - E\\|y_{t}-y^*(x_{t})\\| \leq 0$?
>
> A: Proposition 4.9 does not guarantee that $\mathbb{E}\\|y_{t+1}-y^*(x_{t+1})\\| - \mathbb{E}\\|y_{t}-y^*(x_{t})\\| \leq 0$ for **all** $t$. This is due to the existence of the positive error terms $O(\frac{\eta_y^2}{|I_t|})(\sigma_f^2+\sigma_{th}^2)$, $\mathcal{O}(\eta_y)\frac{1}{n}\sum_{i=1}^n\mathbb{E}\\|v_{i,t}-z_i^*(x_t)\\|^2$, $\mathcal{O}(\eta_x^2/\eta_y)\mathbb{E}\\|\tilde{h}_x^t\\|^2$, $\mathcal{O}(\eta_x^2)\mathbb{E}\\|h_x^t\\|^2$.
>
> These terms have no direct correlation with the negative term $-\mathcal{O}(\eta_y)\mathbb{E}\\|y_{t} - y^*(x_{t})\\|^2$, and hence it cannot guarantee that their summation together is negative.
> Instead, this inequality can be understood as follows. By rearranging Proposition 4.9, we have
> $$
> \mathbb{E}\\|y_{t+1}-y^*(x_{t+1})\\|^2 \leq (1-\mathcal{O}(\eta_y))\mathbb{E}\\|y_{t}-y^*(x_{t})\\|^2 + O\bigg(\frac{\eta_y^2}{|I_t|}\bigg)(\sigma_f^2+\sigma_{th}^2) +\mathcal{O}(\eta_y)\frac{1}{n}\sum_{i=1}^n\mathbb{E}\\|v_{i,t}-z_i^*(x_t)\\|^2 + \mathcal{O}(\eta_x^2/\eta_y)\mathbb{E}\\|\tilde{h}_x^t\\|^2 + \mathcal{O}(\eta_x^2)\mathbb{E}\\|h_x^t\\|^2.
> $$
>
> This indicates the decay of the optimality distance $\mathbb{E}\\|y_{t}-y^*(x_{t})\\|^2$ over iterations. The positive error terms $O(\frac{\eta_y^2}{|I_t|})(\sigma_f^2 + \sigma_{th}^2)$ and $\mathcal{O}(\eta_x^2)\mathbb{E}\\|h_x^t\\|^2$ are negligible in the final convergence analysis because they are proportional to the square of the stepsizes $\eta_x$ and $\eta_y$. By choosing sufficiently small stepsizes, these terms can be made sufficiently small (e.g., see the proofs of Theorem 4.10). The error term $\mathcal{O}(\eta_y)\frac{1}{n}\sum_{i=1}^n\mathbb{E}\\|v_{i,t}-z_i^*(x_t)\\|^2$ can be merged into the descent term of iterate $v_{i,t}$ (see the details in Lemma E.3). The error term $\mathcal{O}(\eta_x^2/\eta_y)\mathbb{E}\\|\tilde{h}_x^t\\|^2$ can be canceled out by the negative term $-\mathbb{E}\\|\tilde{h}_x^t\\|^2$ in Proposition 4.8 for stepsize $\eta_x$ sufficiently small.

---

### Official Review · Reviewer_Mt97 · 2024-07-11

**Soundness:** 3
**Presentation:** 3
**Contribution:** 2
**Rating:** 6
**Confidence:** 3

**Summary:**

This work studies the multi-block minimax bilevel optimization problem. To address the high computation costs and high memory consumption issues of existing algorithms, this work proposes two fully first-order algorithms, i.e., FOSL and MemCS. Specifically, the authors convert the original minimax bilevel problem into a simple minimax problem and propose FOSL (first-order single-loop algorithm) to solve the optimization problem efficiently. Then, they introduce MemCS with cold-start initialization, which avoids storing the weights of blocks and reduces memory consumption. Moreover, the authors also provide convergence analysis for the proposed algorithms. Experimental results on the benchmark datasets show that the proposed method can achieve better or comparable performance than the baseline.

**Strengths:**

1. The authors convert the original minimax bilevel problem into a simple minimax problem and solve it by first-order single-loop algorithm.
2. The authors provide a comprehensive convergence analysis on the proposed method.
3. The proposed method is easy to converge and outperforms the baselines on CIFAR-100, CelebA and OGBG-MolPCBA.

**Weaknesses:**

1. AUC-CT avoids the calculation of second-order matrices and shows comparable efficiency to the proposed first-order algorithm, which reduces the contribution of this work to improving algorithm efficiency. It would be better to provide more discussions.
2. It would be better to provide the memory consumption of baselines for better comparisons.
3. The proposed method only achieves comparable performance to mAUC-CT on CheXpert. It would be better to provide more analysis.

**Questions:**

My main concern is about the contribution of this work to computational efficiency (especially compared to AUC-CT).

**Limitations:**

I cannot find the discussion about the limitations of this work.

---

> ### Author Rebuttal · Authors · 2024-08-06
>
> We thank the reviewer Mt97 for the time and valuable feedback!
>
> **W1: AUC-CT avoids the calculation of second-order matrices and shows comparable efficiency to the proposed first-order algorithm, which reduces the contribution of this work to improving algorithm efficiency. It would be better to provide more discussions.**
>
> A: In the theoretical sections of [1], the authors proposed approximating the hypergradient by estimating second-order matrices. However, in their experiments, they discarded the second-order information for implementation efficiency, leading to a larger approximation error. In contrast, we proposed to address the minimax bilevel optimization problem in a first-order manner, proving that our algorithms maintain a bounded approximation error. This also explains why our method achieves a higher accuracy than AUC-CT in the experiments.
>
> [1] Multi-block min-max bilevel optimization with applications in multi-task deep AUC maximization.
>
> **W2: It would be better to provide the memory consumption of baselines for better comparisons.**
>
> A: Thank you for bringing up this point. We evaluated the memory cost of our MemCS method compared to the baseline method in the robust meta-learning setting on the Mini-ImageNet dataset, using the same training configuration as outlined in Table 1. Our results indicate that the MemCS method achieves better performance and robustness than MAML with lower computational cost. Additionally, as the number of lower-level update steps increases, MAML’s memory cost rises significantly, while MemCS maintains consistent memory usage, demonstrating the scalability of our first-order algorithm. Due to limitations in time and computational resources, we plan to include a more detailed comparison of different settings in a future revision of our paper.
>
> Table 1. Memory cost in robust meta-learning application.
> | Lower-level update step number |   MAML   |  MemCS   |
> |:------------------------------:|:--------:|:--------:|
> |             t = 10             | 8560 MB  | 7762 MB  |
> |             t = 15             | 12006 MB | 7739  MB |
> |             t = 20             | 15478 MB | 7632 MB  |
> |             t = 25             | 18922MB  | 7817 MB  |
> |             t = 30             | 22368 MB | 7444 MB  |
>
> **W3: The proposed method only achieves comparable performance to mAUC-CT on CheXpert. It would be better to provide more analysis.**
>
> A: Thank you for highlighting this phenomenon. The mAUC-CT[1] implementation employs a simpler structure that directly avoids second-order derivative computations, which may be advantageous for the large-scale CheXpert dataset. However, this simplification introduces an additional approximation error, resulting in the mAUC-CT method experiencing significant variance ($\pm 0.1495$), as shown in Table 1 of the main text. In contrast, our algorithm achieves a much smaller variance ($\pm 0.0051$) compared to mAUC-CT.
>
> [1] Multi-block min-max bilevel optimization with applications in multi-task deep AUC maximization.

---

### Official Review · Reviewer_iRgq · 2024-07-13

**Soundness:** 3
**Presentation:** 3
**Contribution:** 3
**Rating:** 6
**Confidence:** 3

**Summary:**

The paper introduces FOSL and MemCS that are two new first-order algorithms for multi-block minimax bilevel optimization, demonstrating superior sample complexities and robust performance in empirical evaluations on deep AUC maximization and robust meta-learning applications.

**Strengths:**

The paper introduces FOSL, a novel fully first-order single-loop algorithm for minimax bilevel optimization, simplifying the problem structure and achieving competitive sample complexity without second-order computations. MemCS is proposed as a memory-efficient method with cold-start initialization, addressing challenges in scenarios with numerous blocks and demonstrating improved sample complexity compared to traditional methods.

**Weaknesses:**

1. The assumptions look quite strong, e.g., Assumption 5.3 requires strong convexity, Assumption 5.4 requires several boundnesses on the derivatives and high-order derivatives, which usually are not satisfied in practice.
2. It appears that the memory costs have not been compared.

**Questions:**

See weakness.

**Limitations:**

Not discussed.

---

> ### Author Rebuttal · Authors · 2024-08-06
>
> We thank the reviewer iRgq for the time and valuable feedback!
>
> **W1: The assumptions look quite strong, e.g., Assumption 5.3 requires strong convexity, and Assumption 5.4 requires several boundnesses on the derivatives and high-order derivatives, which usually are not satisfied in practice.**
>
> A: Thanks for your question! Assumptions 5.3 and 5.4 have been widely adopted by existing works on bilevel optimization, such as [1][2][3]. The same set of assumptions in our paper have also been made by other studies on multi-block minimax bilevel optimization such as [4].
> This is also easily verified in our applications of robust meta-learning and deep AUC maximization. In the robust meta-learning setting, the lower-level problem is optimizing a linear layer with cross-entropy loss, which satisfies the strong convexity assumption. The maximization in the upper-level minimax problem is a combination of a negative hinge function and a linear function, making it a concave function. In deep AUC maximization application, the lower-level function uses square loss, which also satisfies the strong convexity assumption. The maximization in the upper-level minimax problem is a negative quadratic function, which is concave. We intend to investigate relaxed assumptions in our future study.
>
> [1] Approximation methods for bilevel programming.
>
> [2] Bilevel optimization: Convergence analysis and enhanced design.
>
> [3] A framework for bilevel optimization that enables stochastic and global variance reduction algorithms.
>
> [4] Multi-block min-max bilevel optimization with applications in multi-task deep AUC maximization.
>
> **W2: It appears that the memory costs have not been compared.**
>
> A: Thank you for highlighting this point. We evaluated the memory cost of our MemCS method against the baseline method in the robust meta-learning setting on the Mini-ImageNet dataset, using the same training configuration as shown in Table 1 below. Our results demonstrate that the MemCS method achieves better performance and robustness compared to MAML, with lower computational costs. Additionally, as the number of lower-level update steps increases, MAML’s memory cost rises significantly, whereas our MemCS method maintains consistent memory usage, showcasing the scalability of our first-order algorithm. Due to limitations in time and computational resources, we will provide a more detailed comparison of different settings in a future revision of our paper.
>
> Table 1. Memory cost in robust meta-learning application.
> | Lower-level update step number |   MAML   |  MemCS   |
> |:-----------------------:|:--------:|:--------:|
> |         t = 10          | 8560 MB  | 7762 MB  |
> |         t = 15          | 12006 MB | 7739  MB |
> |         t = 20          | 15478 MB | 7632 MB  |
> |         t = 25          | 18922MB  | 7817 MB  |
> |         t = 30          | 22368 MB | 7444 MB  |

---

### Author Response · Authors · 2024-08-13

Dear Reviewers,

Thank you for taking the time to review our paper and for your valuable feedback. We have provided detailed responses to your questions and would greatly appreciate it if you could let us know if our responses adequately address your concerns, as the discussion period is approaching its end.

Best regards,

The Authors

---

### Decision · Program_Chairs · 2024-09-25

**Decision:**

Accept (poster)

**Comment:**

This submission studies the multi-block minimax bilevel optimization problem. Two fully first-order algorithms, FOSL and MemCS, are proposed to address the high computation costs and high memory consumption issues of existing algorithms. The authors convert the original minimax bilevel problem into a simple minimax problem and propose FOSL (first-order single-loop algorithm) to solve the optimization problem efficiently. Then, the authors introduce MemCS with cold-start initialization to avoid storing the weights of blocks and reduce memory consumption.  The authors provide convergence analysis for the proposed algorithms. Experimental results on the benchmark datasets show that the proposed method can achieve better or comparable performance than the baselines.